# Feel-Good Thompson Sampling for Contextual Bandits: a Markov Chain Monte Carlo Showdown

**Emile Anand**[*][†]
Georgia Institute of Technology
School of Computer Science
Atlanta, GA, 30308
emile@gatech.edu

**Sarah Liaw**[*][‡]
Havard University
School of Engineeering and Applied Sciences
Cambridge, MA, 02138
sliaw@g.harvard.edu

## Abstract

Thompson Sampling (TS) is widely used to address the exploration/exploitation tradeoff in contextual bandits, yet recent theory shows that it does not explore aggressively enough in high-dimensional problems. Feel-Good Thompson Sampling (FG-TS) addresses this by adding an optimism bonus that biases toward high-reward models, and it achieves the asymptotically minimax-optimal regret in the linear setting when posteriors are exact. However, its performance with *approximate* posteriors, common in large-scale or neural problems, has not been benchmarked. We provide the first systematic study of FG-TS and its smoothed variant (SFG-TS) across fourteen real-world and synthetic benchmarks. To evaluate their robustness, we compare performance across settings with exact posteriors (linear and logistic bandits) to approximate regimes produced by fast but coarse stochastic-gradient samplers. Ablations over preconditioning, bonus scale, and prior strength reveal a trade-off: larger bonuses help when posterior samples are accurate, but hurt when sampling noise dominates. FG-TS generally outperforms vanilla TS in linear and logistic bandits, but tends to be weaker in neural bandits. Nevertheless, because FG-TS and its variants are competitive and easy-to-use, we recommend them as baselines in modern contextual-bandit benchmarks. Finally, we provide source code for all our experiments in https://github.com/SarahLiaw/ctx-bandits-mcmc-showdown.

## 1 Introduction

The stochastic multi-armed bandit model (Berry and Fristedt, 1985; Bubeck and Cesa-Bianchi, 2012; Lattimore and Szepesvári, 2020) is one of the most prevalent frameworks for sequential decision making under uncertainty. Among its variants, the contextual bandit (Langford and Zhang, 2007) has enabled vast breakthroughs in numerous real-world applications such as recommendation (Yue et al., 2012), predictive control (Lin et al., 2023a), healthcare (Durand et al., 2018), and prompt optimization (Dwaracherla et al., 2024). In each round, the contextual bandit observes a $d$-dimensional feature vector (the "context") for each of its arms, pulls one of them, and receives a reward. The bandit's goal is to choose actions to maximize the total reward over a finite horizon $T$. This limited bandit feedback highlights the exploration/exploitation dilemma: should one choose the myopically better arm to maximize an immediate reward, or an under-sampled arm to potentially improve future rewards?

Hence, designing efficient deep exploration algorithms has been a key aspect of contextual bandit research (Bubeck et al., 2009; Chu et al., 2011; Russo and Roy, 2016; Xu et al., 2022a). One popular

---

[*]These authors contributed equally to this work.
[†]Work done while an intern at Windsurf and Cognition AI.
[‡]Work done while an undergraduate at Caltech.

39th Conference on Neural Information Processing Systems (NeurIPS 2025) Track on Datasets and Benchmarks.

approach is *Thompson Sampling* (TS), which computes the posterior distribution of each arm being optimal for the context, and samples an arm from this distribution (Thompson, 1933). TS has found practical success due to its ease of implementation (Chapelle and Li, 2011; Jose and Moothedath, 2024) and impressive empirical performance. However, in high-dimensional scenarios, it is believed that TS produces suboptimal outcomes (Abbasi-Yadkori et al., 2011; Abeille and Lazaric, 2017; Agrawal and Goyal, 2013; Hamidi and Bayati, 2020) due to insufficient exploration.

To correct this, Zhang (2022) introduced Feel-Good Thompson Sampling (FG-TS) which proposes a modified likelihood function in TS by adding a *feel-good* bonus term that enforces more aggressive exploration. In the fundamental setting of linear contextual bandits, where the information-theoretic regret lower bound is $\Omega(d\sqrt{T})$ (Russo and Roy, 2016), it had been shown that TS achieves $O(d\sqrt{dT})$ frequentist regret bounds (Agrawal and Goyal, 2013), and Hamidi and Bayati (2020) provided matching lower bounds. Conversely, FG-TS obtains the optimal regret of $O(d\sqrt{T})$ (Zhang, 2022). Recently, Huix et al. (2023) adjusted the algorithm by smoothening the bonus term, resulting in the smoothed Feel-Good Thompson Sampling (SFG-TS) algorithm. This, in turn, smoothens the posteriors, making FG-TS amenable to Markov Chain Monte-Carlo (MCMC) methods which have been extensively studied for their use in efficient sampling (Diaconis, 2009; Gelman et al., 2013).

MCMC enables tractable Thompson Sampling because the sequence of posterior distributions can otherwise be computationally expensive to sample from in practice, as it requires inverting a $d \times d$ matrix, which takes $O(d^3)$ time using Cholesky decomposition. Moreover, while one rarely has access to the full posterior, it is generally easier to efficiently approximate the posterior by sampling from it. This connection between contextual bandits and MCMC has been widely explored (May et al., 2012; Munos, 2014; Riquelme et al., 2018), and there is a line of recent work in the literature which uses MCMC methods at each round to obtain approximate samples from the target posterior distribution in Thompson sampling (Mazumdar et al., 2020; Xu et al., 2022a,b; Yi et al., 2024).

While the effect of approximating the posterior distribution in Thompson Sampling is well-studied in the literature (Riquelme et al., 2018), we study the problem of what happens when the posterior approximations are inaccurate in Feel-Good Thompson Sampling from an empirical standpoint. While adding an optimism bonus in FG-TS theoretically improves the theoretical regret for linear contextual bandits, understanding the impact of the degree of optimism and the quality of the approximated posterior is critical: in certain instances, a mismatch in posteriors may not hurt in terms of decision making, and we will still end up with good decisions. Unfortunately, in other cases, the mismatch in posteriors together with its induced feedback loop, coupled with the mismatch in the degree of optimism, could degenerate in a significant loss of performance. We would like to understand the key aspects that influence either outcome. This is an important practical concern, since Zhang (2022)'s theoretical analysis of FG-TS assumes exact posteriors to favor simplicity of the analysis, but what is its impact? Therefore, the main question we address in this paper is how approximate model posteriors affect the performance of FG-TS (and its smoothed variants) in contextual bandits.

In this paper, we develop a benchmark for exploration methods, and provide extensions to deep neural networks. We examine a wide class of exploration algorithms that have been studied for the past two decades (Anand et al., 2025b; Bubeck et al., 2009; Kveton et al., 2020; Lin et al., 2023b, 2024; Mnih et al., 2015; Riquelme et al., 2018; Xu et al., 2022a): we compare a variety of well-established and recent posterior sampling algorithms for approximating the posterior distribution, under the lens of FG-TS and its smoothed variants for contextual bandits. This includes a novel optimal-regret Hamiltonian Monte-Carlo algorithm for SFG-TS, building on the works of Huix et al. (2023); Zhang (2022). We also provide a theoretical analyses of the runtime complexities, convergence guarantees, and regret bounds in Appendix D. Finally, all code and implementations to reproduce the experiments will be available open-source to provide a reproducible benchmark for future development.[1]

**Contributions.** Our key contributions are outlined below.

1. **Linear and logistic contextual bandits.** We compare the performance of FG-TS (and its smoothed variants) with TS in synthetic (and wheel) datasets through a variety of MCMC algorithms, such as Langevin Monte Carlo (LMC), Metropolis-Adjusted Langevin Algorithms (MALA), and Hamiltonian Monte Carlo (HMC). We explore the impact of preconditioning, damping, stochastic variance-reduced gradient methods, and modifying the levels of feel-good bonus' and inverse

---

[1]Our PyTorch implementation is available here and our pip installation instructions are available here.

temperature parameters in the MCMC algorithms. Finally, we compare the performance of these methods with well-studied algorithms, such as $\epsilon$-Greedy, Lin-UCB, Neural-Linear, and Lin-TS.

2. **Neural contextual bandits.** We study the performance of FG-TS and SFG-TS with TS in real-world datasets, through the Langevin Monte Carlo (LMC) MCMC algorithm. In the neural setting, the additional computations done by more sophisticated MCMC algorithms render the computations too expensive to be practical. Nevertheless, we compare the performance of these methods with well-studied algorithms, such as Neural-$\epsilon$-Greedy and Neural UCB.

Our experiments show that FG-TS performs best (through the LMC algorithm) in the linear contextual bandit setting, and that SFG-TS performs best (through the MALA algorithm) in the logistic setting. However, FG-TS (and its smoothed variants) tends to be weaker in the neural contextual bandit setting. Perhaps surprisingly, we observe several instances where its performance degenerates rapidly, where TS remains far more competitive. A priori, one might expect FG-TS to be more amenable to real-world datasets because it favors more aggressive exploration (e.g., by escaping local optima or handling sparse or heterogeneous contexts); however, our experiments indicate that this is not the case.

In Section 2, we discuss TS and its Feel-Good variants, and present the contextual bandit problem. We introduce the various algorithmic approaches we consider for approximating the posterior distributions in Section 3. Finally, we describe our experimental setup and discuss our results in Sections 4 and 6.

## 2 Decision Making via Feel-Good Thompson Sampling

Many sequential decision-making problems can be framed through the contextual bandits framework. In each round $t \in [T]$, an agent observes an action set (or a context) $\mathcal{X}_t \subseteq \mathbb{R}^d$ and (based on its internal context-adaptive algorithm) chooses an arm or action represented by a feature vector $x_t \in \mathcal{X}_t$. A feedback reward $r_t := r_t(x)$ is then generated and returned to the algorithm. In contextual bandit problems, the mean reward of an action $x \in \mathbb{R}^d$ is given by a reward generating function $f_\theta(x)$ and the observed reward is $r_t(x) = f_{\theta^*}(x) + \xi_t$, where $\theta^* \in \mathbb{R}^m$ is an unknown weight parameter shared across all the arms and $\{\xi\}_{t>0}$ is a random noise sequence. For instance, in linear contextual bandits (Agrawal and Goyal, 2013; Sivakumar et al., 2020), $\theta^* \in \mathbb{R}^d$ and $f_{\theta^*}(x) = x^\top \theta^*$; in generalized linear bandit (Filippi et al., 2010; Kveton et al., 2020), we have $f_{\theta^*}(x) = \mu(x^\top \theta^*)$ for some link function $\mu(\cdot)$; and for neural contextual bandits (Terekhov, 2025; Xu et al., 2018; Zhang et al., 2021), $f_{\theta^*}(x)$ is a neural network, where $\theta^*$ is the concatenation of all weight parameters and $x$ is the input.

At time $T$, the total reward gained by the algorithm is $r = \sum_{t=1}^{T} r_t$. Then, the objective of any bandit algorithm is to maximize $r$, which is equivalent to minimizing the *cumulative pseudo regret* (Lattimore and Szepesvári, 2020; Lin et al., 2022) $R(T) = \mathbb{E}[\sum_{t=1}^{T} r(x_t^*) - r(x_t)]$. Here, $x_t^*$ is the arm chosen by the optimal hindsight policy which selects actions by maximizing the expected reward.

**Thompson Sampling.** TS (Thompson, 1933) as presented in Algorithm 1, is a bandit algorithm that is popular for its elegance and practical efficiency (Agrawal and Goyal, 2013; Chapelle and Li, 2011). TS requires that one only needs to sample from the posterior distribution of the reward function. At each round, it draws a sample and takes a greedy action under the optimal policy for the sample. The posterior distribution is then updated after the result of the action is observed. Algorithm 1 considers the standard setting where the likelihood is given by $L^{\mathrm{TS}}(\theta, x, r) = \eta(f_\theta(x) - r)^2$ for some $\eta > 0$.

---

**Algorithm 1** Thompson Sampling for Contextual Bandits

---

**Input:** Likelihood function $L^{\mathrm{TS}}(\theta, x, r)$, reward model function $f_\theta(x), \theta_0 \sim p_0(\cdot)$
**for** time $t = 1, 2, \ldots, T$ **do**
    Observe context $\mathcal{X}_t \subseteq \mathbb{R}^d$,
    Let $\mathcal{L}_t(\theta) = \sum_{s=1}^{t-1} L^{\mathrm{TS}}(\theta, x_s, r_s)$ and sample $\theta_t \sim \exp(-\mathcal{L}_t(\theta))p_0(\theta)$,
    Play arm $x_t = \arg\max_{x \in \mathcal{X}_t} f_{\theta_t}(x)$ and observe reward $r_t$.

---

**Feel-Good Thompson Sampling.** FG-TS (Zhang, 2022) extends Algorithm 1 by defining a *feel-good* likelihood. Given a constant $b$ and tuning parameter $\lambda > 0$, the feel-good likelihood is given by:

$$\mathcal{L}_t(\theta) = \sum_{s=1}^{t-1} L^{\mathrm{FG}}(\theta, x_s, r_s), \quad L^{\mathrm{FG}}(\theta, x, r) = \eta(f_\theta(x) - r)^2 - \lambda \min(b, f_\theta(x)), \quad (1)$$

**Smoothed Feel-Good Thompson Sampling**. SFG-TS (Huix et al., 2023) proposes the smoothening:

$$\mathcal{L}_t(\theta) = \sum_{s=1}^{t-1} L^{\text{SFG}}(\theta, x_s, r_s), \quad L^{\text{SFG}}(\theta, x, r) = \eta(f_\theta(x) - r)^2 - \lambda(b - \Phi_s(b - f_\theta^\star)), \quad (2)$$

where $f_\theta^\star = \max_{x \in \mathcal{X}} f_\theta(x)$ and $\Phi_s(u) = \log(1 + \exp(su))/s$, for $u \in \mathbb{R}$ and parameters $\lambda, \eta, b, s \in \mathbb{R}^+$. In linear contextual bandits, Zhang (2022) showed that FG-TS obtains the minimax-optimal regret of $O(d\sqrt{T})$, and Huix et al. (2023) shows that this regret guarantee is maintained for SFG-TS.

## 3  Algorithms

This section describes the different algorithmic design principles we considered in our simulations in Section 4. These algorithms include (Neural-)$\epsilon$-Greedy, Linear UCB, LinTS, Neural UCB, and our main focus: Markov Chain Monte Carlo methods, with its many variants.

**Linear Methods.** Linear methods follow the exact closed-form updates for Bayesian linear regression in linear contextual bandits (Chapelle and Li, 2011; Gelman et al., 2013). Let $\lambda > 0$ be a tunable parameter. We introduce the design matrix $\mathbf{V}_t$, the cumulative response vector $b_t$, and the ridge regression parameter $\widehat{\theta}_t$ given by $\mathbf{V}_t = \lambda \mathbf{I} + \sum_{s=1}^{t-1} x_s x_s^\top, b_t = \sum_{s=1}^{t-1} r_s x_s$, and $\widehat{\theta}_t = \mathbf{V}_t^{-1} b_t$ (respectively). At round $t$, LinUCB computes a confidence bound $p_t(x) = x^\top \widehat{\theta}_t + \alpha(x^\top \mathbf{V}_t^{-1} x)^{1/2}$, for some exploration parameter $\alpha > 0$, and picks $x_t = \text{argmax}_x p_t(x)$. On the other hand, LinTS samples $\theta_t \sim \mathcal{N}(\widehat{\theta}_t, v_t \mathbf{V}_t^{-1})$, where $v_t > 0$ is a scaling parameter, and picks $x_t = \text{argmax}_x x^\top \theta_t$. Finally, Riquelme et al. (2018) studies a Neural-Linear variation of these linear methods that applies Bayesian linear regression on top of the regression of the last layer of a neural network. Similarly, we further benchmark such neural-linear approaches for the neural MCMC contextual bandits settings.

**Neural Methods.** Neural methods replace the raw actions $x \in \mathbb{R}^d$ by a learned embedding $\phi_\theta(x) \in \mathbb{R}^m$ generated by a neural network with SGD-learned parameters $\theta$ (Xu et al., 2022a). Exploration is done only on the last-layer representation: let $\lambda > 0$ be a tunable parameter and $\phi_s = \phi_\theta(x_s)$ be the last-layer feature embedding. At each round $t$, NeuralUCB maintains the Gram matrix $\mathbf{Z}_t := \lambda \mathbf{I} + \sum_{s=1}^{t-1} \phi_s \phi_s^\top$ be the Gram matrix. It then computes the confidence bound (UCB score) $p_t(x) = r(x) + \alpha(\phi_t \mathbf{Z}_t^{-1} \phi_t)^{1/2}$ and picks the action $x_t = \arg\max_x p_t(x)$. Similarly, NeuralTS (Neural Thompson Sampling) considers weight uncertainty across all features of the neural network when updating its posteriors (Zhang et al., 2021): its mean is the neural network approximator, and its variance is built upon the neural tangent kernel (NTK) features of the neural network (Arora et al., 2019; Chaudhari et al., 2024; Jacot et al., 2018), given by $\widehat{\Theta}(x, x') := \left( \frac{\partial f(x, \theta)}{\partial \theta} \right) \left( \frac{\partial f(x', \theta)}{\partial \theta} \right)^\top$.

**Greedy Methods.** $\epsilon$-Greedy balances the exploration/exploitation tradeoff by selecting the arm whose current reward estimate is highest with probability $1 - \epsilon$, and selecting a random arm otherwise. Next, the algorithm updates its estimate (through a table or linear model) for that arm. Neural $\epsilon$-Greedy extends this by selecting a random action with probability $\epsilon$ for some decaying schedule of $\epsilon$.

### 3.1  Markov Chain Monte Carlo Methods

While drawing from the posterior distribution of the model parameters at each round is easy in conjugate linear-Gaussian models, it becomes intractable when moving to realistic, non-linear, or high-dimensional models (Huix et al., 2023; Xu et al., 2022b). In MCMC, only noisy gradient descent updates are needed for proper exploration in bandit problems. This is especially appealing for deep neural networks as one rarely has access to the full posterior, but can approximately sample from it.

---

**Algorithm 2** MCMC Thompson Sampling

---

**Input:** Loss function $\mathcal{L}_t(\theta)$, reward model function $f_\theta(x), \theta_{1,0} = 0, K_0 = 0$
**for** time $t = 1, \ldots, T$ **do**
    Observe context $\mathcal{X}_t \subseteq \mathbb{R}^d$
    Set $\theta_{t,0} = \theta_{t-1, K_{t-1}}$
    *// MCMC-TS replaces exact posterior sampling in Algorithm 1 with approximate posterior sampling.*
    **for** $k = 1, \ldots, K_t$ **do**
        MCMC update to compute $\theta_{t,k+1}$
    Play arm $x_t = \arg\max_{x \in \mathcal{X}_t} f_{\theta_{t,K_t}}(x)$ and observe reward $r_t$

---

**Langevin Monte Carlo (LMC).** Xu et al. (2022b) proposed a Langevin MCMC to approximate the unknown posterior distribution of the parameter $\theta^*$ upto a high precision: given a schedule of step-sizes $\{\eta_t\}_{t\geq 1}$ and inverse temperatures $\{\beta_t\}_{t\geq 1}$, LMC[2] samples a standard normal vector $\epsilon_{t,k} \sim \mathcal{N}(\mathbf{0}, \mathbf{I})$ and lets $\theta_{t,k+1} = \theta_{t,k} - \eta_t \nabla \mathcal{L}_t(\theta_{t,k-1}) + (2\eta_t \beta_t^{-1})^{1/2} \epsilon_{t,k}$. For sufficiently large epoch lengths $K_t$, this approximately converges to the posterior distribution $\pi_t(\theta) \propto e^{-\beta_t L_t(\theta)}$.

**Metropolis-Adjusted Langevin Algorithms (MALA).** The discretization error of LMC has a bias on the order of the step-size $\eta_t$. To correct the LMC discretization bias, Huix et al. (2023) adds a Metropolis filter at each iteration: with probability $1 - \min\{1, \frac{\exp(\mathcal{L}_t(\theta_{t,k}))}{\exp(\mathcal{L}_t(\theta_{t,k+1}))}\}$, set $\theta_{t,k+1} = \theta_{t,k}$.

**Hamiltonian Monte Carlo (HMC).** In many settings, HMC is believed to outperform other MCMC algorithms such as MALA (Huix et al., 2023) or LMC (Xu et al., 2022b), and is equipped with rich theoretical guarantees (Apers et al., 2024; Chen and Vempala, 2022). HMC moves by integrating the Hamiltonian dynamics $H : \mathbb{R}^d \times \mathbb{R}^d \to \mathbb{R}$ which captures the sum of the potential and kinetic energies of a particle as a function of its position $\theta \in \mathbb{R}^d$ and velocity $v \in \mathbb{R}^d$. We let $H(\theta, v) = \mathcal{L}_t(\theta) + \frac{1}{2}\|v\|^2$. $(\theta_t, v_t)$ follows a discretization of the *Hamiltonian curve* trajectory: $\frac{d\theta}{dt} = v, \frac{dv}{dt} = -\nabla \mathcal{L}_t(\theta)$. We give implementation details and theoretical guarantees in Appendix C.

**Stochastic Variance-Reduced Gradients.** Although the updates in the above MCMC algorithms are presented as a full gradient descent step plus an isotropic noise, we can replace the full gradient $\nabla L_t(\theta_{t,k-1})$ with a variance-reduced stochastic gradient (SVRG) of the loss function $L_t(\theta_{t,k-1})$ computed from a mini-batch $\mathcal{B}_k$ of data (Dubey et al., 2016; Welling and Teh, 2011). Let $\epsilon_t \sim \mathcal{N}(\mathbf{0}, \mathbf{I})$ and let $U(\theta)$ be the full-data loss. Then, the SVRG modification is given by:

$$\widetilde{\mu} = \nabla U(\widetilde{\theta}) = \nabla U_{\text{data}}(\widetilde{\theta}) + 2\alpha\|\widetilde{\theta}\|, \qquad g_k(\theta_t) = \nabla U_{\text{data}}^{(\mathcal{B}_k)}(\theta_t) - \nabla U_{\text{data}}^{(\mathcal{B}_k)}(\widetilde{\theta}) + \widetilde{\mu}, \qquad (3)$$

$$\theta_{t+1} = \theta_t - \eta_t \, g_k(\theta_t) + \sqrt{2\eta_t \beta^{-1}} \epsilon_t.$$

We show in Appendix C that SVRG maintains the theoretical guarantees of the full-gradient version.

**Preconditioning.** Preconditioning can improve the convergence rate of the MCMC methods by a factor of $\kappa_t$, where $\kappa_t = \lambda_{\max}(\mathbf{V}_t)/\lambda_{\min}(\mathbf{V}_t)$ is the condition number of design matrix $\mathbf{V}_t$ (Li et al., 2016). Preconditioned-LMC samples a standard normal vector $\epsilon_{t,k} \sim \mathcal{N}(\mathbf{0}, \mathbf{I})$ and performs the modified LMC update: $\theta_{t,k+1} = \theta_{t,k} - \eta_t \mathbf{V}_t^{-1} \nabla L_t(\theta_{t,k-1}) + (2\eta_t \beta_t^{-1})^{1/2} \mathbf{V}_t^{-1/2} \epsilon_{t,k}$. The preconditioned variants of MALA and HMC follow similarly, with minutiae appearing in Appendix D.

**Underdamped Langevin Monte Carlo.** Discretized Langevin updates can be viewed as an over-damped discretization of the Langevin SDE. We benchmark the analogous underdamped (kinetic) LMC algorithm (ULMC) (Zhang et al., 2023). Like Hamiltonian Monte Carlo, ULMC tracks position $\theta$ and velocity $v$ with a damping coefficient $\gamma > 0$ and learning rate $\eta$. ULMC's update is given by the Ornstein-Uhlenbeck half-update mechanism given by:

$$v_{t,k+\frac{1}{2}} = (1-\gamma\eta)v_{t,k} - \eta\nabla U(\theta_{t,k}) + \sqrt{2\gamma\eta}\xi_{t,k}, \quad \theta_{t,k+1} = \theta_{t,k} + \eta v_{t,k+\frac{1}{2}}, \quad v_{t,k+1} = v_{t,k+\frac{1}{2}} \quad (4)$$

Here, $\xi_{t,k} \sim \mathcal{N}(\mathbf{0}, \mathbf{I})$ is sampled at each iteration. Notably, the higher-order velocity term $v$ allows ULMC to mix faster in rough landscapes (at the cost of extra memory and tuning); moreover, ULMC has been deeply analyzed and has robust theoretical guarantees (Zhang et al., 2023).

## 4    Empirical Evaluation

In this section, we present an empirical evaluation on two simulated bandit generators (linear and logistic) and a suite of real-world classification tasks. Experiments are conducted on AWS EC2 servers with P3/G4 GPUs (6 vCPUs), Google Cloud Compute servers with L4 GPUs and N2 instances (8 vCPUs), and NVIDIA A100 GPU.

### 4.1    Implementation Details

**Metrics.** To assess the performance of the algorithms, we report two metrics: cumulative regret ($R_T$) and simple regret ($\bar{R}_{\text{simp},T}$), where $R_T = \sum_{t=1}^{T}(r_t^\star - r_t)$, and $\bar{R}_{\text{simp},T} = \frac{1}{500}(\sum_{t=T-499}^{T}(r_t^\star - r_t) - R_{T-499})$, where $r_t^\star$ is the reward of the optimal policy. As in Riquelme et al. (2018), we include simple regret as it serves as a useful proxy for the quality of the final policy (Bubeck et al., 2009).

---

[2]The Euler-Maruyama discretization of the Langevin SDE: $d\theta_s = -\nabla \mathcal{L}_t(\theta_s) + \sqrt{2\beta_t^{-1}}dB(s)$

**Error Bars.** Our error bars report on the mean $\pm$ sample standard deviation over 10 seeds in the linear and logistic bandit settings, and over 5 seeds in the 6 real-world classification sets. As in Riquelme et al. (2018), we do not use a data buffer as the datasets are relatively small, and to avoid catastrophic forgetting. Therefore, all observations are uniformly sampled in each stochastic mini-batch.

## 4.2 Testbeds, Baselines, and Common Experimental Setup

**Linear Contextual Bandit.** To assess the performance in a setting with a tractable true posterior, we use a linear contextual bandit environment. At each round $t \in [T]$, the agent observes a context $\mathcal{X}_t \sim \mathcal{N}(\mathbf{0}_4, \mathbf{I}_4)$, chooses an action $x_t \in [K]$ with $K = 5$, and receives a noisy linear reward $r_t = \phi(\mathcal{X}_t, x_t)^\top \theta^\star + \varepsilon_t$, where $\varepsilon_t \sim \mathcal{N}(0, \sigma^2)$, $\sigma = 0.5$ and $\theta^\star \in \mathbb{R}^{20}$. We place a Gaussian prior, $\theta_0 \sim \mathcal{N}(\mathbf{0}_{20}, \sigma_0^2 \mathbf{I})$ with $\sigma_0 = 0.01$. The feature map $\phi : \mathbb{R}^4 \times [K] \to \mathbb{R}^{20}$ is the standard block concatenation $(\phi(\mathcal{X}_t, 0), \ldots, \phi(\mathcal{X}_t, K))$ where $\phi(\mathcal{X}_t, i) = e_i \cdot \mathcal{X}_t$ (where $e_i$ is the $i$'th standard basis vector). Now, $\theta^\star$ naturally decomposes into $k$ context-specific blocks. The time horizon is $T = 10\,000$. Since the likelihood and prior are both Gaussian, Thompson Sampling admits a closed-form posterior update, so we have a convenient ground-truth baseline for our approximate sampling methods.

**Wheel Bandit.** The wheel bandit, as defined in Riquelme et al. (2018), is a contextual bandit problem with the following structure. Let $d = 2$ be the context dimension and $\delta \in (0, 1)$ be the exploration parameter. Contexts are sampled uniformly at random from the unit circle in $\mathbb{R}^2$, denoted as $X \sim \mathcal{U}(D)$. The problem consists of $k = 5$ possible actions $a_1, \ldots, a_5$. Action $a_1$ provides reward $r \sim \mathcal{N}(\mu_1, \sigma^2)$, independent of context. In the inner region, where $\|X\| \leq \delta$, $a_2, \ldots, a_5$ are sub-optimal with $r \sim \mathcal{N}(\mu_2, \sigma^2)$, where $\mu_2 < \mu_1$. In the outer region, where $\|X\| > \delta$, the optimal action depends on the quadrant of the context $X = (X_1, X_2)$ where for $(X_1 > 0, X_2 > 0)$, $a_2$ is optimal, $(X_1 > 0, X_2 < 0)$, $a_3$ is optimal, $(X_1 < 0, X_2 < 0)$, $a_4$ is optimal, and $(X_1 < 0, X_2 > 0)$, $a_5$ is optimal. The optimal action provides $r \sim \mathcal{N}(\mu_3, \sigma^2)$ for $\mu_3 \gg \mu_1$, whereas other actions (including $a_1$) provide $r \sim \mathcal{N}(\mu_2, \sigma^2)$. We set $\mu_1 = 1.2$, $\mu_2 = 1.0$, $\mu_3 = 50.0$, and $\sigma = 0.01$, and let the horizon of the game be $T = 5000$. As the probability of a context falling in the high-reward region is $1 - \delta^2$, we expect algorithms to get stuck repeatedly selecting $a_1$ for large $\delta$.

**Logistic Contextual Bandit.** We further consider a logistic contextual bandit to introduce non-linear reward dependencies. With $T = 10,000$ and $K = 50$ arms, at each round $t \in [T]$, the learner observes a collection of arm-specific context vectors $\mathcal{X}_{t,a} \sim \mathcal{N}(\mathbf{0}_{20}, I_{20})$, where $a = 1, \ldots, 50$ each of which is then normalized to unit norm. The learner selects an arm $a_t \in [K]$ and obtains a Bernoulli reward $r_t \sim \mathrm{Bern}\big(\sigma\big(\phi(\widetilde{\mathcal{X}}_{t,a_t})^\top \theta^\star\big)\big)$, where $\sigma(u) = \frac{1}{1+e^{-u}}$, $\theta^\star \sim \mathcal{N}(\mathbf{0}_{20}, \mathbf{I}_{20})$ (scaled to unit norm), and $\phi : \mathbb{R}^{20} \to \mathbb{R}^{20}$ where $\phi(\mathcal{X}_t) = \mathcal{X}_t$ is the identity feature map extracting the 20-dimensional context for each arm. We place a Gaussian prior $\theta_0 \sim \mathcal{N}(\mathbf{0}_{20}, \sigma_0^2 \mathbf{I})$, with $\sigma_0 = 0.01$.

**Real-World Tasks.** Following Riquelme et al. (2018), we evaluate our MCMC Thompson Sampling algorithms on UCI datasets—ADULT, SHUTTLE, MAGICTELESCOPE, MUSHROOM, JESTER, COVERTYPE, and RESTAURANTRATINGS—plus the FINANCIAL dataset and two vision benchmarks, MNIST_784 and CIFAR-10 (Elmachtoub et al., 2017; Krizhevsky, 2009). These are standard datasets studied in contextual bandits and multi-agent settings (Chaudhari et al., 2024; Riquelme et al., 2018), as they span a variety of features such as reward stochasticity, size, and number of optimal actions. Following the protocol of Kveton et al. (2020); Riquelme et al. (2018), each $N$-class example $x \in \mathbb{R}^d$ is converted into $K$ arm–context vectors $x^{(1)}, \ldots, x^{(k)}$, where $x^{(i)} = x \cdot e_i$ where $e_i$ is the $i$'th standard basis vector of $\mathbb{R}^d$. Pulling arm $j$ returns reward 1 if $j$ matches the ground-truth label, and 0 otherwise. We provide statistics of the benchmark datasets in Appendix F.

**Network Architectures and Training Protocol.** We adopt the training protocol and hyperparameter search strategy of Xu et al. (2022b): each method chooses the better of a fully connected two-layer MLP (width 100) and four-layer MLP (width 50); the activation is selected from $\{\texttt{ReLU}, \texttt{LeakyReLU}\}$. Networks are updated for 100 SGD steps every round. FG-augmented variants inherit identical network architectures, optimizers, and training schedules as their non-FG counterparts to ensure fair comparisons. Unless stated otherwise, we run each bandit for $T = 10,000$ rounds (COVERTYPE: $T = 15,000$). Additional implementation minutiae appear in Appendix A of Riquelme et al. (2018).

# 5 Results

**Linear contextual bandits.** Table 1 (first 3 columns) reports the final cumulative regret (mean $\pm$ sample standard deviation over 10 seeds) in the synthetic linear bandit setting. Across both dimen-

Table 1: Final cumulative regret for synthetic datasets. We report mean $\pm$ sample std. The feel-good parameter $\lambda$ is set to 0.5 in each of the FG and SFG variants.

| Algorithm | Linear-20d$^{\star}$ $(\beta=10^3)$ | Linear-20d $(\beta=1)$ | Linear-40d $(\beta=10^3)$ | Logistic-20d $(\beta=10^3)$ · |
|---|---|---|---|---|
| LinUCB | $73.0 \pm 13.8$ | – | $126.3 \pm 19.3$ | $176.9 \pm 41.9$ |
| EpsGreedy | $19879 \pm 7454.3$ | – | $31170.6 \pm 2764.1$ | $2899.2 \pm 677.9$ |
| LinTS | $114.7 \pm 8.8$ | – | $204.6 \pm 19.1$ | $179.9 \pm 53.2$ |
| LMCTS | $62.6 \pm 9.5$ | $94.0 \pm 18.2$ | $129.1 \pm 16.1$ | $202.7 \pm 44.1$ |
| PLMCTS | $134.4 \pm 19.9$ | $132.3 \pm 28.2$ | $302.9 \pm 38.5$ | $889.7 \pm 248.0$ |
| FGLMCTS | $213.0 \pm 126.0$ | $296.7 \pm 174.0$ | $163.6 \pm 22.4$ | $184.8 \pm 38.0$ |
| PFGLMCTS | $204.1 \pm 105.3$ | $171.0 \pm 27.5$ | $330.2 \pm 30.8$ | $963.3 \pm 258.6$ |
| SFGLMCTS | $178.9 \pm 138.1$ | $241.0 \pm 143.8$ | $177.2 \pm 27.4$ | $221.7 \pm 42.1$ |
| PSFGLMCTS | $229.0 \pm 149.7$ | $206.8 \pm 90.2$ | $338.9 \pm 34.8$ | $771.9 \pm 274.5$ |
| SVRGLMCTS | $73.2 \pm 31.1$ | $103.6 \pm 23.8$ | $19236.8 \pm 15560.8$ | $216.0 \pm 69.7$ |
| HMCTS | $241.2 \pm 107.0$ | $226.7 \pm 78.9$ | $354.4 \pm 86.7$ | $449.8 \pm 86.2$ |
| PHMCTS | $90.0 \pm 9.2$ | $94.0 \pm 15.3$ | $162.2 \pm 12.5$ | $218.9 \pm 16.1$ |
| FGHMCTS | $262.5 \pm 85.7$ | $246.4 \pm 99.9$ | $399.0 \pm 112.5$ | $462.3 \pm 79.6$ |
| PFGHMCTS | $282.7 \pm 156.0$ | $291.6 \pm 171.5$ | $211.6 \pm 18.2$ | $411.1 \pm 65.4$ |
| SFGHMCTS | $395.9 \pm 504.7$ | $331.2 \pm 389.2$ | $416.9 \pm 58.1$ | $545.9 \pm 280.4$ |
| PSFGHMCTS | $284.1 \pm 147.4$ | $303.6 \pm 175.4$ | $212.2 \pm 26.1$ | $239.8 \pm 75.2$ |
| MALATS | $\mathbf{61.3} \pm 26.6$ | $\mathbf{56.5} \pm 11.2$ | $\mathbf{100.6} \pm 10.0$ | $\mathbf{194.0} \pm 76.9$ |
| FGMALATS | $220.0 \pm 159.9$ | $178.7 \pm 112.3$ | $139.7 \pm 16.1$ | $212.7 \pm 65.9$ |
| SFGMALATS | $189.3 \pm 135.3$ | $229.4 \pm 162.4$ | $142.1 \pm 19.5$ | $198.4 \pm 52.3$ |

*As implemented in Xu et al. (2022b).

sions $d \in \{20, 40\}$, MALATS consistently attains the lowest cumulative regret: $< 62$ when $d = 20$ (vs 62.6 for unadjusted LMCTS, 73.0 for LinUCB when $\beta = 10^3$) and around 100.6 when $d = 40$ (vs 129.1 for LMCTS, 126.3 for LinUCB). Adding a feel-good optimism bonus (FGMALATS and SFGMALATS) of $\lambda = 0.5$ does not improve regret in these Gaussian-conjugate settings. However, for smaller values of $\lambda$, such as $\lambda = 0.01$ in Table 4, the performance of SFGMALATS improves significantly to $56.2 \pm 22.8$, surpassing the other vanilla-TS variants. When $d = 40$, PHMCTS incurs $162.2 \pm 12.5$ and SVRGLMCTS incurs $19236.8 \pm 15560.8$, indicating that shifting the posterior due to preconditioning and variance-reduction distorts the quadratic curvature of the true posterior.

**Logistic contextual bandits** ($d = 20$). Table 1 (fourth column) reports the final cumulative regret in the logistic bandit setting. Although the reward is Bernoulli with a logistic link, in moderate signal-to-noise regimes, the posterior over the linear parameters is very nearly Gaussian (the log-likelihood is strongly convex and concentrates around its mode). The main benefit of HMC to explore 'curved'[3] or multimodal posteriors more effectively is not useful here, since we observe that simpler samplers like LMC and MALA closely match the closed-form Gaussian baselines. As expected, the exact-Gaussian methods—LinUCB ($176.9 \pm 41.9$) and LinTS ($179.9 \pm 53.2$)—lead overall. Among the MCMC-based samplers, FGLMCTS comes closest at $184.8 \pm 38.0$, effectively matching LinTS with a smaller variance, while MALATS performs worse at $194.0 \pm 76.9$.

**Neural contextual bandits.** Table 2 reports the final cumulative regret of a subset of our algorithms on several real-world problems, with complete results deferred to Appendix E.5. We additionally assess policy quality via simple regret in Table 3. When the posterior is approximated by SFG-TS, we observe a partial reversal: LMCTS becomes competitive with, and often outperforms, its feel-good extensions on the majority of the datasets. The only consistent advantage of the feel-good bonus emerges on MAGICTELESCOPE, where SFGLMCTS reduces cumulative regret by 27 and achieves the lowest simple regret of $82.8 \pm 8.9$. Meanwhile, NEURAL-$\epsilon$-GREEDY and NEURALUCB remain competitive, attaining the lowest regret on MUSHROOM and ADULT, respectively. Overall, the greedy neural methods yield the lowest simple regret on three tasks.

## 5.1 Ablation Studies

**Feel-Good Parameter.** Our comparison on the FG parameter $\lambda$ (ceteris paribus) in the linear setting is provided in Table 4, where $\beta = 10^3$, $d = 20$, and we sweep over $\lambda \in \{0, 0.01, 0.1, 0.5, 1.0\}$.

---

[3]This refers to the fact that the parameter $\theta$ lies in a Euclidean manifold on $\mathbb{R}^d$, rather than an affine subspace.

Table 2: Final cumulative regret after the full horizon (10k–15k steps, 5 seeds).

| Dataset | LMCTS | FGLMCTS | SFGLMCTS | Neural-$\epsilon$-Greedy | NeuralUCB |
|---|---|---|---|---|---|
| Adult | $2456.6 \pm 36.5$ | $3505.0 \pm 2257.5$ | $4505.6 \pm 2772.0$ | $2658.0 \pm 362.7$ | $\mathbf{2444.4} \pm 160.1$ |
| Covertype | $7594.0 \pm 892.0$ | $7567.8 \pm 454.5$ | $8006.0 \pm 1035.6$ | $\mathbf{4629.4} \pm 132.3$ | $4798.4 \pm 102.2$ |
| Magic Telescope | $2220.0 \pm 40.7$ | $2197.6 \pm 167.7$ | $2193.2 \pm 34.0$ | $\mathbf{2005.2} \pm 53.5$ | $2112.2 \pm 16.6$ |
| Mushroom | $324.6 \pm 102.6$ | $283.2 \pm 20.2$ | $440.6 \pm 89.5$ | $\mathbf{124.0} \pm 41.4$ | $145.6 \pm 25.2$ |
| Shuttle | $\mathbf{210.2} \pm 49.0$ | $214.4 \pm 51.6$ | $1503.0 \pm 2721.0$ | $372.4 \pm 425.8$ | $2981.2 \pm 4225.9$ |
| MNIST_784 | $2854.6 \pm 2945.9$ | $\mathbf{2542.6} \pm 2366.2$ | $2935.0 \pm 3349.5$ | $3248.0 \pm 1709.0$ | $5442.8 \pm 356.2$ |

Table 3: Simple regret statistics over the last 500 steps.

| Dataset | LMCTS | FGLMCTS | SFGLMCTS | Neural-$\epsilon$-Greedy | NeuralUCB |
|---|---|---|---|---|---|
| Adult | $121.6 \pm 5.20$ | $176.6 \pm 98.04$ | $220.6 \pm 124.55$ | $117.8 \pm 7.17$ | $\mathbf{113.6} \pm 6.83$ |
| Covertype | $232.0 \pm 31.58$ | $222.8 \pm 38.15$ | $245.6 \pm 38.69$ | $\mathbf{112.2} \pm 12.24$ | $120.8 \pm 9.41$ |
| Magic Telescope | $88.6 \pm 2.42$ | $91.4 \pm 3.44$ | $\mathbf{82.8} \pm 8.91$ | $86.4 \pm 5.46$ | $92.4 \pm 10.84$ |
| Mushroom | $1.2 \pm 1.30$ | $1.2 \pm 1.30$ | $1.2 \pm 1.30$ | $\mathbf{0.0} \pm 0.0$ | $1.8 \pm 2.49$ |
| Shuttle | $3.6 \pm 1.02$ | $3.0 \pm 0.89$ | $13.2 \pm 18.08$ | $\mathbf{2.8} \pm 0.75$ | $110.6 \pm 194.65$ |
| MNIST_784 | $109.4 \pm 163.1$ | $\mathbf{94.0} \pm 109.6$ | $124.6 \pm 168.5$ | $108.6 \pm 98.0$ | $235.6 \pm 21.9$ |

Here, $\lambda = 0$ corresponds to vanilla TS. We sweep over $\lambda$ and $\beta \in \{1, 10^3\}$ in the linear and logistic setting in Appendix E for $d = \{20, 40\}$, and discuss the results in Section 6.

**Preconditioning**. We additionally list preconditioning-incorporated algorithms in Table 1. We explicitly contrast the effect of preconditioning in Appendix E.6 and discuss the results in Section 6.

**Inverse Temperature parameter.** We experimented with two initial choices of the inverse-temperature parameter $\beta \in \{1, 10^3\}$ in Table 1. Only the $\beta = 10^3$ initialized experiments follow a $\beta_t^{-1} \propto d \log T$ schedule, as in Xu et al. (2022b). Following Xu et al. (2022b), we use this initialization for the remainder of the experiments (logistic and neural contextual bandits).

**Underdampened Langevin Monte Carlo.** We studied the impact of adding a dampening coefficient $\gamma = 0.1$ in the LMC suite of experiments, while fixing $\lambda = 0.01, \beta = 10^3$ in the feel-good regimes. Similarly, $\beta$ followed a $\beta_t^{-1} \propto d \log T$ schedule. We discuss our results in Appendix E.3.

## 6 Discussion

Our experiments reveal a significant room for improvement in applying MCMC algorithms to the contextual bandit problem. For instance, as Riquelme et al. (2018) mentions, the difficulty of online uncertainty estimation, which does not appear in supervised learning, is that the model has to be frequently updates as data is accumulated in an online fashion. Therefore, methods that converge slowly pose a natural disadvantage as they present a natural trade-off through truncating the degree of optimization in order to make the algorithm tractable. Moreover, combining methods that approximate posteriors leads to a worse posterior approximation, causing rapid performance degeneration.

We discuss our main findings for each class of algorithms.

**Neural-$\epsilon$-Greedy/NeuralUCB.** Empirically, Neural-$\epsilon$-Greedy attains the lowest simple regret on 4 tasks, while NeuralUCB is competitive, as seen in Tables 2 and 3. This is likely due to the mini-batch SGD and dropout already injecting substantial noise; adding a small, schedule-controlled $\epsilon$-greedy component yields enough exploration. On the other hand, NeuralUCB linearizes only the last layer and keeps a closed-form ridge covariance, resulting in UCB bonuses that are far less sensitive to posterior misspecification than full network-Langevin samplers (Zhou et al., 2020).

**LinUCB.** LinUCB is a strong baseline in low dimensions as it achieves the minimax-optimal regret and has good performance in linear settings. However, LinUCB requires computing a matrix inverse at each step, which can be expensive as it takes $O(d^3)$ worst-case time using Cholesky decompositions.

**PHMCTS.** Adding a preconditioner to HMCTS reduces the cumulative regret from $240 \pm 35.4$ to $90 \pm 16.2$ in the linear setting. The gain confirms the sensitivity of HMC to ill-conditioned posteriors: scaling by an estimate of $\mathbf{V}_t^{-1/2}$ reduces the posterior's anisotropy, allowing the leapfrog integrator to take larger, more isotropic steps (Girolami and Calderhead, 2011).

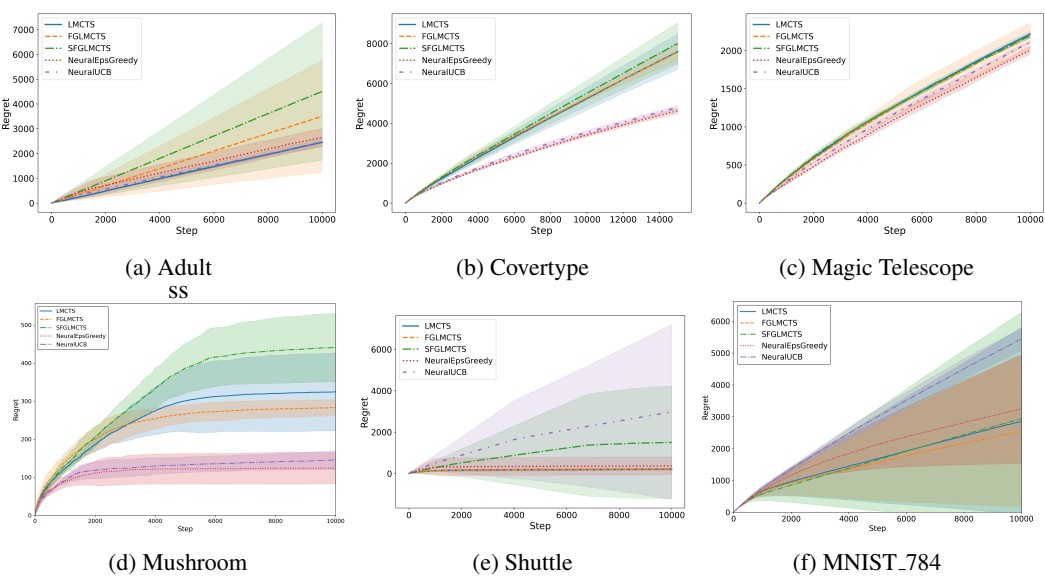

Figure 1: Mean regret comparison on simulated bandit problems. The shaded band around each mean curve represents ±1 sample standard deviation across 5 independent runs.

Table 4: Final cumulative regret for synthetic datasets. Sweeps over feel-good parameter $\lambda$ at $\beta = 10^3$.

| Algorithm | $\lambda = 0$ | $\lambda = 0.01$ | $\lambda = 0.1$ | $\lambda = 0.5$ | $\lambda = 1.0$ |
|---|---|---|---|---|---|
| FGLMCTS | $62.6 \pm 9.5$ | $62.7 \pm 12.0$ | $\mathbf{50.8} \pm 15.8$ | $235.8 \pm 185.8$ | $612.7 \pm 340.4$ |
| FGMALATS | $\mathbf{61.3} \pm 26.6$ | $62.7 \pm 25.5$ | $63.7 \pm 14.5$ | $212.8 \pm 170.0$ | $528.3 \pm 377.2$ |
| PFGLMCTS | $134.4 \pm 19.9$ | $132.7 \pm 19.1$ | $140.3 \pm 24.4$ | $193.2 \pm 47.2$ | $332.9 \pm 221.2$ |
| PSFGLMCTS | $134.4 \pm 19.9$ | $137.1 \pm 27.5$ | $130.5 \pm 13.8$ | $183.1 \pm 37.3$ | $\mathbf{286.1} \pm 134.1$ |
| SFGLMCTS | $62.6 \pm 9.5$ | $65.2 \pm 12.3$ | $82.0 \pm 32.6$ | $223.8 \pm 190.1$ | $546.0 \pm 357.6$ |
| SFGMALATS | $\mathbf{61.3} \pm 26.6$ | $\mathbf{56.2} \pm 22.8$ | $68.4 \pm 26.2$ | $\mathbf{173.4} \pm 110.1$ | $486.2 \pm 347.1$ |

**MALATS.** Our linear bandit experiments (with $d = \{20, 40\}$) show that MALATS achieve best cumulative regret performance. The gains from MALATS-based methods can be attributed to the discretization correction and faster mixing in the posterior sampler. Unadjusted Langevin dynamics uses Euler discretization of the posterior's Langevin diffusion; while it is fast, it does not converge to the true posterior as an exact stationary distribution, and incurs a bias on the order of the step size. The bias implies that unadjusted Langevin dynamics may sample from a shifted distribution if the step is not infinitesimally small. MALATS addresses this by applying a Metropolis-Hastings correction at each step to eliminate discretization error, enabling sampling from the true posterior.

**FG in Neural Contextual Bandits.** Despite its stronger theoretical guarantees in the linear setting, FG-TS degrades rapidly in neural contextual bandits, where vanilla-TS remains *more* competitive. Stochastic gradients already inject substantial "free" exploration, and the FG bonus can push the sampler further from high-density regions. Neural posteriors are very non-Gaussian: mode-connectivity results show that SGD minima are linked by low-loss paths, forming a connected manifold rather than isolated valleys (Draxler et al., 2019; Garipov et al., 2018). Such topologies defies Gaussian or mixture assumptions of well-separated modes. Moreover, the quadratic surrogates underlying FG assume symmetric curvature, while Izmailov et al. (2019) show that SGD converges near the edge of wide, anisotropic basins with asymmetric directions, making quadratic fits reliable only locally. Consequently, FG-induced optimism amplifies model mismatch and yields sub-optimal arm choices. Vanilla LMCTS thus remains a strong baseline. In our experiments, moderate exploration, whether through SGD noise or $\epsilon$-greedy perturbations, outperforms the optimism of FG-TS. From a different perspective, our observations that FG-TS underperforms in the neural setting is the empirical consequence of applying an algorithm based on quadratic assumptions to this complex, non-quadratic setting.

**Preconditioning.** Our preconditioned samplers perform Langevin updates in a whitened parameter space injecting Gaussian noise with covariance proportional to the inverse design matrix $\mathbf{V}_t^{-1}$. When $\mathbf{V}_t$ is ill-conditioned, as it typically is in high dimensions, some directions receive huge stochastic kicks while others hardly move, leading to erratic exploration and exploding variance, as visible in SVRGLMCTS's large regret. These imply a necessity to regularize or clip the spectrum of $\mathbf{V}_t^{-1}$ to keep noise injection balanced across directions, and to also include a Metropolis filter when using a mass-matrix preconditioner. Finally, our results on PSFGHMCTS indicate that it has a higher standard deviation than mean. This skew serves as an argument for incorporating additional metrics (such as the median) for studying the aggregate performance of MCMC contextual bandit algorithms.

**Stochastic Gradient Mini-batch.** Mini-batch algorithms such as SVRGLMCTS balance computational savings against the extra gradient noise they introduce: smaller batches reduce per-round cost, yet the heightened noise perturbs Langevin proposals, lowers Metropolis–Hastings acceptance rates, and inflates posterior variance. When $d = 20$, all SVRG methods remain competitive—SVRGLMCTS even matches LinUCB at $73 \pm 31$—showing that the control-variate suppresses noise for small $d$. However, at $d = 40$, the snapshot gradient used in the control-variate grows stale: the variance of the correction term scales as $O(d)$, acceptance probabilities shrink, and the mean-regret of SVRGLMCTS degenerates to $19236 \pm 15560$, leading to highly unstable actions.

**Smoothening the FGTS Objective.** Once the linear posterior has already concentrated around its true mean, injecting additional "feel-good" optimism introduces bias that the SFG objective can reduce. Our experiments show that the smoothed FG objective is helpful only in regimes where the posterior is still under-concentrated—for example, at very small horizons $T$ or under high observation noise, where the additional optimism can accelerate exploration. In well-concentrated regimes, FGMALATS with $\lambda = 0.01$ (without smoothening the FGTS bonus) becomes a more robust choice.

**Damping.** Table 31 reports LMC results with varying damping in the linear contextual bandit setting. Overdamped (vanilla) Langevin methods consistently outperform their underdamped counterparts, with the performance gap shrinking monotonically as the damping coefficient $\gamma$ increases. Although underdamped LMC enjoys stronger theoretical guarantees, its empirical performance deteriorates rapidly, which is likely due to posterior mismatch effects that are amplified at lower damping.

**Run-to-run sensitivity.** The variability in our experiments arises due to reward stochasticity, noise in mini-batch gradients within the MCMC samplers, and errors in posterior approximation with finitely many MCMC steps. Unfortunately, the contextual bandit feedback loop may amplify these errors, which is an important consideration for practitioners in the community. We reveal this sensitivity as a data point for the community to note that while MCMC methods are powerful, their application in online learning requires careful management of approximation quality for stability. Such variability in vanilla TS has already been noted (Guha and Munagala, 2014; Mazumdar et al., 2020); however, to our knowledge, our paper is the first to systematically document and analyze this for FGTS.

## 7    Conclusion

We present the first systematic evaluation of FGTS and its smoothed variants under exact and approximate MCMC-derived posterior regimes. Our ablations analyze the effects of preconditioning, feel-good bonuses, and temperature scaling. Across 14 real-world and synthetic datasets, we find that high-fidelity posteriors combined with FGTS's optimism reduce regret in linear and logistic settings, whereas neural bandits underperform compared to vanilla TS. We also identify cases where larger bonuses cause severe performance degradation under approximate posteriors.

**Limitations and Future Work.** Our experiments reflect the sensitivity of the samplers to learning-rate schedules, mini-batch noise, and other hyperparameters. While we used a modest grid search for tuning, our study was not exhaustive. Recent work (Luo and Bayati, 2025) shows that geometry-aware variants of TS can achieve minimax-optimal regret in linear bandits. Benchmarking such algorithms, along withs noise-adaptive TS variants (Xu et al., 2023) and MCMC-based methods on generalized linear reward models (Li et al., 2017), would be a promising direction for future work.

**Societal Impacts.** This work is empirical in nature. While it enables better understanding of the behavior of bandit algorithms in numerous synthetic and real-world benchmarks, it is not tied to any specific applications or deployments.

# 8 Acknowledgements

This work was supported by NSF Grant CCF 2338816. We express our thanks to Eric Mazumdar, Jan van den Brand, Jake Abernethy, Haque Ishfaq, Yunbum Kook, Mirabel Reid, and Kabeer Thockchom for insightful discussions.

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

**Outline of the Appendices.**

- Appendix A has mathematical background and additional remarks.
- Appendix B provides a more comprehensive review of existing Markov Chain Monte Carlo (MCMC) methods.
- Appendix C gives theoretical guarantees for preconditioned Langevin Monte Carlo for Thompson Sampling (LMCTS), Hamiltonian Monte Carlo for Thompson Sampling (HM-CTS), and their feel-good (FG) variants.
- Appendix D has descriptions of algorithms and hyperparameter settings.
- Appendix E has further experimental details.
- Appendix F has details on the real-world tasks (five UCI datasets and MNIST_784).

## A    Mathematical Background and Additional Remarks

**Notation.** Let $\mathbb{Z}_+$ denote the set of strictly positive integers, and $\mathbb{R}^d$ denote the set of $d$-dimensional reals. We use $[k]$ to denote a set $\{1, \ldots, k\}, k \in \mathbb{N}^+$. Let $\|\mathbf{x}\|_2 = \sqrt{\mathbf{x}^\top \mathbf{x}}$ be the Euclidean norm of a vector $\mathbf{x} \in \mathbb{R}^d$. For a matrix $\mathbf{V} \in \mathbb{R}^{m \times n}$, we denote by $\|\mathbf{V}\|_2$ and $\|\mathbf{V}\|_F$ its operator and Frobenius norm respectively. For a positive semi-definite (PSD) matrix $\mathbf{V} \in \mathbb{R}^{d \times d}$ and a vector $\mathbf{x} \in \mathbb{R}^d$, denote the Mahalanobis (local) norm as $\|\mathbf{x}\|_{\mathbf{V}} = \sqrt{\mathbf{x}^\top \mathbf{V} \mathbf{x}}$. For a function $f(T)$, we use the common big-O notation $O(f(T))$ to hide constant factors with respect to $T$, and use $\widetilde{O}(f(T))$ to omit the logarithmic dependence on $T$. Finally, we summarize the notation specific to contextual bandits in Table 5.

Table 5: Important notations in this paper.

| Notation | Meaning |
|---|---|
| $T$ | $T \in \mathbb{N}$ is the number of rounds in the game; |
| $\mathcal{X}_t$ | $\mathcal{X}_t \subseteq \mathbb{R}^d$ is the action set (or context) of the agent; |
| $x_t$ | $x_t \in \mathcal{X}_t$ is the feature vector that represents an arm or action; |
| $\theta^*$ | $\theta^* \in \mathbb{R}^m$ is an unknown weight parameter shared across all arms; |
| $r_t$ | $r_t := r_t(x)$ is the feedback reward generated and returned to the algorithm; |
| $f_\theta(x)$ | $f_\theta(x)$ is the reward generating function: the observed reward is $r_t(x) = f_{\theta^*}(x) + \xi_t$; |
| $\mathcal{L}^{\text{TS}}$ | $\mathcal{L}^{\text{TS}}$ is the Thompson Sampling loss-likelihood given by $\mathcal{L}_t(\theta) = \sum_{s=1}^{t-1} \eta(f_\theta(x_s) - r_s)^2$; |
| $\mathcal{L}^{\text{FG}}$ | $\mathcal{L}^{\text{FG}}$ is the Feel-Good Thompson Sampling loss likelihood given in Equation (1); |
| $\mathcal{L}^{\text{SFG}}$ | $\mathcal{L}^{\text{SFG}}$ is the smoothed Feel-Good Thompson Sampling loss likelihood given in Equation (2) |
| $\lambda$ | $\lambda > 0$ is the Feel-Good parameter in FGTS; |

## B    Relevant Literature

The field of contextual bandits has a rich and extensive history, encompassing various theoretical and algorithmic advancements. Providing an exhaustive review of the entire domain is impractical here; thus, we refer interested readers to comprehensive surveys such as Bubeck et al. (2009); Lattimore and Szepesvári (2020). Below we focus on work most relevant to Thompson sampling, Markov chain Monte Carlo (MCMC) methods, and associated variants.

**Regret bounds in Linear Contextual Bandits.** Linear contextual bandits have been extensively studied, particularly in relation to regret bounds achieved by policies based on the Upper Confidence Bound (UCB) principle, where frequentist regret bounds of $\widetilde{O}(d\sqrt{T})$ are attainable (Zhou et al., 2020). Thompson Sampling (TS), which involves sampling from the posterior distribution of the reward function and selecting actions accordingly, demonstrates strong empirical performance and applies naturally to both Bayesian and frequentist frameworks. Russo and Roy (2016) showed that the best-known Bayesian regret bound for TS matches the minimax optimal $\widetilde{O}(d\sqrt{T})$. However, the best frequentist regret bound proven to date for TS is $O(d\sqrt{dT})$ (Agrawal and Goyal, 2013) due primarily to inflated posterior variance of order $\widetilde{O}(d)$ Significant contributions to this area leverage concentration inequalities for self-normalized martingales, as seen in Abbasi-Yadkori et al. (2011);

Agarwal et al. (2014, 2021); Agrawal and Goyal (2013); Bercu and Touati (2019); Bietti et al. (2021). There is also a body of work that shows that in cases where structural assumptions are made on the linear contextual bandit problem, there are algorithms that achieve the optimal regret bound; for instance, Chu et al. (2011) studies the case with linear payoff functions and Reid et al. (2025) studies the case with budget constraints. Both works derive the optimal $O(d\sqrt{T})$ regret bounds.

**Preconditioning Methods.** Recent works in iterative optimization consider preconditioning as a technique to enhance algorithmic performance in contextual bandits and reinforcement learning. In such methods, the Richardson iteration of $x_{t+1} = x_t - \eta\nabla L(x_t)$ is replaced by $x_{t+1} = x_t - \eta P^{-1}\nabla L(x_t)$, where $P$ is a preconditioning matrix which satisfies the property that $P^{-1}\nabla L(\cdot)$ has condition number smaller than $\nabla L(\cdot)$, which in turn accelerates convergence. Terekhov (2025) investigates contextual bandits utilizing pretrained neural networks, leveraging preconditioning to improve exploration and convergence. Similarly, Millard et al. (2025) explores matrix preconditioning strategies within actor-critic methods in reinforcement learning, demonstrating improved stability and performance. Techniques from Gaussian Process UCB (Srinivas et al., 2010) and deep conditioning on overparameterized neural networks (Agarwal et al., 2021) have also been influential. Staib et al. (2019) provides critical insights into escaping saddle points using adaptive gradient methods such as RMSprop, which have implications for optimization in preconditioned gradient descent. Further advancements include preconditioned stochastic gradient descent (Li, 2018), preconditioned stochastic gradient Langevin dynamics (SGLD) (Li et al., 2016), and contextual bandit learning with predictable rewards (Agarwal et al., 2012).

**Beyond Standard Linear Settings.** Expanding beyond linear setting, substantial progress has been made in generalized linear bandits (Filippi et al., 2010), where the true reward $r$ from arm $x \in \mathcal{X}_t$ at round $t$ is assumed to be from a generalized linear model (GLM). In a GLM, conditional on feature vector $x$, $r$ follows an exponential family distribution with mean $\mu(x^\top\theta^*)$ where $\theta^*$ is an unknown weight parameter shared across all arms and $\mu(\cdot)$ is called the link function. Sivakumar et al. (2020) provides a geometric perspective on contextual bandits, introducing smoothness through Gaussian perturbations in adversarial contexts. Further, adaptive noise strategies for Thompson sampling have been analyzed rigorously, resulting in refined regret bounds (Ishfaq et al., 2024a,b; Xu et al., 2023).

**Stochastic Sampling Methods.** Stochastic sampling algorithms have significantly advanced the computational efficiency and effectiveness of contextual bandit strategies. Ahn et al. (2012) integrated Fisher information matrices to create efficient optimizers during burn-in phases. Patterson and Teh (2013) developed stochastic gradient Riemannian Langevin dynamics for exploring distributions over the probability simplex, particularly in the context of Latent Dirichlet Allocation where one seeks to uncover hidden thematic structures in large corpora. Similarly, stochastic gradient Hamiltonian monte carlo (SGHMC) methods (Anand and Qu, 2024; Chen et al., 2014) have become popular tools for managing uncertainty and improving sampling efficiency in complex spaces.

# C  Algorithm Details

This section provides details on the MCMC algorithms that we evaluate for feel-good Thompson sampling as well as Thompson sampling in various contextual bandit tasks. We present sample complexities, give convergence guarantees of the MCMC sampling algorithms, and prove tight regret bounds of each variant that we consider.

## C.1  Algorithm Implementation Details

**Cholesky Factorization and Sherman-Woodbury-Morrison.** In a number of the algorithms we consider, it is often imperative to dynamically maintain the inverse matrix $\mathbf{V}_t^{-1/2}$. This can be done in $O(d^3)$ by using Sherman-Woodbury-Morrison to dynamically maintain a Cholesky factorization, and has been shown to be numerically stable in finite bit precision (Anand et al., 2024, 2025a).

**Preconditioned FGLMCTS.** The preconditioned SFGLMCTS update is given by $\theta_{t,k+1} = \theta_{t,k} - \eta\mathbf{V}_t^{-1/2}\nabla\mathcal{L}_t(\theta_{t,k-1}) + \sqrt{2\eta_t\beta_t^{-1}}\mathbf{V}_t^{-1/2}\epsilon_{t,k}$, where $\mathcal{L}_t$ is the smoothed feel-good likelihood, and $\epsilon_{t,k} \sim \mathcal{N}(\mathbf{0}, \mathbf{I})$ is an isotropic Gaussian vector. Here, we have preconditioned the discretized Langevin update using the matrix $\mathbf{V}_t$.

Table 6: Complexity and Regret Analysis for Linear contextual bandits (LCB)

| Algorithm | Number of LCB Iterations | LCB Regret |
|---|---|---|
| UCB | $\widetilde{O}(T \log T \sqrt{dT})$ | $\widetilde{O}(d\sqrt{T})$. |
| LMCTS | $\widetilde{O}(\kappa T \log \sqrt{dT \log T^3}/\epsilon)$ | $\widetilde{O}(d^{3/2}\sqrt{T})$ |
| Under/Over damped LMC-TS | $O(\kappa T d/\epsilon^2)$ | $\widetilde{O}(d^{3/2}\sqrt{T})$ |
| FG-TS | $\widetilde{O}(\kappa d T^4)$ | $O(d\sqrt{T})$. |
| FGLMCTS | $\widetilde{O}(dT^4)$ | $O(d\sqrt{T})$ |
| FGMALATS | $\widetilde{O}(dT^2)$ | $O(d\sqrt{T})$ |
| Preconditioned SFGLMCTS | $\widetilde{O}(T \log \sqrt{dT})$ | $O(d\sqrt{T})$ |
| SFGHMCTS | $\widetilde{O}(\sqrt{k}d^{1/4}T)$ | $O(d\sqrt{T})$ |
| Preconditioned FGHMCTS | $\widetilde{O}(d^{1/4}T)$ | $O(d\sqrt{T})$ |

**Stochastic gradient update.** This can be extended to a stochastic gradient algorithm (SG-PFGLMCTS) by letting $L_t(\theta) = \sum_{s=1}^{t-1} \ell(r_s, f(x_s, \theta)) + \lambda R\|\theta\|^2$, and approximating the gradient by $\nabla \widehat{L}_t(\theta) = \sum_{s \in B_t} \nabla_\theta \ell(r_s, f(x_s, \theta)) + \lambda \nabla_\theta R(\theta)$. Further, the variance can be reduced by a stochastic variance-reduced gradient algorithm (SVRGLMCTS) which we consider in our experiments.

**Preconditioned Stochastic variance reduced Gradient LMC Thompson Sampling.** Suppose there are $n$ component functions $\{f_i(\cdot)\}_{i=1}^n$ and the objective is $F(\theta) = \frac{1}{n} \sum_{i=1}^n f_i(\theta)$. Let $\theta_{\text{ref}} \in \mathbb{R}^d$ be a "snapshot" parameter vector, which is updated once every *outer loop*. At each iteration $k$ of an *inner loop*, we form a mini-batch $I_k \subset \{1, \ldots, n\}$ such that $|I_k| = B$ and compute the SVRG gradient estimate as follows:

$$\nabla \widehat{F}_k(\theta) = \underbrace{\frac{1}{B} \sum_{i \in I_k} [\nabla f_i(\theta) - \nabla f_i(\theta_{\text{ref}})]}_{\text{variance-reduction correction}} + \underbrace{\nabla F(\theta_{\text{ref}})}_{\text{snapshot full-gradient}} .$$

This estimate $\nabla \widehat{F}_k(\theta)$ unbiasedly approximates $\nabla F(\theta)$ but with reduced variance compared to vanilla stochastic-gradient estimates, especially once $\theta$ is close to $\theta_{\text{ref}}$.

### C.2  Proof of Convergence of Preconditioned SFGLMCTS

The preconditioned SFGLMCTS update is given by

$$\theta_{t,k+1} = \theta_{t,k} - \eta \mathbf{V}_t^{-1/2} \nabla \mathcal{L}_t(\theta_{t,k-1}) + \sqrt{2\eta_t \beta_t^{-1}} \mathbf{V}_t^{-1/2} \epsilon_{t,k}, \tag{5}$$

where $\mathcal{L}_t$ is the smoothed feel-good likelihood, and $\epsilon_{t,k} \sim \mathcal{N}(\mathbf{0}, \mathbf{I})$ is an isotropic Gaussian vector. Note that this is identical to the preconditioned LMCTS update by setting the feel-good parameter $\lambda$ in $\mathcal{L}_t$ to 0. The $\theta$-update in Equation (5) can be viewed as the Euler-Maruyama discretization of the preconditioned Langevin dynamics from physics given by the stochastic differential equation

$$d\theta(s) = -\mathbf{V}_t^{-1} \nabla L_t(\theta(s)) ds + \sqrt{2\beta_t^{-1} \mathbf{V}_t^{-1}} dB(s), \tag{6}$$

where $s > 0$ is a time index, $\beta > 0$ is the inverse temperature parameter, and $B(t) \in \mathbb{R}^d$ is the Brownian motion. The SDE in Equation (6) is a special instance of the Riemannian Langevin Dynamics presented in Girolami and Calderhead (2011); Patterson and Teh (2013) which coincides with Langevin Dynamics on a flat manifold where $V_t$ is the metric tensor matrix. Under suitable conditions on $\mathbf{V}$ and $L$, this dynamics converges to a unique stationary point given by $\pi(dx) \propto e^{-\beta L(x)} dx$ (Lemma C.1). So, the discretization approximates sampling from an arbitrary distribution $\pi_t \propto \exp(-\beta_t L_t(\theta))$. Lemma C.1 shows that LMC converges to the Feel-Good Thompson Sampling likelihood of $\mathcal{L}_t(\theta)$. Using standard techniques, the proof of Lemma C.1 further extends to variants of LMC with stochastic gradients (Jose and Moothedath, 2024; Patterson and Teh, 2013; Welling and Teh, 2011), dampeners (Zheng et al., 2024), and preconditioners (Bhattacharya and Jiang, 2023; Li et al., 2016; Titsias, 2023).

**Lemma C.1** (Convergence of Preconditioned Langevin Dynamics). *If $\mathbf{V}_t \in \mathbb{R}^{d \times d}$ is an invertible symmetric matrix and $\mathcal{L} : \mathbb{R}^d \to \mathbb{R}$ is a smooth function, the preconditioned FGLMCTS dynamics corresponding to the stochastic differential equation given by*

$$d\theta(s) = -\mathbf{V}_t^{-1}\nabla\mathcal{L}_t(\theta(s))\mathrm{d}s + \sqrt{2(\mathbf{V}_t\beta_t)^{-1}}\mathrm{d}\mathbf{B}(s)$$

*converges to a unique stationary distribution $\pi(\mathrm{d}\theta) \propto e^{-\beta_t\mathcal{L}_t(\theta)}\mathrm{d}\theta$.*

*Proof.* Let $\rho_s$ denote the distribution of $\theta(s)$. By the Fokker-Planck (forward Kolmogorov) equation, the distribution of $\rho_s$ is given by

$$\frac{\mathrm{d}\rho_s}{ds} = -\sum_i \frac{\partial}{\partial\theta_i}(\rho_s\mu(\theta(s))_i) + \frac{1}{2}\sum_{i,j}\frac{\partial^2}{\partial\theta_i\partial\theta_j}(\rho_s(\sigma\sigma^\top)_{i,j}),$$

where $\mu(\theta(s))_i := -(\mathbf{V}_t^{-1}\nabla\mathcal{L}_t(\theta(s)))_i$ and $\sigma := \sqrt{2(\mathbf{V}_t\beta_t)^{-1}}$. As $\mathbf{V}_t$ is symmetric, $\sigma\sigma^\top = 2\beta_t^{-1}\mathbf{V}_t^{-1}$. The stationary distribution $\pi$ is obtained by setting $\mathrm{d}\rho/\mathrm{d}s = 0$. Then, $\pi$ satisfies

$$\frac{d\rho_s}{ds} = 0 = \underbrace{\sum_i \frac{\partial}{\partial\theta_i}\big(\pi(\theta)[\mathbf{V}_t^{-1}\nabla\mathcal{L}_t(\theta(s))]_i\big)}_{\text{(I)}} + \underbrace{\sum_{i,j}\frac{\partial^2}{\partial\theta_i\partial\theta_j}(\pi(\theta)\beta_t^{-1}(\mathbf{V}_t^{-1})_{i,j})}_{\text{(II)}}$$

When $\pi(\theta) \propto e^{-\beta_t\mathcal{L}_t(\theta)}$, $\frac{\partial}{\partial\theta_j}\pi(\theta) = -\pi(\theta) \cdot \beta_t\nabla\mathcal{L}_t(\theta_j)$ for any $j \in [d]$. Then, we write

$$\text{(I)} = \sum_i \frac{\partial}{\partial\theta_i}\big(\pi(\theta)[\mathbf{V}_t^{-1}\nabla\mathcal{L}_t(\theta(s))]_i\big) = \sum_i \frac{\partial}{\partial\theta_i}\pi(\theta)\sum_j(\mathbf{V}_t^{-1})_{i,j}\nabla\mathcal{L}_t(\theta_j)$$

$$\text{(II)} = \beta_t^{-1}\sum_i \frac{\partial}{\partial\theta_i}\left(\sum_j(\mathbf{V}_t^{-1})_{i,j}\frac{\partial}{\partial\theta_j}\pi(\theta)\right) = -\sum_i \frac{\partial}{\partial\theta_i}\pi(\theta)\sum_j(\mathbf{V}_t^{-1})_{i,j}\nabla\mathcal{L}_t(\theta_j)$$

As (I) $= -$ (II), the density proportional to $e^{-\beta_t\mathcal{L}_t(\theta)}$ is the stationary distribution for the SDE. Since $L_t$ and $\beta_t$ are measurable, Lemma 5.2.1 in Øksendal (2003) (or Theorem B.3.1 in Bakry et al. (2013)) gives that the SDE admits a unique stationary distribution, which proves the lemma. $\qquad\square$

## C.3 Regret Analysis of PSFGLMCTS for Linear Contextual Bandits

We provide the regret analysis of PSFGLMCTS for Linear Contextual Bandits, which follows from a reduction to Huix et al. (2023). We first state the assumption on the details of the model.

**Assumption C.2.** There is an unknown parameter $\theta^* \in \mathbb{R}^d$ such that for any arm $x \in \mathcal{X} \subseteq \mathbb{R}^d$, the reward is $r(x) = x^\top\theta^* + \zeta$, where $\zeta$ is an $R$-sub-Gaussian random variable for some constant $R > 0$. Note that this assumption is automatically satisfied if the rewards are bounded almost surely.

We first state the $\widetilde{O}(d\sqrt{T})$-regret bound for linear contextual bandits from the Langevin Monte Carlo smoothed Feel-Good Thompson Sampling (SFGLMCTS) methodology in Huix et al. (2023).

**Theorem C.3** (Theorem 5 in Huix et al. (2023)). *Let the prior distribution $p_0$ for FGLMCTS satisfy a $m_0$-strongly log concave property for some $m_0 \geq 0$. The regret for FGLMCTS when run for $T$ time steps (under Assumption C.2) in linear contextual bandits is bounded by*

$$R(T) \leq C\left(\frac{d}{1+m_0} + \sqrt{R} + m_0\right)\sqrt{T\log^3(dT)} \leq \widetilde{O}(d\sqrt{T}).$$

**Theorem C.4.** *Under Assumption C.2, the PSFGLMCTS procedure achieves $\widetilde{O}(d\sqrt{T})$ regret for linear contextual bandits when run for $T$ time steps.*

*Proof.* The proof follows by a direct reduction from the FGLMCTS setting in Huix et al. (2023). From Lemma C.6, each inner iteration of the MCMC algorithm converges to $\pi(\theta) \propto e^{-\mathcal{L}_t(\theta)}$, where $\mathcal{L}_t(\theta)$ is the same smoothed posterior (SFG-TS) that each inner loop in FGLMCTS also converges to. Then, since FGLMCTS achieves $\widetilde{O}(d\sqrt{T})$ regret via Theorem C.3, the regret for PSFGLMCTS is also $\widetilde{O}(d\sqrt{T})$, proving the theorem. $\qquad\square$

**Remark C.4.1.** *This regret bound of $\widetilde{O}(d\sqrt{T})$ also extends to variants of SFGLMCTS with stochastic gradients (Zou et al., 2018), dampeners (Patterson and Teh, 2013), and preconditioners (Li et al., 2016), since these factors each only affect the convergence rate in each inner loop of the algorithm that approximates the posterior.*

## C.4 Analysis of Hamiltonian Monte Carlo Feel-Good Thompson Sampling

Recall that the Hamiltonian $H : \mathbb{R}^d \times \mathbb{R}^d \to \mathbb{R}$ captures the sum of the potential and kinetic energies of a particle as a function of its position $\theta \in \mathbb{R}^d$ and velocity $v \in \mathbb{R}^d$. We let $H(\theta, v) = \mathcal{L}_t(\theta) + \frac{1}{2}\|v\|^2$. Then, the parameters $(\theta_t, v_t)$ follow the *Hamiltonian curve* trajectory given by

$$\frac{\mathrm{d}\theta}{\mathrm{d}t} = v, \qquad \frac{\mathrm{d}v}{\mathrm{d}t} = -\nabla\mathcal{L}_t(\theta). \tag{7}$$

---

**Algorithm 3 H**amiltonian **M**onte **C**arlo **F**eel-**G**ood **T**hompson **S**ampling (HMC-FG-TS)

---

**Input:** step-size $\varepsilon$, leapfrog steps $L$, inner iters $K, K'$, FG-weight $\lambda$, cap $b$, temperature $\eta$, prior-precision $\tau = \sigma_{\text{prior}}^{-2}$, arms $[A]$, feature maps $\phi, \phi_a$
**Define:** $U_{\text{FG}}(\theta) = \eta\sum_{s=1}^{t-1}(\langle x_s, \theta\rangle - r_s)^2 + \sigma_{\text{prior}}^{-2}\|\theta\|^2 - \lambda\sum_{s=1}^{t-1}\min(b, \max_{a \leq A}\langle V_{s,a}, \theta\rangle)$
*// Setting $\lambda = 0$ above recovers vanilla Thompson sampling*
**for** $t = 1, \ldots, T$ **do**
    Observe context $x_t$, set $\mathbf{V}_t = \phi(x_t) \in \mathbb{R}^{A\times d}$.
    $K_t \leftarrow K$ if posterior updated else $K'$.
    **for** $k = 1, \ldots, K_t$ **do**
        Sample $p \sim \mathcal{N}(0, \mathbf{I}_d)$.
        $(\theta', p') \leftarrow \text{Leapfrog}_\varepsilon(\theta, p)$.
        $\Delta H = \left[U_{\text{FG}}(\theta) + \frac{1}{2}\|p\|^2\right] - \left[U_{\text{FG}}(\theta') + \frac{1}{2}\|p'\|^2\right]$.
        Set $\theta \leftarrow \theta'$ with probability $\min(1, e^{-\Delta H})$.
    Choose $a_t = \arg\max_a V_{t,a}^\top\theta$, observe $r_t$ and update accordingly.

---

Under certain conditions on $-\nabla\mathcal{L}_t(\theta)$, the distribution of the Markov chain in the inner loop of Algorithm 3 converges to a unique stationary distribution $\pi(\mathrm{d}\theta) \propto e^{-\mathcal{L}_t(\theta)}\mathrm{d}\theta$ (see Lemma C.6). We provide details on sampling $p'$ using numerical leapfrog integrators in Algorithm 4, and show in Theorem C.4 that HMC-FG-TS in linear contextual bandits satisfies a regret of $\widetilde{O}(d\sqrt{T})$.

---

**Algorithm 4** Leapfrog Integration to sample $v \propto \exp(-H(x^{(k)}, v))$

---

**Input:** current position $\theta \in \mathbb{R}^d$, step size $\varepsilon > 0$, number of drift–kick steps $L \in \mathbb{N}$, differentiable potential energy $U(\cdot) = \mathcal{L}_t(\cdot) + \sigma_{\text{prior}}^{-2}\|\cdot\|^2 - \lambda\sum_{s<t}\min(b, g_s^\star(\cdot))$.
Draw fresh momentum $p \sim \mathcal{N}(\mathbf{0}, I_d)$
$p \leftarrow p - \frac{\varepsilon}{2}\nabla U(\theta)$
**for** $\ell = 1$ **to** $L$ **do**
    $\theta \leftarrow \theta + \varepsilon\, p$
    $g \leftarrow \nabla U(\theta)$
    **if** $\ell < L$ **then**
        $p \leftarrow p - \varepsilon\, g$
$p \leftarrow p - \frac{\varepsilon}{2}\, g$
**Output:** candidate state $(\theta, p)$

---

## C.5 Proof of Convergence of Smoothed Feel-Good HMCTS

Lemma C.6 shows that HMC converges to the Feel-Good Thompson Sampling likelihood of $\mathcal{L}_t(\theta)$. Using standard techniques, the proof of Lemma C.6 further extends to variants of HMC with stochastic gradients (Chen et al., 2014; Zou et al., 2018), dampeners (Patterson and Teh, 2013), and preconditioners, (Pidstrigach, 2022).We first introduce a crucial theorem of Sard (1942).

**Theorem C.5** (Morse-Sard Theorem (Sard, 1942))**.** *If $g : N \to M$ is a $C^k$-function (i.e., $k$ times continuously differentiable) for $k \geq \max\{n - m + 1, 1\}$ and if $A_r \subseteq N$ is the set of points $x \in N$ such that $dg_x$ has rank $< r$, then the $r$-dimensional Hausdorff measure of $g(A_r)$ is zero.*

Lemma C.6 next proves the convergence of SFGHMCTS, and the proofs follows from a line of arguments in the HMC literature (Apers et al., 2024; Mangoubi and Smith, 2021; Mangoubi and Vishnoi, 2018; Pidstrigach, 2022; Zou and Gu, 2021).

**Lemma C.6.** *The Hamiltonian Monte Carlo smoothed Feel-Good Thompson Sampling (SFGHMCTS) dynamics in Equation* (2) *converges to a stationary distribution given by $\pi(\mathrm{d}\theta) \propto e^{-\mathcal{L}_t(\theta)} \mathrm{d}\theta$, where $\mathcal{L}_t$ is the smoothed feel-good Thompson sampling posterior in Equation* (2).

*Proof.* We show a stronger statement of time-reversibility of the underlying Markov chain. Let $p_\theta(\theta')$ be the probability density of one step of SFGHMCTS starting at $\theta$, and let $\pi(\theta) = \int_{v \in \mathbb{R}^n} e^{-H(\theta, v)} \mathrm{d}v \propto e^{-\mathcal{L}_t(\theta)}$. We show that the reversibility condition

$$\pi(\theta) p_\theta(\theta') = \pi(\theta') p_{\theta'}(\theta)$$

holds almost everywhere in $\theta$ and $\theta'$.

Fix $\theta$ and $\theta'$. Let $F_\delta^\theta(v)$ be the $\theta$'th component of the map $T_\delta(\theta, v)$. Following the split subsets argument by Lee and Vempala (2025), let

$$V_+ = \{v : F_\delta^\theta(v) = \theta'\}$$
$$V_- = \{v : F_{-\delta}^\theta(v) = \theta'\}.$$

By Theorem C.5, for any $N := \{y : DF_s^\theta(v) \text{ is not invertible}\}$, $F_\delta^\theta(N)$ has measure zero. So, $DF_\delta^\theta$ is invertible everywhere except for a measure-zero subset. Therefore,

$$\pi(\theta) p_\theta(\theta') = \frac{1}{2} \int_{v \in V_+} \frac{e^{-H(\theta, v)}}{|\det(DF_\delta^\theta(v))|} + \frac{1}{2} \int_{y \in V_-} \frac{e^{-H(\theta, v)}}{|\det(DF_{-\delta}^\theta(v))|}$$

By applying time-reversal lemma (Livingstone et al., 2019) for the Hamiltonian curve, we have that for $V_+$ and $V_-$,

$$\pi(\theta') p_{\theta'}(\theta) = \frac{1}{2} \int_{y \in V_+} \frac{e^{-H(\theta', v')}}{|\det(DF_{-\delta}^{\theta'}(v'))|} + \frac{1}{2} \int_{y \in V_-} \frac{e^{-H(\theta', v')}}{|\det(DF_\delta^{\theta'}(v'))|}, \tag{8}$$

where $v'$ denotes the $v$'th component of $T_\delta(\theta, v)$ and $T_{-\delta}(\theta, v)$ in the first and second integrals.

Following the proof of Lee and Vempala (2025), let $DT_\delta(\theta, v) = \begin{pmatrix} \mathbf{A} & \mathbf{B} \\ \mathbf{C} & \mathbf{D} \end{pmatrix}$. Since $T_\delta \circ T_{-\delta} = \mathbf{I}$ and $T_\delta(\theta, v) = (\theta', v')$, the inverse function theorem gives that $DT_{-\delta}(\theta', v') = (DT_\delta(\theta, v))^{-1}$.

By the Schur complement formula, the upper-right block of $DT_{-\delta}(\theta', v')$ is $F_{-\delta}^{\theta'}(v') = -\mathbf{A}^{-1}\mathbf{B}(\mathbf{D} - \mathbf{C}\mathbf{A}^{-1}\mathbf{B})^{-1}$ and $F_\delta^\theta(v) = \mathbf{B}$. Since $\det DF_\delta^\theta(v) = 1$ by Lemma C.8, we have

$$|\det(DF_{-\delta}^{\theta'})(v')| = |\det \mathbf{A}^{-1} \det \mathbf{B} \det(\mathbf{D} - \mathbf{C}\mathbf{A}^{-1}\mathbf{B})^{-1}|$$
$$= \frac{|\det \mathbf{B}|}{|\det DT_\delta(\theta, v)|}$$
$$= |\det DF_\delta^\theta(v)|$$

Then, since $e^{-H(\theta, v)} = e^{-H(\theta', v')}$ by Lemma C.7

$$\frac{1}{2} \int_{v \in V_+} \frac{e^{-H(\theta, v)}}{|\det(DF_\delta^\theta(v))|} = \frac{1}{2} \int_{v \in V_+} \frac{e^{-H(\theta, v)}}{|\det(DF_{-\delta}^{\theta'}(v'))|}$$
$$= \frac{1}{2} \int_{v \in V_+} \frac{e^{-H(\theta', v')}}{|\det(DF_{-\delta}^{\theta'}(v'))|}$$

Repeating this calculation for the second term in Equation (8),

$$\frac{1}{2} \int_{v \in V_-} \frac{e^{-H(\theta, v)}}{|\det(DF_{-\delta}^\theta(v))|} = \frac{1}{2} \int_{y \in V_+} \frac{e^{-H(\theta', v')}}{|\det(DF_\delta^{\theta'}(v'))|},$$

which proves the reversibility condition.

Since the kernel of the HMC Markov chain is ergodic, we have that there exists a unique stationary distribution (Anand and Umans, 2023). The reversibility of HMC implies that $\exp(-H(\theta, v))$ is the (unique) stationary distribution of the joint Markov chain on $(\theta, v)$. Therefore,

$$\pi(\theta) = \int \pi(\theta, v)\mathrm{d}v \propto \int \exp(-H(\theta, v))\mathrm{d}v,$$

which proves the lemma. $\qquad \square$

**Lemma C.7.** *Any Hamiltonian curve* $(\theta(t), v(t))$ *satisfies* $\frac{\mathrm{d}}{\mathrm{d}t}H(\theta(t), v(t)) = 0$.

*Proof.* By applying the high-dimensional chain-rule on the Hamiltonian curve, we have that

$$\begin{aligned}
\frac{\mathrm{d}}{\mathrm{d}t}H(\theta(t), v(t)) &= \frac{\partial H}{\partial \theta}\frac{\mathrm{d}\theta}{\mathrm{d}t} + \frac{\partial H}{\partial v}\frac{\mathrm{d}v}{\mathrm{d}t} \\
&= \frac{\partial H}{\partial \theta}\frac{\partial H}{\partial v} - \frac{\partial H}{\partial v}\frac{\partial H}{\partial \theta} \\
&= 0,
\end{aligned}$$

which proves the lemma. $\qquad \square$

The following lemma (and proof) appears in Lee and Vempala (2025). We restate these below for completeness of the proof of Lemma C.6.

**Lemma C.8** (Measure Preservation of Hamiltonian curves). *For any* $t \geq 0$, *let* $DT_t(\theta, v)$ *be the Jacobian of the map* $T_t$ *at the point* $(\theta, v)$. *Then,*

$$\det(DT_t(\theta, v)) = 1$$

*Proof.* Let $(\theta(t, s), v(t, s))$ be a family of Hamiltonian curves given by $T_t(\theta + sd_\theta, v + sd_v)$.

Let $u(t) = \frac{\partial}{\partial s}\theta(t, s)|_{s=0}$ and $v(t) = \frac{\partial}{\partial s}v(t, s)|_{s=0}$. Differentiating the Hamiltonian equation in Equation (7), we have

$$\frac{\mathrm{d}u}{\mathrm{d}t} = \frac{\partial^2 H(\theta, v)}{\partial v\partial \theta}u + \frac{\partial^2 H(\theta, v)}{\partial v\partial v}v,$$

$$\frac{\mathrm{d}v}{\mathrm{d}t} = -\frac{\partial^2 H(\theta, v)}{\partial\theta\partial v}u - \frac{\partial^2 H(\theta, v)}{\partial\theta\partial v}v, \text{ and}$$

$$(u(0), v(0)) = (d_\theta, d_v).$$

Since $DT_t(\theta, v)(d_\theta \quad d_v)^\top = (u(t) \quad v(t))^\top = \mathbf{A}(t)(d_\theta \quad d_v)^\top$, the dynamics $\frac{\mathrm{d}u}{\mathrm{d}t}$ and $\frac{\mathrm{d}v}{\mathrm{d}t}$ are equivalent to the first-order matrix differential equation given by

$$\frac{\mathrm{d}\mathbf{A}}{\mathrm{d}t} = \begin{pmatrix} \frac{\partial^2 H(\theta(t), v(t))}{\partial v\partial \theta} & \frac{\partial^2 H(\theta(t), v(t))}{\partial v\partial v} \\ -\frac{\partial^2 H(\theta(t), v(t))}{\partial\theta\partial\theta} & -\frac{\partial^2 H(\theta(t), v(t))}{\partial\theta\partial v} \end{pmatrix}\mathbf{A}(t)$$

$$\mathbf{A}(0) = \mathbf{I}$$

Hence, $DT_t(\theta, v) = \mathbf{A}(t)$. Next, we have that

$$\begin{aligned}
\frac{\mathrm{d}}{\mathrm{d}t}\log\det\mathbf{A}(t) &= \mathrm{tr}\left(\mathbf{A}(t)^{-1}\frac{\mathrm{d}}{\mathrm{d}t}\mathbf{A}(t)\right) \\
&= \mathrm{tr}\begin{pmatrix} \frac{\partial^2 H(\theta(t), v(t))}{\partial v\partial \theta} & \frac{\partial^2 H(\theta(t), v(t))}{\partial v\partial v} \\ -\frac{\partial^2 H(\theta(t), v(t))}{\partial\theta\partial\theta} & -\frac{\partial^2 H(\theta(t), v(t))}{\partial\theta\partial v} \end{pmatrix} \\
&= 0
\end{aligned}$$

Hence, $\det\mathbf{A}(t) = \det\mathbf{A}(0) = 1$, which proves the lemma. $\qquad \square$

## C.6 Regret Analysis of SFGHMCTS for Linear Contextual Bandits

We provide the regret analysis of SFGHMCTS for Linear Contextual Bandits, which follows from a reduction to Huix et al. (2023). We first state the assumption on the details of the model.

**Assumption C.9.** There is an unknown parameter $\theta^* \in \mathbb{R}^d$ such that for any arm $x \in \mathcal{X} \subseteq \mathbb{R}^d$, the reward is $r(x) = x^\top \theta^* + \zeta$, where $\zeta$ is an $R$-subGaussian random variable for some constant $R > 0$. Note that this assumption is automatically satisfied if the rewards are bounded almost surely.

We first state the $\widetilde{O}(d\sqrt{T})$-regret bound for linear contextual bandits from the Langevin Monte Carlo smoothed Feel-Good Thompson Sampling (SFGLMCTS) methodology in Huix et al. (2023).

**Theorem C.10** (Theorem 5 in Huix et al. (2023))**.** *Let the prior distribution $p_0$ for FGLMCTS satisfy a $m_0$-strongly log concave property for some $m_0 \geq 0$. The regret for FGLMCTS when run for $T$ time steps (under Assumption C.2) in linear contextual bandits is bounded by*

$$R(T) \leq C\left(\frac{d}{1+m_0} + \sqrt{R} + m_0\right)\sqrt{T \log^3(dT)} \leq \widetilde{O}(d\sqrt{T}).$$

**Theorem C.11.** *Under Assumption C.2, the SFGHMCTS procedure in Algorithm 3 achieves $\widetilde{O}(d\sqrt{T})$ regret for linear contextual bandits when run for $T$ time steps.*

*Proof.* The proof follows by a direct reduction from the FGLMCTS setting in Huix et al. (2023). From Lemma C.6, each inner iteration of Algorithm 3 converges to $\pi(\theta) \propto e^{-\mathcal{L}_t(\theta)}$, where $\mathcal{L}_t(\theta)$ is the same smoothed posterior (SFGTS) that each inner loop in FGLMCTS also converges to. Then, since FGLMCTS achieves $\widetilde{O}(d\sqrt{T})$ regret via Theorem C.3, the regret for SFGHMCTS is also $\widetilde{O}(d\sqrt{T})$, proving the theorem. ☐

**Remark C.11.1.** *This regret bound of $\widetilde{O}(d\sqrt{T})$ also extends to variants of SFGHMCTS with stochastic gradients (Zou et al., 2018), dampeners (Patterson and Teh, 2013), and preconditioners (Li et al., 2016), since these factors each only affect the convergence rate in each inner loop of the algorithm that approximates the posterior.*

# D Algorithm Descriptions

This section provides a detailed parameter description of each algorithm we conduct numerical experiments on.

Table 7: Detailed Description of the Algorithms in the Linear Experiments.

| Algorithm | Description |
|---|---|
| LinUCB (Li et al., 2010) | $\alpha = 0.1$ |
| EpsGreedy | $\epsilon = 0.01$. |
| LinTS | $\eta = 1$. |
| LMCTS | $\eta = 1$ |
| PLMCTS | $\eta = 1, \lambda_{\text{reg}} = 1$. |
| FGLMCTS-L1B1 | $\lambda = 0.01, \beta = 10^3, b = 1000, \eta = 1$. |
| FGLMCTS-L2B1 | $\lambda = 0.1, \beta = 10^3, b = 1000, \eta = 1$. |
| FGLMCTS-L3B1 | $\lambda = 0.5, \beta = 10^3, b = 1000, \eta = 1$. |
| FGLMCTS-L4B1 | $\lambda = 1, \beta = 10^3, b = 1000, \eta = 1$. |
| FGLMCTS-L1B2 | $\lambda = 0.01, \beta = 1, b = 1000, \eta = 1$. |
| FGLMCTS-L2B2 | $\lambda = 0.1, \beta = 1, b = 1000, \eta = 1$. |
| FGLMCTS-L3B2 | $\lambda = 0.5, \beta = 1, b = 1000, \eta = 1$. |
| FGLMCTS-L4B2 | $\lambda = 1, \beta = 1, b = 1000, \eta = 1$. |
| PFGLMCTS-L1B1 | $\lambda = 0.01, \beta = 10^3, b = 1000, \eta = 1$. |
| PFGLMCTS-L2B1 | $\lambda = 0.1, \beta = 10^3, b = 1000, \eta = 1$. |
| PFGLMCTS-L3B1 | $\lambda = 0.5, \beta = 10^3, b = 1000, \eta = 1$. |
| PFGLMCTS-L4B1 | $\lambda = 1, \beta = 10^3, b = 1000, \eta = 1$. |
| PFGLMCTS-L1B2 | $\lambda = 0.01, \beta = 1, b = 1000, \eta = 1$. |
| PFGLMCTS-L2B2 | $\lambda = 0.1, \beta = 1, b = 1000, \eta = 1$. |
| PFGLMCTS-L3B2 | $\lambda = 0.5, \beta = 1, b = 1000, \eta = 1$. |
| PFGLMCTS-L4B2 | $\lambda = 1, \beta = 1, b = 1000, \eta = 1$. |
| SFGLMCTS-L1B1 | $\lambda = 0.01, \beta = 10^3, b = 1000, \eta = 1, \lambda_{\text{reg}} = 1, s = 10$. |
| SFGLMCTS-L2B1 | $\lambda = 0.1, \beta = 10^3, b = 1000, \eta = 1, \lambda_{\text{reg}} = 1, s = 10$. |
| SFGLMCTS-L3B1 | $\lambda = 0.5, \beta = 10^3, b = 1000, \eta = 1, \lambda_{\text{reg}} = 1, s = 10$. |
| SFGLMCTS-L4B1 | $\lambda = 1, \beta = 10^3, b = 1000, \eta = 1, \lambda_{\text{reg}} = 1, s = 10$. |
| SFGLMCTS-L1B2 | $\lambda = 0.01, \beta = 1, b = 1000, \eta = 1, \lambda_{\text{reg}} = 1, s = 10$. |
| SFGLMCTS-L2B2 | $\lambda = 0.1, \beta = 1, b = 1000, \eta = 1, \lambda_{\text{reg}} = 1, s = 10$. |
| SFGLMCTS-L3B2 | $\lambda = 0.5, \beta = 1, b = 1000, \eta = 1, \lambda_{\text{reg}} = 1, s = 10$. |
| SFGLMCTS-L4B2 | $\lambda = 1, \beta = 1, b = 1000, \eta = 1, \lambda_{\text{reg}} = 1, s = 10$. |
| PSFGLMCTS-L1B1 | $\lambda = 0.01, \beta = 10^3, b = 1000, \eta = 1, \lambda_{\text{reg}} = 1, s = 10$. |
| PSFGLMCTS-L2B1 | $\lambda = 0.1, \beta = 10^3, b = 1000, \eta = 1, \lambda_{\text{reg}} = 1, s = 10$. |
| PSFGLMCTS-L3B1 | $\lambda = 0.5, \beta = 10^3, b = 1000, \eta = 1, \lambda_{\text{reg}} = 1, s = 10$. |
| PSFGLMCTS-L4B1 | $\lambda = 1, \beta = 10^3, b = 1000, \eta = 1, \lambda_{\text{reg}} = 1, s = 10$. |
| PSFGLMCTS-L1B2 | $\lambda = 0.01, \beta = 1, b = 1000, \eta = 1, \lambda_{\text{reg}} = 1, s = 10$. |
| PSFGLMCTS-L2B2 | $\lambda = 0.1, \beta = 1, b = 1000, \eta = 1, \lambda_{\text{reg}} = 1, s = 10$. |
| PSFGLMCTS-L3B2 | $\lambda = 0.5, \beta = 1, b = 1000, \eta = 1, \lambda_{\text{reg}} = 1, s = 10$. |
| PSFGLMCTS-L4B2 | $\lambda = 1, \beta = 1, b = 1000, \eta = 1, \lambda_{\text{reg}} = 1, s = 10$. |
| SVRGLMCTS | $\eta = 1$, batch size $= 64$. |
| HMCTS | $\eta = 1$. |
| PHMCTS | $\eta = 1$, leapfrog-step $= 10$. |
| FGHMCTS | $\lambda = 0.5, \eta = 1, b = 1000$, leapfrog-step $= 10$. |
| PFGHMCTS | $\lambda = 0.5, \eta = 1, b = 1000, \lambda_{\text{reg}} = 1$, leapfrog-step $= 10$. |
| SFGHMCTS | $\lambda = 0.5, \eta = 1, b = 1000, s = 10$, leapfrog-step $= 10$. |
| PSFGHMCTS | $\lambda = 0.5, \eta = 1, b = 1000, s = 10, \lambda_{\text{reg}} = 1$, leapfrog-step $= 10$. |
| MALATS | $\eta = 1$. |
| FGMALATS-L1B1 | $\lambda = 0.01, \beta = 10^3, b = 1000, \eta = 1$. |
| FGMALATS-L2B1 | $\lambda = 0.1, \beta = 10^3, b = 1000, \eta = 1$. |
| FGMALATS-L3B1 | $\lambda = 0.5, \beta = 10^3, b = 1000, \eta = 1$. |
| FGMALATS-L4B1 | $\lambda = 1, \beta = 10^3, b = 1000, \eta = 1$. |
| FGMALATS-L1B2 | $\lambda = 0.01, \beta = 1, b = 1000, \eta = 1$. |
| FGMALATS-L2B2 | $\lambda = 0.1, \beta = 1, b = 1000, \eta = 1$. |
| FGMALATS-L3B2 | $\lambda = 0.5, \beta = 1, b = 1000, \eta = 1$. |
| FGMALATS-L4B2 | $\lambda = 1, \beta = 1, b = 1000, \eta = 1$. |
| SFGMALATS-L1B1 | $\lambda = 0.01, \beta = 10^3, b = 1000, \eta = 1, \lambda_{\text{reg}} = 1, s = 10$. |
| SFGMALATS-L2B1 | $\lambda = 0.1, \beta = 10^3, b = 1000, \eta = 1, \lambda_{\text{reg}} = 1, s = 10$. |
| SFGMALATS-L3B1 | $\lambda = 0.5, \beta = 10^3, b = 1000, \eta = 1, \lambda_{\text{reg}} = 1, s = 10$. |
| SFGMALATS-L4B1 | $\lambda = 1, \beta = 10^3, b = 1000, \eta = 1, \lambda_{\text{reg}} = 1, s = 10$. |
| SFGMALATS-L1B2 | $\lambda = 0.01, \beta = 1, b = 1000, \eta = 1, \lambda_{\text{reg}} = 1, s = 10$. |
| SFGMALATS-L2B2 | $\lambda = 0.1, \beta = 1, b = 1000, \eta = 1, \lambda_{\text{reg}} = 1, s = 10$. |
| SFGMALATS-L3B2 | $\lambda = 0.5, \beta = 1, b = 1000, \eta = 1, \lambda_{\text{reg}} = 1, s = 10$. |
| SFGMALATS-L4B2 | $\lambda = 1, \beta = 1, b = 1000, \eta = 1, \lambda_{\text{reg}} = 1, s = 10$. |
| ULMC | $\eta = 1, \gamma = 0.1$ |
| UFGLMCTS | $\lambda = 0.01, \beta = 10^3, b = 1000, \eta = 1, \gamma = 0.1$ |
| USFGLMCTS | $\lambda = 0.01, \beta = 10^3, b = 1000, \eta = 1, \gamma = 0.1, \lambda_{\text{reg}} = 1, s = 10$ |
| Uniform | probability $= 0.5$. |

Table 8: Detailed Description of the Algorithms in the Logistic Experiments.

| Algorithm | Description |
| --- | --- |
| EpsGreedy | $\epsilon = 0.01$. |
| LinTS | $\eta = 1$. |
| LMCTS | $\eta = 1$ |
| PLMCTS | $\eta = 1$, $\lambda_{\text{reg}} = 1$. |
| FGLMCTS-L1B1 | $\lambda = 0.01$, $\beta = 10^3$, $b = 1000$, $\eta = 10$. |
| FGLMCTS-L2B1 | $\lambda = 0.1$, $\beta = 10^3$, $b = 1000$, $\eta = 10$. |
| FGLMCTS-L3B1 | $\lambda = 0.5$, $\beta = 10^3$, $b = 1000$, $\eta = 10$. |
| FGLMCTS-L4B1 | $\lambda = 1$, $\beta = 10^3$, $b = 1000$, $\eta = 10$. |
| FGLMCTS-L1B2 | $\lambda = 0.01$, $\beta = 1$, $b = 1000$, $\eta = 10$. |
| FGLMCTS-L2B2 | $\lambda = 0.1$, $\beta = 1$, $b = 1000$, $\eta = 10$. |
| FGLMCTS-L3B2 | $\lambda = 0.5$, $\beta = 1$, $b = 1000$, $\eta = 10$. |
| FGLMCTS-L4B2 | $\lambda = 1$, $\beta = 1$, $b = 1000$, $\eta = 10$. |
| PFGLMCTS-L1B1 | $\lambda = 0.01$, $\beta = 10^3$, $b = 1000$, $\eta = 10$. |
| PFGLMCTS-L2B1 | $\lambda = 0.1$, $\beta = 10^3$, $b = 1000$, $\eta = 10$. |
| PFGLMCTS-L3B1 | $\lambda = 0.5$, $\beta = 10^3$, $b = 1000$, $\eta = 10$. |
| PFGLMCTS-L4B1 | $\lambda = 1$, $\beta = 10^3$, $b = 1000$, $\eta = 10$. |
| PFGLMCTS-L1B2 | $\lambda = 0.01$, $\beta = 1$, $b = 1000$, $\eta = 10$. |
| PFGLMCTS-L2B2 | $\lambda = 0.1$, $\beta = 1$, $b = 1000$, $\eta = 10$. |
| PFGLMCTS-L3B2 | $\lambda = 0.5$, $\beta = 1$, $b = 1000$, $\eta = 10$. |
| PFGLMCTS-L4B2 | $\lambda = 1$, $\beta = 1$, $b = 1000$, $\eta = 10$. |
| SFGLMCTS-L1B1 | $\lambda = 0.01$, $\beta = 10^3$, $b = 1000$, $\eta = 10$, $\lambda_{\text{reg}} = 1$, $s = 10$. |
| SFGLMCTS-L2B1 | $\lambda = 0.1$, $\beta = 10^3$, $b = 1000$, $\eta = 10\lambda_{\text{reg}} = 1$, $s = 10$. |
| SFGLMCTS-L3B1 | $\lambda = 0.5$, $\beta = 10^3$, $b = 1000$, $\eta = 10$, $\lambda_{\text{reg}} = 1$, $s = 10$. |
| SFGLMCTS-L4B1 | $\lambda = 1$, $\beta = 10^3$, $b = 1000$, $\eta = 10$, $\lambda_{\text{reg}} = 1$, $s = 10$. |
| SFGLMCTS-L1B2 | $\lambda = 0.01$, $\beta = 1$, $b = 1000$, $\eta = 10$, $\lambda_{\text{reg}} = 1$, $s = 10$. |
| SFGLMCTS-L2B2 | $\lambda = 0.1$, $\beta = 1$, $b = 1000$, $\eta = 10$, $\lambda_{\text{reg}} = 1$, $s = 10$. |
| SFGLMCTS-L3B2 | $\lambda = 0.5$, $\beta = 1$, $b = 1000$, $\eta = 10$, $\lambda_{\text{reg}} = 1$, $s = 10$. |
| SFGLMCTS-L4B2 | $\lambda = 1$, $\beta = 1$, $b = 1000$, $\eta = 10$, $\lambda_{\text{reg}} = 1$, $s = 10$. |
| PSFGLMCTS-L1B1 | $\lambda = 0.01$, $\beta = 10^3$, $b = 1000$, $\eta = 10$, $\lambda_{\text{reg}} = 1$, $s = 10$. |
| PSFGLMCTS-L2B1 | $\lambda = 0.1$, $\beta = 10^3$, $b = 1000$, $\eta = 10$, $\lambda_{\text{reg}} = 1$, $s = 10$. |
| PSFGLMCTS-L3B1 | $\lambda = 0.5$, $\beta = 10^3$, $b = 1000$, $\eta = 10$, $\lambda_{\text{reg}} = 1$, $s = 10$. |
| PSFGLMCTS-L4B1 | $\lambda = 1$, $\beta = 10^3$, $b = 1000$, $\eta = 10$, $\lambda_{\text{reg}} = 1$, $s = 10$. |
| PSFGLMCTS-L1B2 | $\lambda = 0.01$, $\beta = 1$, $b = 1000$, $\eta = 10$, $\lambda_{\text{reg}} = 1$, $s = 10$. |
| PSFGLMCTS-L2B2 | $\lambda = 0.1$, $\beta = 1$, $b = 1000$, $\eta = 10$, $\lambda_{\text{reg}} = 1$, $s = 10$. |
| PSFGLMCTS-L3B2 | $\lambda = 0.5$, $\beta = 1$, $b = 1000$, $\eta = 10$, $\lambda_{\text{reg}} = 1$, $s = 10$. |
| PSFGLMCTS-L4B2 | $\lambda = 1$, $\beta = 1$, $b = 1000$, $\eta = 10$, $\lambda_{\text{reg}} = 1$, $s = 10$. |
| SVRGLMCTS | $\eta = 1$, batch size $= 64$. |
| MALATS | $\eta = 1$. |
| FGMALATS-L1B1 | $\lambda = 0.01$, $\beta = 10^3$, $b = 1000$, $\eta = 10$. |
| FGMALATS-L2B1 | $\lambda = 0.1$, $\beta = 10^3$, $b = 1000$, $\eta = 10$. |
| FGMALATS-L3B1 | $\lambda = 0.5$, $\beta = 10^3$, $b = 1000$, $\eta = 10$. |
| FGMALATS-L4B1 | $\lambda = 1$, $\beta = 10^3$, $b = 1000$, $\eta = 10$. |
| FGMALATS-L1B2 | $\lambda = 0.01$, $\beta = 1$, $b = 1000$, $\eta = 10$. |
| FGMALATS-L2B2 | $\lambda = 0.1$, $\beta = 1$, $b = 1000$, $\eta = 10$. |
| FGMALATS-L3B2 | $\lambda = 0.5$, $\beta = 1$, $b = 1000$, $\eta = 10$. |
| FGMALATS-L4B2 | $\lambda = 1$, $\beta = 1$, $b = 1000$, $\eta = 10$. |
| SFGMALATS-L1B1 | $\lambda = 0.01$, $\beta = 10^3$, $b = 1000$, $\eta = 10$, $\lambda_{\text{reg}} = 1$, $s = 10$. |
| SFGMALATS-L2B1 | $\lambda = 0.1$, $\beta = 10^3$, $b = 1000$, $\eta = 10\lambda_{\text{reg}} = 1$, $s = 10$. |
| SFGMALATS-L3B1 | $\lambda = 0.5$, $\beta = 10^3$, $b = 1000$, $\eta = 10$, $\lambda_{\text{reg}} = 1$, $s = 10$. |
| SFGMALATS-L4B1 | $\lambda = 1$, $\beta = 10^3$, $b = 1000$, $\eta = 10$, $\lambda_{\text{reg}} = 1$, $s = 10$. |
| SFGMALATS-L1B2 | $\lambda = 0.01$, $\beta = 1$, $b = 1000$, $\eta = 10$, $\lambda_{\text{reg}} = 1$, $s = 10$. |
| SFGMALATS-L2B2 | $\lambda = 0.1$, $\beta = 1$, $b = 1000$, $\eta = 10$, $\lambda_{\text{reg}} = 1$, $s = 10$. |
| SFGMALATS-L3B2 | $\lambda = 0.5$, $\beta = 1$, $b = 1000$, $\eta = 10$, $\lambda_{\text{reg}} = 1$, $s = 10$. |
| SFGMALATS-L4B2 | $\lambda = 1$, $\beta = 1$, $b = 1000$, $\eta = 10$, $\lambda_{\text{reg}} = 1$, $s = 10$. |

Table 9: Detailed Description of the Algorithms in the Neural Experiments: Adult (with $\ell_2$ loss).

| Algorithm | Description |
| --- | --- |
| Neural-$\epsilon$-Greedy | $\epsilon = 0.01$. |
| LMCTS | $\beta^{-1} = 0.00001$, LeakyReLU. |
| FGLMCTS | $\beta^{-1} = 0.00001$, LeakyReLU, $\lambda = 0.1$, $b = 10$. |
| SFGLMCTS | $\beta^{-1} = 0.00001$, LeakyReLU, $\lambda = 0.1$, $b = 10$, $s = 10$. |
| NeuralUCB | $\lambda_{\text{reg}} = 0.001$. |
| NeuralTS | ReLU, $\nu = 0.000001$, $\lambda_{\text{reg}} = 0.01$ |
| FG-NeuralTS | ReLU, $\nu = 0.000001$, $\lambda_{\text{reg}} = 0.01$, $\lambda = 1$, $b = 1$ . |
| SFG-NeuralTS | ReLU, $\nu = 0.000001$, $\lambda_{\text{reg}} = 0.01$, $\lambda = 1$, $b = 1$, $s = 10$. |

Table 10: Detailed Description of the Algorithms in the Neural Experiments: Covertype ($\ell_2$ loss).

| Algorithm | Description |
|---|---|
| Neural-$\epsilon$-Greedy | $\epsilon = 0.4$. |
| LMCTS | BCE loss, $\beta^{-1} = 0.000001$, LeakyReLU. |
| FGLMCTS | BCE loss, $\beta^{-1} = 0.00001$, LeakyReLU, $\lambda = 0.05$, $b = 10$. |
| SFGLMCTS | BCE loss, $\beta^{-1} = 0.00001$, LeakyReLU, $\lambda = 0.05$, $b = 10$, $s = 10$. |
| NeuralUCB | BCE loss, $\lambda_{\mathrm{reg}} = 0.001$. |
| NeuralTS | ReLU, $\nu = 0.01$, $\lambda_{\mathrm{reg}} = 0.001$. |
| FG-NeuralTS | ReLU, $\nu = 0.01$, $\lambda_{\mathrm{reg}} = 0.001$, $\lambda = 1.0$, $b = 1.0$. |
| SFG-NeuralTS | ReLU, $\nu = 0.01$, $\lambda_{\mathrm{reg}} = 0.001$, $\lambda = 1.0$, $b = 1.0$, $s = 10$. |

Table 11: Detailed Description of the Algorithms in the Neural Experiments: Magic ($\ell_2$ loss).

| Algorithm | Description |
|---|---|
| Neural-$\epsilon$-Greedy | $\epsilon = 0.1$. |
| LMCTS | $\beta^{-1} = 0.001$, ReLU. |
| FGLMCTS | $\beta^{-1} = 0.001$, ReLU, $\lambda = 0.05$, $b = 10$. |
| SFGLMCTS | $\beta^{-1} = 0.001$, ReLU, $\lambda = 0.05$, $b = 10$, $s = 10$. |
| NeuralUCB | $\lambda_{\mathrm{reg}} = 0.001$. |
| NeuralTS | ReLU, $\nu = 0.00001$. |
| FG-NeuralTS | ReLU, $\nu = 0.00001$, $\lambda = 1$, $b = 1$. |
| SFG-NeuralTS | ReLU, $\nu = 0.00001$, $\lambda = 1$, $b = 1$, $s = 10$. |

Table 12: Detailed Description of the Algorithms in the Neural Experiments: Mushroom ($\ell_2$ loss).

| Algorithm | Description |
|---|---|
| Neural-$\epsilon$-Greedy | $\epsilon = 0.1$. |
| LMCTS | $\beta^{-1} = 0.00001$, LeakyReLU. |
| FGLMCTS | $\beta^{-1} = 0.00001$, LeakyReLU, $\lambda = 0.05$, $b = 10$. |
| SFGLMCTS | $\beta^{-1} = 0.00001$, LeakyReLU, $\lambda = 0.05$, $b = 10$, $s = 10$. |
| NeuralUCB | $\lambda_{\mathrm{reg}} = 0.00001$. |
| NeuralTS | ReLU, $\nu = 0.00001$, $\lambda_{\mathrm{reg}} = 0.00001$. |
| FG-NeuralTS | ReLU, $\nu = 0.00001$, $\lambda_{\mathrm{reg}} = 0.00001$, $\lambda = 1$, $b = 1$. |
| SFG-NeuralTS | ReLU, $\nu = 0.00001$, $\lambda_{\mathrm{reg}} = 0.00001$, $\lambda = 1$, $b = 1$, $s = 10$. |

Table 13: Detailed Description of the Algorithms in the Neural Experiments: Shuttle ($\ell_2$ loss).

| Algorithm | Description |
|---|---|
| Neural-$\epsilon$-Greedy | $\epsilon = 0.01$. |
| LMCTS | $\beta^{-1} = 0.0001$, LeakyReLU. |
| FGLMCTS | $\beta^{-1} = 0.0001$, LeakyReLU, $\lambda = 0.05$, $b = 10$. |
| SFGLMCTS | $\beta^{-1} = 0.0001$, LeakyReLU, $\lambda = 0.05$, $b = 10$, $s = 10$. |
| NeuralUCB | $\lambda_{\mathrm{reg}} = 0.0001$. |
| NeuralTS | ReLU, $\nu = 0.000001$, $\lambda_{\mathrm{reg}} = 0.01$. |
| FG-NeuralTS | ReLU, $\nu = 0.000001$, $\lambda_{\mathrm{reg}} = 0.01$, $\lambda = 1$, $b = 1$. |
| SFG-NeuralTS | ReLU, $\nu = 0.000001$, $\lambda_{\mathrm{reg}} = 0.01$, $\lambda = 1$, $b = 1$, $s = 1-$. |

Table 14: Detailed Description of the Algorithms in the Neural Experiments: MNIST_784 ($\ell_2$ loss).

| Algorithm | Description |
|---|---|
| Neural-$\epsilon$-Greedy | $\epsilon = 0.01$. |
| LMCTS | $\beta^{-1} = 0.00001$, LeakyReLU. |
| FGLMCTS | $\beta^{-1} = 0.00001$, LeakyReLU, $\lambda = 0.05$, $b = 10$. |
| SFGLMCTS | $\beta^{-1} = 0.00001$, LeakyReLU, $\lambda = 0.05$, $b = 10$, $s = 10$. |
| NeuralUCB | $\lambda_{\text{reg}} = 0.00001$. |
| NeuralTS | ReLU, $\nu = 0.00001$, $\lambda_{\text{reg}} = 0.00001$. |
| FG-NeuralTS | ReLU, $\nu = 0.00001$, $\lambda_{\text{reg}} = 0.00001$, $\lambda = 0.05$, $b = 10$. |
| SFG-NeuralTS | ReLU, $\nu = 0.00001$, $\lambda_{\text{reg}} = 0.00001$, $\lambda = 0.05$, $b = 10$, $s = 10$. |

Table 15: Detailed Description of the Algorithms in the Neural Experiments: FINANCIAL ($\ell_2$ loss).

| Algorithm | Description |
|---|---|
| Neural-$\epsilon$-Greedy | $\epsilon = 0.1$. |
| LMCTS | $\beta^{-1} = 0.001$, LeakyReLU. |
| FGLMCTS | $\beta^{-1} = 0.001$, LeakyReLU, $\lambda = 0.1$, $b = 1.0$. |
| SFGLMCTS | $\beta^{-1} = 0.001$, LeakyReLU, $\lambda = 0.1$, $b = 1.0$, $s = 10$. |
| NeuralUCB | $\lambda_{\text{reg}} = 0.0001$ |
| NeuralTS | ReLU, $\nu = 0.000001$, $\lambda_{\text{reg}} = 0.01$. |
| FG-NeuralTS | ReLU, $\nu = 0.000001$, $\lambda_{\text{reg}} = 0.01$, $\lambda = 0.01$, $b = 1$. |
| SFG-NeuralTS | ReLU, $\nu = 0.000001$, $\lambda_{\text{reg}} = 0.01$, $\lambda = 0.01$, $b = 1$, $s = 10$. |

Table 16: Detailed Description of the Algorithms in the Neural Experiments: JESTER ($\ell_2$ loss).

| Algorithm | Description |
|---|---|
| Neural-$\epsilon$-Greedy | $\epsilon = 0.1$. |
| LMCTS | $\beta^{-1} = 0.001$, LeakyReLU. |
| FGLMCTS | $\beta^{-1} = 0.001$, LeakyReLU, $\lambda = 0.1$, $b = 1.0$. |
| SFGLMCTS | $\beta^{-1} = 0.001$, LeakyReLU, $\lambda = 0.1$, $b = 1.0$, $s = 10$. |
| NeuralUCB | $\lambda_{\text{reg}} = 0.0001$. |
| NeuralTS | ReLU, $\nu = 0.000001$, $\lambda_{\text{reg}} = 0.01$. |
| FG-NeuralTS | ReLU, $\nu = 0.000001$, $\lambda_{\text{reg}} = 0.01$, $\lambda = 1$, $b = 1$. |
| SFG-NeuralTS | ReLU, $\nu = 0.000001$, $\lambda_{\text{reg}} = 0.01$, $\lambda = 1$, $b = 1$, $s = 10$. |

Table 17: Detailed Description of the Algorithms in the Neural Experiments: RESTAURANTRATINGS ($\ell_2$ loss).

| Algorithm | Description |
|---|---|
| Neural-$\epsilon$-Greedy | $\epsilon = 0.01$. |
| LMCTS | $\beta^{-1} = 0.0001$, LeakyReLU. |
| FGLMCTS | $\beta^{-1} = 0.0001$, LeakyReLU, $\lambda = 0.01$, $b = 1.0$. |
| SFGLMCTS | $\beta^{-1} = 0.0001$, LeakyReLU, $\lambda = 0.01$, $b = 1.0$, $s = 10$. |
| NeuralUCB | $\lambda_{\text{reg}} = 0.01$. |
| NeuralTS | ReLU, $\nu = 0.00001$, $\lambda_{\text{reg}} = 0.01$. |
| FG-NeuralTS | ReLU, $\nu = 0.00001$, $\lambda_{\text{reg}} = 0.01$, $\lambda = 0.01$, $b = 1.0$. |
| SFG-NeuralTS | ReLU, $\nu = 0.00001$, $\lambda_{\text{reg}} = 0.01$, $\lambda = 0.01$, $b = 1.0$, $s = 10$. |

Table 18: Detailed Description of the Algorithms in the Neural Experiments: CIFAR-10 ($\ell_2$ loss).

| Algorithm | Description |
|---|---|
| Neural-$\epsilon$-Greedy | $\epsilon = 0.01$. |
| LMCTS | $\beta^{-1} = 0.000001$, LeakyReLU. |
| FGLMCTS | $\beta^{-1} = 0.000001$, LeakyReLU, $\lambda = 0.01, b = 1.0$. |
| SFGLMCTS | $\beta^{-1} = 0.000001$, LeakyReLU, $\lambda = 0.01, b = 1.0, s = 10$. |
| NeuralUCB | $\lambda_{\text{reg}} = 0.01$. |
| NeuralTS | ReLU, $\nu = 0.00001, \lambda_{\text{reg}} = 0.01$. |
| FG-NeuralTS | ReLU, $\nu = 0.00001, \lambda_{\text{reg}} = 0.01, \lambda = 0.01, b = 1.0$. |
| SFG-NeuralTS | ReLU, $\nu = 0.00001, \lambda_{\text{reg}} = 0.01, \lambda = 0.01, b = 1.0, s = 10$. |

# E  Further Experimental Results

This section lists all our main experimental results.

## E.1  Linear Contextual Bandits

This subsection provides our main experimental results for our linear contextual bandit experiments.

**Low-dimensional setting.** For the $d = 20$ setting, Tables 19 and 20 provide our experiments over linear contextual bandits for $\beta = 10^3$ and $\beta = 1$ (respectively), where we modify the feel-good parameter $\lambda$ in the loss-likelihood for values $\lambda \in \{0.01, 0.1, 0.5, 1.0\}$. The corresponding vanilla Thompson sampling cumulative regret values are equivalent to setting $\lambda = 0$. For these values, we direct the reader to Table 1. These experiments strongly indicate that setting $\lambda = 0.01$ outperforms the case where $\lambda = 0$ (as well as other choices of $\lambda$). Especially for SFGMALATS, which enjoys optimal regret in this setting, the regret is considerably smaller than that of LMCTS and FGLMCTS.

Table 19: Cumulative Regret incurred by the Linear bandits in $d = 20$ and $\beta = 10^3$. Values reported are the mean over 10 independent trials with standard deviation.

| Algorithm | $\lambda = 0.01$ | $\lambda = 0.1$ | $\lambda = 0.5$ | $\lambda = 1.0$ |
|---|---|---|---|---|
| FGLMCTS | $62.7 \pm 12.0$ | $\mathbf{50.8} \pm 15.8$ | $235.8 \pm 185.8$ | $612.7 \pm 340.4$ |
| FGMALATS | $62.7 \pm 25.5$ | $63.7 \pm 14.5$ | $212.8 \pm 170.0$ | $528.3 \pm 377.2$ |
| PFGLMCTS | $132.7 \pm 19.1$ | $140.3 \pm 24.4$ | $193.2 \pm 47.2$ | $332.9 \pm 221.2$ |
| PSFGLMCTS | $137.1 \pm 27.5$ | $130.5 \pm 13.8$ | $183.1 \pm 37.3$ | $286.1 \pm 134.1$ |
| SFGLMCTS | $65.2 \pm 12.3$ | $82.0 \pm 32.6$ | $223.8 \pm 190.1$ | $546.0 \pm 357.6$ |
| SFGMALATS | $\mathbf{56.2} \pm 22.8$ | $68.4 \pm 26.2$ | $\mathbf{173.4} \pm 110.1$ | $\mathbf{486.2} \pm 347.1$ |

Table 20: Cumulative Regret incurred by the Linear bandits in $d = 20$ and $\beta = 1$. Values reported are the mean over 10 independent trials with standard deviation.

| Algorithm | $\lambda = 0.01$ | $\lambda = 0.1$ | $\lambda = 0.5$ | $\lambda = 1$ |
|---|---|---|---|---|
| FGLMCTS | $2019.4 \pm 163.3$ | $2031.8 \pm 163.3$ | $2058.1 \pm 203.8$ | $2164.5 \pm 203.2$ |
| FGMALATS | $\mathbf{61.5} \pm 32.9$ | $\mathbf{58.1} \pm 11.0$ | $\mathbf{337.7} \pm 162.4$ | $760.0 \pm 310.2$ |
| PFGLMCTS | $2177.0 \pm 168.8$ | $2070.4 \pm 189.0$ | $2419.6 \pm 384.9$ | $2472.2 \pm 409.3$ |
| PSFGLMCTS | $2196.4 \pm 338.4$ | $2252.5 \pm 286.5$ | $2296.2 \pm 324.4$ | $2157.8 \pm 414.3$ |
| SFGLMCTS | $2003.2 \pm 162.2$ | $2018.2 \pm 170.6$ | $2062.7 \pm 181.2$ | $2112.4 \pm 211.7$ |
| SFGMALATS | $70.3 \pm 43.5$ | $669.1 \pm 1839.6$ | $366.0 \pm 183.6$ | $\mathbf{758.7} \pm 303.5$ |

**High-dimensional setting.** For the $d = 40$ setting, Tables 21 and 22 provide our experiments over linear contextual bandits for $\beta = 10^3$ and $\beta = 1$ (respectively), where we modify the feel-good parameter $\lambda$ in the loss-likelihood for values $\lambda \in \{0.01, 0.1, 0.5, 1.0\}$. As before, we see that setting the feel-good parameter $\lambda$ to be 0.01 outperforms the case where $\lambda = 0$ (as well as other choices of $\lambda$). As before, this particularly carries forward for FGMALATS and SFGMALATS which experience the lowest cumulative regrets.

Table 21: Cumulative Regret incurred by the Linear bandits in $d = 40$ and $\beta = 10^3$. Values reported are the mean over 10 independent trials with standard deviation.

| Algorithm | $\lambda = 0.01$ | $\lambda = 0.1$ | $\lambda = 0.5$ | $\lambda = 1.0$ |
|---|---|---|---|---|
| FGLMCTS | $131.2 \pm 11.7$ | $122.1 \pm 15.1$ | $178.4 \pm 21.0$ | $241.1 \pm 33.8$ |
| FGMALATS | $100.4 \pm 20.9$ | $\mathbf{99.0} \pm 20.6$ | $153.9 \pm 25.1$ | $269.3 \pm 122.6$ |
| PFGLMCTS | $150.1 \pm 15.8$ | $135.5 \pm 15.4$ | $179.8 \pm 23.2$ | $274.2 \pm 39.0$ |
| PSFGLMCTS | $142.0 \pm 15.1$ | $169.5 \pm 20.2$ | $208.4 \pm 24.8$ | $277.9 \pm 43.1$ |
| SFGLMCTS | $126.2 \pm 21.7$ | $131.1 \pm 6.5$ | $176.3 \pm 18.7$ | $262.8 \pm 76.5$ |
| SFGMALATS | $\mathbf{98.2} \pm 19.5$ | $102.6 \pm 14.4$ | $\mathbf{143.0} \pm 20.2$ | $\mathbf{215.1} \pm 54.0$ |

Table 22: Cumulative Regret incurred by the Linear bandits in $d = 40$ and $\beta = 1$. Values reported are the mean over 10 independent trials with standard deviation.

| Algorithm | $\lambda = 0.01$ | $\lambda = 0.1$ | $\lambda = 0.5$ | $\lambda = 1.0$ |
|---|---|---|---|---|
| FGLMCTS | $4527.6 \pm 169.2$ | $4514.4 \pm 179.7$ | $4447.9 \pm 147.8$ | $4400.5 \pm 151.3$ |
| FGMALATS | $\mathbf{94.3} \pm 23.4$ | $\mathbf{98.2} \pm 24.2$ | $140.8 \pm 25.4$ | $428.7 \pm 166.7$ |
| PFGLMCTS | $4930.6 \pm 321.8$ | $4802.5 \pm 444.9$ | $4725.0 \pm 491.9$ | $4642.6 \pm 328.6$ |
| PSFGLMCTS | $4855.1 \pm 350.1$ | $4852.9 \pm 351.2$ | $4708.2 \pm 300.2$ | $4665.1 \pm 435.1$ |
| SFGLMCTS | $4499.2 \pm 118.1$ | $4486.4 \pm 135.7$ | $4406.7 \pm 139.6$ | $4395.4 \pm 166.3$ |
| SFGMALATS | $100.1 \pm 19.2$ | $110.6 \pm 12.7$ | $\mathbf{130.1} \pm 16.7$ | $\mathbf{425.7} \pm 155.3$ |

### E.1.1 Posterior Analysis

A central theme of our work is that given MCMC-based TS variants compute an approximate posterior $\widetilde{\pi}_t$ through sampling procedures, we wish to measure how faithfully these MCMC approximations capture the true Bayesian posterior? To disentangle these factors, following Riquelme et al. (2018), we conduct a controlled posterior quality analysis where all algorithms observe identical data, allowing us to isolate approximation quality from exploration strategy. Our analysis is motivated by the following observation: if $\widetilde{\pi}_t$ is consistently close to the true posterior $\pi_t$ across timesteps and arms, then the MCMC approximation preserves the Bayesian reasoning that underpins TS' theoretical properties. Conversely, significant divergence between $\widetilde{\pi}_t$ and $\pi_t$ suggests that the algorithm's beliefs are systematically biased, potentially undermining exploration-exploitation balance.

We consider a linear contextual bandit with $K = 6$ arms in $d = 20$ dimensions, following the model proposed by Russo and Roy (2016). For each arm $i \in \{1, \ldots, K\}$, the reward function is:

$$r_{i,t} = X_t^\top \beta_i + \epsilon_t, \quad \epsilon_t \sim \mathcal{N}(0, \sigma^2)$$

where $X_t \in \mathbb{R}^d$ is the context vector and $\beta_i \in \mathbb{R}^d$ is the arm-specific parameter. We set the prior $\beta_i \sim \mathcal{N}(0, \lambda^{-1}\mathbf{I}d)$ with $\lambda = 1.0$ and observation noise $\sigma = 0.5$.

To ensure a fair comparison across algorithms, we generate a fixed dataset that all algorithms observe. We first sample *true parameters* $\beta_i \sim \mathcal{N}(0, \mathbf{I}_d)$ for each arm $i$, representing the ground truth (which is unknown to the algorithms). To generate the contexts, we form $T = 2000$ context vectors with planted correlation structures to induce non-isotropic posteriors. We begin with $X_t^{(0)} \sim \mathcal{N}(0, \mathbf{I}_d)$ and apply transformations $X_t^{(1)} \leftarrow 1.7 \cdot X_t^{(1)}$ and $X_t^{(2)} \leftarrow 0.55 \cdot X_t^{(2)} + 0.6 \cdot X_t^{(1)}$. This creates elliptical posteriors as seen in Figure 2. Next, to generate the rewards, for each time step $t$, each algorithm chooses action $a_t$ and observes reward $r_t = X_t^\top \beta_{a_t} + \epsilon_t$ where $\epsilon_t \sim \mathcal{N}(0, \sigma^2)$.

**Computing the true posterior.** For the linear-Gaussian model, the posterior distribution admits a closed-form solution. Let $D_t^{(i)} = \{(X_s, r_s) : a_s = i, s \leq t\}$ denote the data collected for arm $i$ up to time $t$, with $n_t^{(i)} = |D_t^{(i)}|$ observations. The posterior for $\beta_i$ is Gaussian with

$$\Lambda_{\text{post}}^{(i)} = \lambda I_d + \frac{\eta}{\sigma^2} \sum_{(X,r) \in D_t^{(i)}} XX^\top, \tag{9}$$

$$\Sigma_{\text{post}}^{(i)} = (\Lambda_{\text{post}}^{(i)})^{-1}, \tag{10}$$

$$\mu_{\text{post}}^{(i)} = \Sigma_{\text{post}}^{(i)} \cdot \frac{\eta}{\sigma^2} \sum_{(X,r) \in D_t^{(i)}} Xr, \tag{11}$$

where $\eta$ is the inverse temperature parameter. This analytical posterior for each $i$ is $(\pi_t^\star)^{(i)} \sim \mathcal{N}(\mu_{post}^{(i)}, \Sigma_{post}^{(i)})$, which serves as our ground truth for comparison. The linear bandit problem with block-diagonal feature map $\phi(X, a)$ results in independent posteriors for each arm. When arm $j \neq i$ is played, the posterior $\beta_i$ remains unchanged. This allows us to analyze each arm's posterior separately.

For each algorithm, we extract posterior samples at the final timestep $T$. LinTS maintains the exact posterior analytically via Sherman-Morrison updates. Therefore, we directly sample from $\mathcal{N}(\mu_{\text{LinTS}}, \Sigma_{\text{LinTS}})$ to obtain 1500 posterior samples. However, for the MCMC variants such as LMCTS, MALATS, and preconditioned variants, we extract samples by repeatedly invoking the underlying MCMC sampling algorithm. Each call performs $K = 100$ MCMC iterations (Langevin steps, MALA steps, or leapfrog integrations for HMC), producing one sample from the approximate posterior $\widetilde{\pi}_t$. We provide 2D scatter plots of $(\beta_1, \beta_2)$ projections showing 1500 samples from $\pi_t^\star$ (green) and $\widetilde{\pi}_t$ (red). Overlapping clouds indicate good approximation, whereas separation reveals bias.

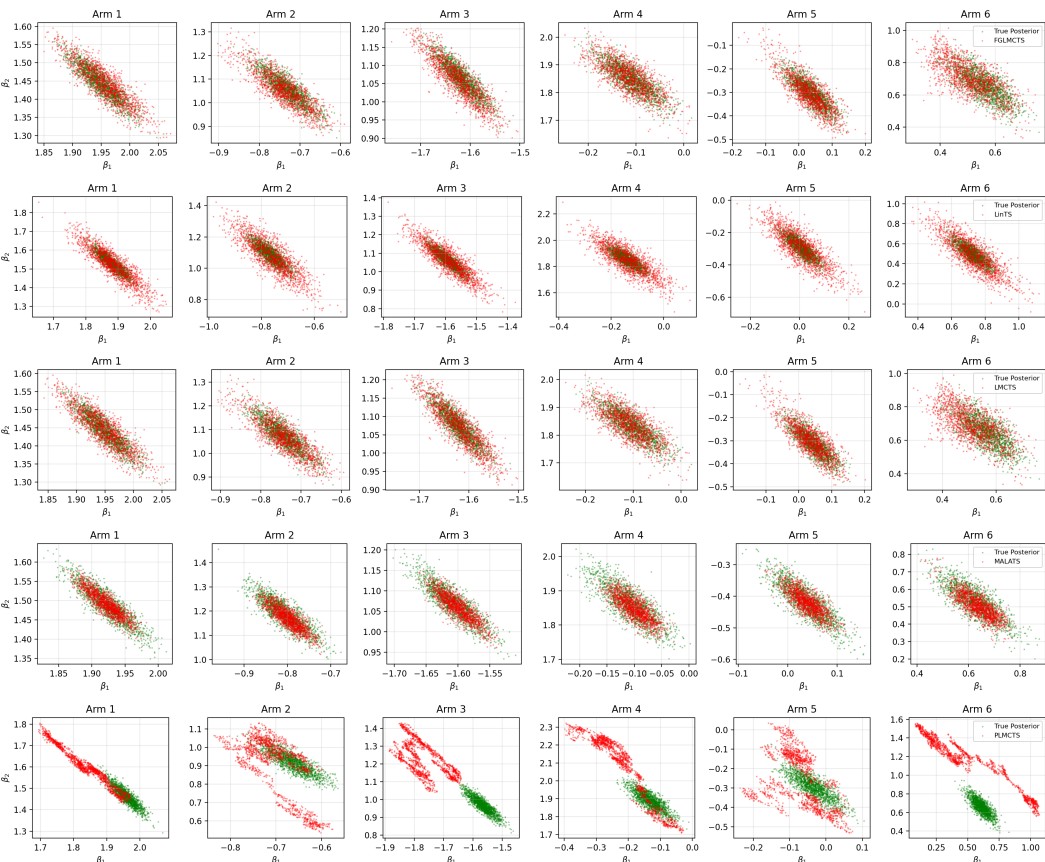

Figure 2: MCMC Sampling with the True Linear Posteriors. From top to bottom, we plot the true posteriors in green, and the sampled posteriors of Feel-Good LMCTS, Linear TS, LMCTS, MALATS, and PLMCTS in blue.

Based on Figure 2, we see that an algorithm may exhibit high cumulative regret due to under-exploration or due to poor approximation. Posterior analysis disambiguates these failure modes. For instance, PLMCTS' approximation of the true posterior is most egregious and displays a bias in its approximation; conversely, MALATS' and LinTS' posterior approximation are tight and match the true posterior. While LMCTS and FGLMCTS appear to be unbiased estimators of the true posterior, their variances can be much larger in under-explored arms.

### E.2 Logistic Bandits

This subsection provides our main experimental results for our logistic bandit experiments.

**Low-dimensional setting.** For the $d = 20$ setting, Tables 23 and 24 provide our experiments over logistic bandits for $\beta = 10^3$ and $\beta = 1$ (respectively), where we modify the feel-good parameter $\lambda$ in the loss-likelihood for values $\lambda \in \{0.01, 0.1, 0.5, 1.0\}$. As before, we see that setting the feel-good parameter $\lambda$ to be 0.01 in FGMALATS and SFGMALATS outperforms the case where $\lambda = 0$ (as well as other choices of $\lambda$).

Table 23: Cumulative Regret incurred by the Logistic bandits in $d = 20$ and $\beta = 10^3$. Values reported are the mean over 10 independent trials with standard deviation.

| Algorithm | $\lambda = 0.01$ | $\lambda = 0.1$ | $\lambda = 0.5$ | $\lambda = 1.0$ |
|---|---|---|---|---|
| FGLMCTS | **263.2** $\pm$ 70.4 | 285.2 $\pm$ 58.0 | **229.8** $\pm$ 37.0 | 265.6 $\pm$ 70.0 |
| FGMALATS | 293.1 $\pm$ 96.1 | 284.7 $\pm$107.2 | 295.6 $\pm$106.5 | 288.2 $\pm$ 95.9 |
| PFGLMCTS | 913.6 $\pm$322.3 | 791.5 $\pm$208.8 | 849.6 $\pm$230.2 | 868.9 $\pm$226.6 |
| PSFGLMCTS | 821.9 $\pm$246.9 | 755.2 $\pm$203.7 | 691.2 $\pm$127.6 | 733.1 $\pm$204.1 |
| SFGLMCTS | 295.8 $\pm$ 63.2 | **297.8** $\pm$ 78.3 | 305.6 $\pm$109.9 | 242.3 $\pm$ 58.1 |
| SFGMALATS | 297.7 $\pm$ 81.1 | 326.1 $\pm$ 52.5 | 283.5 $\pm$107.1 | **226.7** $\pm$ 67.5 |

Table 24: Cumulative Regret incurred by the Logistic bandits in $d = 20$ and $\beta = 1$. Values reported are the mean over 10 independent trials with standard deviation.

| Algorithm | $\lambda = 0.01$ | $\lambda = 0.1$ | $\lambda = 0.5$ | $\lambda = 1.0$ |
|---|---|---|---|---|
| FGLMCTS | 794.0 $\pm$ 169.3 | 828.8 $\pm$ 108.1 | 838.1 $\pm$ 101.5 | 755.1 $\pm$ 156.7 |
| FGMALATS | 299.0 $\pm$ 50.8 | 282.7 $\pm$ 49.5 | **254.6** $\pm$ 121.9 | 245.5 $\pm$ 90.5 |
| PFGLMCTS | 2412.8 $\pm$ 535.9 | 2347.9 $\pm$ 519.8 | 2844.7 $\pm$ 1068.6 | 3106.0 $\pm$ 975.9 |
| PSFGLMCTS | 2619.3 $\pm$ 1072.6 | 2832.3 $\pm$ 771.7 | 2478.7 $\pm$ 814.6 | 3119.9 $\pm$ 736.4 |
| SFGLMCTS | 831.7 $\pm$ 195.5 | 884.1 $\pm$ 159.9 | 747.9 $\pm$ 76.2 | 842.3 $\pm$ 118.3 |
| SFGMALATS | **237.5** $\pm$ 88.9 | **241.4** $\pm$ 68.5 | 246.2 $\pm$ 54.6 | **233.0** $\pm$ 39.9 |

**High-dimensional setting.** For the $d = 40$ setting, Tables 25 and 26 provide our experiments over logistic bandits for $\beta = 10^3$ and $\beta = 1$ (respectively), where we modify the feel-good parameter $\lambda$ in the loss-likelihood for values $\lambda \in \{0.01, 0.1, 0.5, 1.0\}$. Here, setting the feel-good parameter $\lambda$ to be either 0.01 or 0.1 favors each algorithm differently. These benefits are most evident in FGMALATS, SFGLMCTS, and SFGMALATS.

Table 25: Cumulative Regret incurred by the Logistic bandits in $d = 40$ and $\beta = 10^3$. Values reported are the mean over 10 independent trials with standard deviation.

| Algorithm | $\lambda = 0.01$ | $\lambda = 0.1$ | $\lambda = 0.5$ | $\lambda = 1.0$ |
|---|---|---|---|---|
| FGLMCTS | 391.1 $\pm$ 103.8 | 415.6 $\pm$ 66.3 | 396.8 $\pm$ 77.5 | 354.9 $\pm$ 73.6 |
| FGMALATS | 394.3 $\pm$ 69.5 | **365.7** $\pm$ 50.0 | 385.0 $\pm$ 51.1 | 353.1 $\pm$ 39.8 |
| PFGLMCTS | 964.3 $\pm$ 148.7 | 1025.6 $\pm$ 141.3 | 954.8 $\pm$ 230.7 | 1009.6 $\pm$ 173.7 |
| PSFGLMCTS | 1006.8 $\pm$ 242.5 | 972.3 $\pm$ 182.0 | 999.3 $\pm$ 167.4 | 1014.0 $\pm$ 193.9 |
| SFGLMCTS | 413.6 $\pm$ 64.8 | 400.5 $\pm$ 72.8 | 371.7 $\pm$ 65.9 | **340.3** $\pm$ 77.7 |
| SFGMALATS | **413.1** $\pm$ 77.6 | 367.6 $\pm$ 52.8 | **356.6** $\pm$ 54.0 | 364.9 $\pm$ 70.3 |

Table 26: Cumulative Regret incurred by the Logistic bandits in $d = 40$ and $\beta = 1$. Values reported are the mean over 10 independent trials with standard deviation.

| Algorithm | $\lambda = 0.01$ | $\lambda = 0.1$ | $\lambda = 0.5$ | $\lambda = 1.0$ |
|---|---|---|---|---|
| FGLMCTS | $1486.4 \pm 114.0$ | $1531.5 \pm 128.0$ | $1606.5 \pm 120.0$ | $1521.5 \pm 153.4$ |
| FGMALATS | $\mathbf{359.6} \pm 58.2$ | $372.1 \pm 61.1$ | $401.0 \pm 56.6$ | $428.0 \pm 43.5$ |
| PFGLMCTS | $2315.0 \pm 207.2$ | $2523.4 \pm 222.3$ | $2787.6 \pm 348.2$ | $2816.6 \pm 281.8$ |
| PSFGLMCTS | $3206.1 \pm 200.1$ | $1572.3 \pm 100.1$ | $1532.0 \pm 230.7$ | $1723.7 \pm 139.9$ |
| SFGLMCTS | $475.3 \pm 46.4$ | $480.1 \pm 52.8$ | $527.7 \pm 85.5$ | $540.0 \pm 91.1$ |
| SFGMALATS | $464.4 \pm 78.2$ | $\mathbf{399.1} \pm 62.1$ | $\mathbf{416.0} \pm 68.0$ | $\mathbf{385.5} \pm 58.2$ |

### E.3 Wheel Bandits

This subsection provides our experimental results for the wheel bandit problem. Table 27 shows our experiments for the wheel bandit in the $d = 2$ setting, where we fix the feel-good parameter $\lambda = 0.01$ and inverse temperature $\beta = 10^3$. We observe that PSFGLMCTS, PFGLMCTS, and FGMALATS generally outperform the other variants. This is also true as $\delta = 0.99$ and the performance of all algorithms increases (as expected). We observe here that the (surprising) performance of the preconditioned feel-good variants of LMCTS can be attributed to approximating the posterior shifted due to the preconditioning, which (due to the sensitivity of the wheel bandit task) is benign.

Table 27: Cumulative Regret incurred by the Wheel bandits in $d = 2$ dimensions. All the Feel-Good variants had $\lambda = 0.01, \beta = 10^3$, and all the smoothed Feel-Good variants have $s = 10$. The values reported are the mean over 10 independent trials with standard deviation.

| Algorithm | $\delta = 0.01$ | $\delta = 0.1$ | $\delta = 0.5$ | $\delta = 0.99$ |
|---|---|---|---|---|
| LMCTS | $3350.5 \pm 10.0$ | $3349.2 \pm 12.6$ | $2581.0 \pm 9.9$ | $316.0 \pm 4.7$ |
| SVRGLMCTS | $3392.5 \pm 16.9$ | $3334.7 \pm 42.2$ | $2568.3 \pm 23.6$ | $313.3 \pm 13.0$ |
| PLMCTS | $3375.5 \pm 18.1$ | $3352.1 \pm 2.7$ | $2577.0 \pm 14.2$ | $320.6 \pm 14.7$ |
| MALATS | $3404.6 \pm 12.8$ | $3316.2 \pm 14.5$ | $2614.9 \pm 21.9$ | $314.1 \pm 35.8$ |
| SFGMALATS | $3383.9 \pm 15.7$ | $3339.5 \pm 18.3$ | $2569.8 \pm 11.2$ | $328.3 \pm 7.5$ |
| FGMALATS | $3349.2 \pm 14.2$ | $\mathbf{3284.9} \pm 21.6$ | $\mathbf{2561.3} \pm 13.8$ | $309.0 \pm 6.9$ |
| PSFGLMCTS | $\mathbf{3340.4} \pm 16.8$ | $3347.0 \pm 19.4$ | $2569.3 \pm 12.7$ | $\mathbf{294.3} \pm 8.2$ |
| FGLMCTS | $3417.7 \pm 13.5$ | $3350.6 \pm 17.9$ | $2562.0 \pm 14.6$ | $322.0 \pm 9.1$ |
| PFGLMCTS | $3349.5 \pm 15.9$ | $3357.4 \pm 20.1$ | $2577.8 \pm 11.9$ | $304.3 \pm 7.8$ |
| SFGLMCTS | $3344.2 \pm 14.7$ | $3378.1 \pm 16.5$ | $2577.8 \pm 13.2$ | $321.4 \pm 8.5$ |

### E.4 Other Experiments

Table 28 provides details of the linear contextual bandits experiments with every LMC variant. We see here that LMCTS, PLMCTS, and SVRGLMCTS have the lowest standard deviation, and that LMCTS and SVRGLMCTS have the lowest regret. Next, Table 29 provides similar experiments over the MALA suite of algorithms. Our experiments show that MALATS, FGMALATS, and SFGMALATS conclusively outperform any of the experiments from the LMC suite. Moreover, PHMCTS, while competitive (and with minimal standard deviation), is not as performant as MALATS. Finally, we provide results for the LMC algorithms with damping for the linear contextual bandit setting in Table 31. These results show that overdamped (vanilla) Langevin methods significantly outperform any of the corresponding underdamped experiments.

Table 28: LMC-TS variants ($d = 20$ dimensions and $\beta = 10^3$)

| Algorithm | Cumulative regret |
|---|---|
| LMCTS | **62.6** $\pm$ 9.5 |
| PLMCTS | 134.4 $\pm$ 19.9 |
| FGLMCTS-L1B1 | 263.2 $\pm$ 70.4 |
| FGLMCTS-L2B1 | 285.2 $\pm$ 58.0 |
| FGLMCTS-L3B1 | 229.8 $\pm$ 37.0 |
| FGLMCTS-L4B1 | 265.6 $\pm$ 70.0 |
| PFGLMCTS-L1B1 | 913.6 $\pm$ 322.3 |
| PFGLMCTS-L2B1 | 791.5 $\pm$ 208.8 |
| PFGLMCTS-L3B1 | 849.6 $\pm$ 230.2 |
| PFGLMCTS-L4B1 | 868.9 $\pm$ 226.6 |
| SFGLMCTS-L1B1 | 295.8 $\pm$ 63.2 |
| SFGLMCTS-L2B1 | 297.8 $\pm$ 78.3 |
| SFGLMCTS-L3B1 | 305.6 $\pm$ 109.9 |
| SFGLMCTS-L4B1 | 242.3 $\pm$ 58.1 |
| PSFGLMCTS-L1B1 | 821.9 $\pm$ 246.9 |
| PSFGLMCTS-L2B1 | 755.2 $\pm$ 203.7 |
| PSFGLMCTS-L3B1 | 691.2 $\pm$ 127.6 |
| PSFGLMCTS-L4B1 | 733.1 $\pm$ 204.1 |
| SVRGLMCTS | 73.2 $\pm$ 31.1 |

Table 29: MALA-TS/HMC-TS variants ($d = 20$ and $\beta = 10^3$)

| Algorithm | Cumulative regret |
|---|---|
| MALATS | 61.3 $\pm$ 26.6 |
| FGMALATS-L1B1 | 62.6 $\pm$ 25.5 |
| FGMALATS-L2B1 | 63.7 $\pm$ 14.5 |
| FGMALATS-L3B1 | 212.8 $\pm$ 170.0 |
| FGMALATS-L4B1 | 528.3 $\pm$ 377.2 |
| SFGMALATS-L1B1 | **56.2** $\pm$ 22.8 |
| SFGMALATS-L2B1 | 68.4 $\pm$ 26.2 |
| SFGMALATS-L3B1 | 173.4 $\pm$ 110.1 |
| SFGMALATS-L4B1 | 486.2 $\pm$ 347.1 |
| HMCTS | 241.2 $\pm$ 107.0 |
| PHMCTS | 90.0 $\pm$ 9.2 |
| FGHMCTS | 262.5 $\pm$ 85.7 |
| PFGHMCTS | 282.7 $\pm$ 156.0 |
| SFGHMCTS | 395.9 $\pm$ 504.7 |
| PSFGHMCTS | 248.1 $\pm$ 147.4 |

Table 30: Other Benchmarked linear and logistic experiments (with $d = 20$ and $\beta = 10^3$)

| Algorithm | Cumulative regret in linear setting | Cumulative regret in logistic setting |
|---|---|---|
| LinUCB | **73.0** $\pm$ 13.8 | **176.9** $\pm$ 41.9 |
| EpsGreedy | 19879 $\pm$ 7454.3 | 2899.2 $\pm$ 677.9 |
| LinTS | 114.7 $\pm$ 8.8 | 179.9 $\pm$ 53.2 |
| Uniform | 5010 $\pm$ 2500 | 5008 $\pm$ 2500 |

Table 31: Linear experiments on Damping (with $d = 20$, $\lambda = 0.01$, $\gamma = 0.1$, and $\beta = 10^3$)

| Algorithm | Cumulative regret without damping | Cumulative regret with damping |
|---|---|---|
| LMCTS/ULMCTS | **62.6** ± 9.5 | 2609.9 ± 238.1 |
| FGLMCTS/UFGLMCTS | **62.7** ± 12.0 | 3155.0 ± 291.6 |
| SFGLMCTS/USFGLMCTS | **65.2** ± 12.3 | 2781.2 ± 164.8 |

## E.5 Neural Bandits

This subsection provides our main experimental results for our neural bandit experiments. Tables 32 and 33 detail the results of our experiments on classification tasks on 7 UCI datasets ((ADULT, SHUTTLE, MAGICTELESCOPE, MUSHROOM, COVERTYPE), the RESTAURANTRATINGS (SCI) and JESTER datasets), FINANCIAL dataset and two vision benchmarks, MNIST_784 and CIFAR-10. Our experiments show that FGLMCTS generally loses its competitive edge on the neural tasks: for instance, the simple/cumulative regret of SFGLMCTS and FGLMCTS is much worse than vanilla LMCTS on a number of tasks.

Table 32: Cumulative Regret incurred by the Neural Models. Values reported are the mean over 5 independent trials with standard deviation.

| Dataset | LMCTS | FGLMCTS | SFGLMCTS | Neural-$\epsilon$-Greedy | NeuralUCB |
|---|---|---|---|---|---|
| Adult | 2456.6 ± 36.5 | 3505.0 ± 2257.5 | 4505.6 ± 2772.0 | 2658.0 ± 362.7 | **2444.4** ± 160.1 |
| Covertype | 7594.0 ± 892.0 | 7567.8 ± 454.5 | 8006.0 ± 1035.6 | **4629.4** ± 132.3 | 4798.4 ± 102.2 |
| Magic Telescope | 2220.0 ± 40.7 | 2197.6 ± 167.7 | **2193.2** ± 34.0 | 2005.2 ± 53.5 | 2112.2 ± 16.6 |
| Mushroom | 324.6 ± 102.6 | 283.2 ± 20.2 | 440.6 ± 89.5 | **124.0** ± 41.4 | 145.6 ± 25.2 |
| Shuttle | **210.2** ± 49.0 | 214.4 ± 51.6 | 1503.0 ± 2721.0 | 372.4 ± 425.8 | 2981.2 ± 4225.9 |
| MNIST_784 | 2854.6 ± 2945.9 | **2542.6** ± 2366.2 | 2935.0 ± 3349.5 | 3248.0 ± 1709.0 | 5442.8 ± 356.2 |
| Financial | 474.6 ± 23.7 | 475.8 ± 7.6 | 478.4 ± 6.1 | 471.5 ± 4.7 | **431.5** ± 3.1 |
| Jester | 3468.8 ± 16.8 | 3469.7 ± 11.9 | 3497.7 ± 21.5 | **3492.3** ± 25.5 | 3505.9 ± 15.2 |
| RestaurantRatings | 8452.6 ± 237.8 | **8362.0** ± 389.8 | 8646.2 ± 268.9 | 8814.8 ± 2.2 | 8826.6 ± 1.8 |
| CIFAR-10 | **16962.8** ± 693.2 | 17344.8 ± 297.8 | 17686.8 ± 914.2 | 17217.4 ± 356.3 | 20815.0 ± 203.3 |

Table 33: Simple Regret incurred by the Neural Models. Values reported are the mean over 5 independent trials with standard deviation.

| Dataset | LMCTS | FGLMCTS | SFGLMCTS | Neural-$\epsilon$-Greedy | NeuralUCB |
|---|---|---|---|---|---|
| Adult | 121.6 ± 5.2 | 176.6 ± 98.0 | 220.6 ± 124.6 | 117.8 ± 7.2 | **113.6** ± 6.8 |
| Covertype | 232.0 ± 31.6 | 222.8 ± 38.1 | 245.6 ± 38.7 | **112.2** ± 12.2 | 120.8 ± 9.4 |
| Magic Telescope | 88.6 ± 2.4 | 91.4 ± 3.4 | **82.8** ± 8.9 | 86.4 ± 5.5 | 92.4 ± 10.8 |
| Mushroom | 1.2 ± 1.30 | 1.2 ± 1.30 | 1.2 ± 1.30 | **0.0** ± 0.0 | 1.8 ± 2.49 |
| Shuttle | 3.6 ± 1.0 | **3.0** ± 0.9 | 13.2 ± 18.1 | 2.8 ± 0.8 | 110.6 ± 194.7 |
| MNIST_784 | 109.4 ± 163.1 | **94.0** ± 109.6 | 124.6 ± 168.5 | 108.6 ± 98.0 | 235.6 ± 21.9 |
| Financial | 24.6 ± 2.5 | 24.1 ± 0.7 | 24.2 ± 1.5 | 24.3 ± 1.1 | **21.9** ± 0.7 |
| Jester | 173.3 ± 4.9 | 172.4 ± 5.2 | **171.5** ± 6.9 | 175.4 ± 3.9 | 171.4 ± 2.8 |
| RestaurantRatings | 421.2 ± 13.4 | **415.0** ± 22.6 | 439.0 ± 11.3 | 444.0 ± 0.0 | 439.0 ± 0.0 |
| CIFAR-10 | **300.8** ± 19.1 | 314.6 ± 15.4 | 325.0 ± 25.5 | 323.8 ± 11.9 | 403.4 ± 8.6 |

Table 34: Cumulative Regret incurred by the Neural Thompson Sampling models. Values reported are the mean over 5 independent trials with standard deviation.

| Dataset | NeuralTS | FG-NeuralTS | SFG-NeuralTS |
|---|---|---|---|
| Adult | $3128.0 \pm 1187.5$ | $2659.4 \pm 73.0$ | $\mathbf{2483.8} \pm 29.5$ |
| Covertype | $\mathbf{5867.4} \pm 319.6$ | $12816.0 \pm 13.8$ | $8868.4 \pm 271.0$ |
| Magic Telescope | $\mathbf{2089.2} \pm 32.1$ | $3732.0 \pm 50.9$ | $4668.2 \pm 1625.1$ |
| Mushroom | $\mathbf{117.2} \pm 33.4$ | $4018.6 \pm 100.3$ | $5234.2 \pm 1293.8$ |
| Shuttle | $\mathbf{757.0} \pm 1178.9$ | $2867.0 \pm 264.4$ | $8414.4 \pm 3482.5$ |
| MNIST_784 | $\mathbf{2505.4} \pm 570.9$ | $8960.4 \pm 45.0$ | $8990.2 \pm 40.0$ |
| Financial | $\mathbf{6324.0} \pm 917.4$ | $7131.2 \pm 196.8$ | $6403.2 \pm 365.8$ |
| Jester | $\mathbf{38.6} \pm 42.3$ | $2295.8 \pm 1593.0$ | $435.2 \pm 306.0$ |
| RestaurantRatings | $\mathbf{8695.0} \pm 33.2$ | $8675.80 \pm 19.31$ | $8759.6 \pm 23.7$ |
| CIFAR-10 | $\mathbf{20568.6} \pm 81.5$ | $21252.60 \pm 162.90$ | $21291.8 \pm 203.9$ |

Table 35: Simple Regret incurred by the Neural Thompson Sampling models. Values reported are the mean over 5 independent trials with standard deviation.

| Dataset | NeuralTS | FG-NeuralTS | SFG-NeuralTS |
|---|---|---|---|
| Adult | $\mathbf{107.6} \pm 12.8$ | $121.6 \pm 6.1$ | $127.8 \pm 6.3$ |
| Covertype | $\mathbf{141.6} \pm 9.6$ | $424.6 \pm 10.0$ | $258.6 \pm 20.1$ |
| Magic Telescope | $\mathbf{91.4} \pm 7.9$ | $174.6 \pm 13.0$ | $233.6 \pm 80.3$ |
| Mushroom | $\mathbf{0.0} \pm 0.0$ | $187.6 \pm 7.8$ | $265.8 \pm 56.8$ |
| Shuttle | $\mathbf{3.6} \pm 3.6$ | $111.4 \pm 8.5$ | $419.8 \pm 168.8$ |
| MNIST_784 | $\mathbf{36.4} \pm 5.8$ | $446.8 \pm 7.2$ | $449.0 \pm 11.0$ |
| Financial | $\mathbf{311.2} \pm 49.1$ | $356.2 \pm 8.5$ | $311.2 \pm 19.9$ |
| Jester | $\mathbf{0.0} \pm 0.0$ | $45.4 \pm 45.7$ | $0.2 \pm 0.4$ |
| RestaurantRatings | $432.0 \pm 3.8$ | $\mathbf{430.80} \pm 5.07$ | $432.2 \pm 5.8$ |
| CIFAR-10 | $\mathbf{397.4} \pm 9.6$ | $839.60 \pm 9.7$ | $6.50 \pm 9.0$ |

### E.6  Ablation Studies for Preconditioning

We detail our ablation study on the effect of preconditioning in MCMC-TS and MCMC-FGTS. Preconditioning is widely believed to be useful for faster convergence of various optimization routines (Bhattacharya and Jiang, 2023; Li et al., 2016; Millard et al., 2025; Pidstrigach, 2022; Titsias, 2023). Nevertheless, our experiments reveal that in linear and logistic bandits, adding a preconditioner leads to generally higher regrets (with an exception in HMCTS, where adding a preconditioner has a generally positive effect on the cumulative regret).

**Linear Bandits.** Table 36 details the effect of preconditioning in our linear experiments.

Table 36: Cumulative regret of linear bandits with and without preconditioning ($d = 20, \beta = 10^3$)

| Algorithm | No Preconditioning | Preconditioning |
|---|---|---|
| LMCTS | $\mathbf{62.6} \pm 9.5$ | $134.4 \pm 19.9$ |
| FGLMCTS-L1B1 | $\mathbf{263.2} \pm 70.4$ | $913.6 \pm 322.3$ |
| FGLMCTS-L2B1 | $\mathbf{285.2} \pm 58.0$ | $791.5 \pm 208.8$ |
| FGLMCTS-L3B1 | $\mathbf{229.8} \pm 37.0$ | $849.6 \pm 230.2$ |
| FGLMCTS-L4B1 | $\mathbf{265.6} \pm 70.0$ | $868.9 \pm 226.6$ |
| SFGLMCTS-L1B1 | $\mathbf{295.8} \pm 63.2$ | $821.9 \pm 246.9$ |
| SFGLMCTS-L2B1 | $\mathbf{297.8} \pm 78.3$ | $755.2 \pm 203.7$ |
| SFGLMCTS-L3B1 | $\mathbf{305.6} \pm 109.9$ | $691.2 \pm 127.6$ |
| SFGLMCTS-L4B1 | $\mathbf{242.3} \pm 58.1$ | $733.1 \pm 204.1$ |
| HMCTS | $241.2 \pm 107.0$ | $\mathbf{90.0} \pm 9.2$ |
| FGHMCTS | $\mathbf{262.5} \pm 85.7$ | $282.7 \pm 156.0$ |
| SFGHMCTS | $395.9 \pm 504.7$ | $\mathbf{248.1} \pm 147.4$ |

**Logistic Bandits.** Table 37 details the effect of preconditioning in our experiments over the logistic bandit setting.

Table 37: Cumulative regret of logistic bandits with and without preconditioning $d = 20, \beta = 10^3$.

| Algorithm | No Preconditioning | Preconditioning |
|---|---|---|
| LMCTS | **202.7** $\pm$ 44.1 | 889.7 $\pm$ 248.0 |
| FGLMCTS-L1B1 | **263.2** $\pm$ 70.4 | 913.6 $\pm$ 322.3 |
| FGLMCTS-L2B1 | **285.2** $\pm$ 58.0 | 791.5 $\pm$ 208.8 |
| FGLMCTS-L3B1 | **229.8** $\pm$ 37.0 | 849.6 $\pm$ 230.2 |
| FGLMCTS-L4B1 | **265.5** $\pm$ 70.0 | 868.9 $\pm$ 226.6 |
| SFGLMCTS-L1B1 | **295.8** $\pm$ 63.2 | 821.9 $\pm$ 246.9 |
| SFGLMCTS-L2B1 | **297.8** $\pm$ 78.3 | 755.2 $\pm$ 203.7 |
| SFGLMCTS-L3B1 | **305.6** $\pm$ 109.9 | 691.2 $\pm$ 127.6 |
| SFGLMCTS-L4B1 | **242.3** $\pm$ 58.1 | 733.1 $\pm$ 240.1 |

# F   Datasets

This section lists the datasets we benchmark our MCMC algorithms on.

Table 38: Statistics of the benchmark datasets. Context dimension equals #arms $\times$ #attributes.

| Dataset | Attributes ($d$) | Arms ($N$) | Context dim ($Nd$) | Instances |
|---|---|---|---|---|
| Adult | 14 | 2 | 28 | 48 842 |
| Covertype | 54 | 7 | 378 | 581 012 |
| Magic Telescope | 10 | 2 | 20 | 19 020 |
| Mushroom | 22 | 2 | 48 * | 8 124 |
| Shuttle | 9 | 7 | 63 | 58 000 |
| MNIST_784 | 784 | 10 | 7 840 | 70 000 |
| Jester | 32 | 8 | 256 | 19 181 |
| Financial | 21 | 8 | 168 | 3713 |
| CIFAR-10 | 3072 | 10 | 30 720 | 10 000 |
| Restaurant | 128 | 127 | 16 256 | 1 161 |
| Random-Synthetic-Linear-20 | 20 | 5 | 100 | 10 000 |
| Random-Synthetic-Linear-40 | 40 | 5 | 200 | 10 000 |
| Random-Synthetic-Logistic-20 | 20 | 50 | 1 000 | 10 000 |
| Random-Synthetic-Logistic-40 | 40 | 50 | 2 000 | 10 000 |
| Wheel-2-$\delta$a | 2 | 5 | 10 | 5 000 |
| Wheel-2-$\delta$b | 2 | 5 | 10 | 5 000 |
| Wheel-2-$\delta$c | 2 | 5 | 10 | 5 000 |
| Wheel-2-$\delta$d | 2 | 5 | 10 | 5 000 |

*After one-hot encoding of categorical attributes.

## F.1   Real-World Datasets

We add to the collection of datasets that Riquelme et al. (2018) tested Thompson Sampling on. For convenience, we provide a description of these datasets below.

**Mushroom.** The Mushroom Dataset (Lincoff, 1997) contains 22 mushroom features and labels for whether the mushroom is safe to consume: {"poisonous", "safe" }. As in Blundell et al. (2015), we create a bandit problem where the agent must decide whether to eat the mushroom. In this setting, eating a safe mushroom delivers a deterministic reward of $+5$, and eating a poisonous mushroom provides a randomized reward of $+5$ with probability $1/2$ and $-35$ reward with probability $1/2$. If the agent does not eat a mushroom, then the reward is $0$. In this case, we set $T = 10000$.

**Shuttle (Statlog).** The Shuttle Statlog Dataset (Asuncion et al., 2007) provides the value of $d = 9$ indicators during a space shuttle flight, and the goal is to predict the state of the radiator subsystem of

the shuttle. There are $k = 7$ possible states, and if the agent selects the right state, then reward 1 is generated. Otherwise, the agent obtains no reward ($r = 0$). We set $T = 10000$.

**Covertype.** The Covertype Dataset (Asuncion et al., 2007) classifies the cover type of northern Colorado forest areas in $k = 7$ classes, based on $d = 54$ features, including elevation, slope, aspect, and soil type. Again, the agent obtains reward 1 if the correct class is selected, and 0 otherwise. We run the bandit for $T = 15000$ iterations.

**Financial.** Following Riquelme et al. (2018), we created a Stock Dataset by pulling the prices of $d = 21$ publicly-traded companies in NYSE and NASDAQ for the last 14 years ($T = 3713$ days). For each day, the context was the open-to-close price changes for each stock. We synthetically created the arms to be linear combinations of the contexts, representing $k = 8$ different potential portfolios.

**Jester.** Following Riquelme et al. (2018), we created a recommendation system bandit problem as follows. The Jester Dataset (Goldberg et al. (2001)) provides continuous ratings in $[-10, 10]$ for 100 jokes from a total of 73421 users. We find a complete subset of $T = 19181$ users rating all 40 jokes. As in Riquelme et al. (2018), we take $d = 32$ of the ratings as the context of the user, and $k = 8$ as the arms. The agent recommends one joke, and obtains the reward corresponding to the rating of the user for the selected joke.

**Adult.** The Adult Dataset (Asuncion et al., 2007; Kohavi, 1996) comprises personal information from the US Census Bureau database, and we consider the $d = 14$ different occupations as feasible actions, based on 94 covariates. As in previous datasets, the agent obtains reward 1 for making the right prediction, and 0 otherwise. We set $T = 10000$.

**Telescope.** The MAGIC Telescope dataset (Asuncion et al., 2007) comprises of $d = 10$ real-valued features on cosmic ray events (length, energy, etc.), and a binary label for whether the event is a gamma-ray or a hadron. As in before, the agent obtains a reward 1 for making the right prediction, and 0 otherwise. We set $T = 10000$.

**MNIST_784**. MNIST (LeCun, 2010) comprises of $d = 784$ real-valued features corresponding to flattened $26 \times 26$ images of $k = 10$ hand-drawn digits from 0 to 9. The task again is to make a prediction $a \in \{0, \dots, 9\}$ of the digit in the image, where the agent gets reward 1 for making the right prediction and 0 otherwise. We set the horizon of the game to be $T = 10000$.

**CIFAR-10**. CIFAR-10 (Krizhevsky, 2009) comprises of $d = 3072$ real-valued features corresponding to flattened $32 \times 32 \times 3$ RGB images of $k = 10$ object classes (airplanes, cars, birds, cats, deer, dogs, frogs, horses, ships, and trucks). The task is to make a prediction $a \in \{0, \dots, 9\}$ of the object class in the image, where the agent gets reward 1 for making the correct prediction and 0 otherwise. We set the horizon of the game to be $T = 25000$.

**RestaurantRatings**. Restaurant (Medelln and Serna, 2011) contains of $d = 128$ real-valued features corresponding to user and restaurant contextual information from the UCI Restaurant and Consumer Data. The dataset contains $k = 127$ restaurants and the task is to make a recommendation $a \in \{0, \dots, 126\}$ for a restaurant, where the agent gets reward 1 for a positive rating and 0 otherwise. We set $T = 10000$.

### F.2 Synthetic Datasets

**Linear Contextual Bandit.** Fix a dimension $d$ and horizon $T = 10\,000$. The linear contextual bandit environment is given as follows: at round $t \in [T]$, the agent observes a context $\mathcal{X}_t \sim \mathcal{N}(\mathbf{0}_4, \mathbf{I}_4)$, chooses action $x_t \in [K]$ with $K = 5$, and receives a noisy linear reward $r_t = \phi(\mathcal{X}_t, x_t)^\top \theta^\star + \varepsilon_t$, where $\varepsilon_t \sim \mathcal{N}(0, \sigma^2)$, $\sigma = 0.5$ and $\theta^\star \in \mathbb{R}^d$. We place a Gaussian prior, $\theta_0 \sim \mathcal{N}(\mathbf{0}_d, \sigma_0^2 \mathbf{I})$ with $\sigma_0 = 0.01$. The feature map $\phi : \mathbb{R}^4 \times [K] \to \mathbb{R}^d$ is the standard block concatenation $(\phi(\mathcal{X}_t, 0), \dots, \phi(\mathcal{X}_t, K))$ where $\phi(\mathcal{X}_t, i) = e_i \cdot \mathcal{X}_t$ (where $e_i$ is the $i$'th standard basis vector). Now, $\theta^\star$ naturally decomposes into $k$ context-specific blocks. Since the likelihood and prior are Gaussian, TS admits a closed-form posterior update, giving a convenient ground-truth baseline for our approximate sampling methods. We derive the following synthetic datasets:

1. **Random-Synthetic-Linear-20:** set $d = 20$ in the linear contextual bandit environment.

2. **Random-Synthetic-Linear-40:** set $d = 40$ in the linear contextual bandit environment.

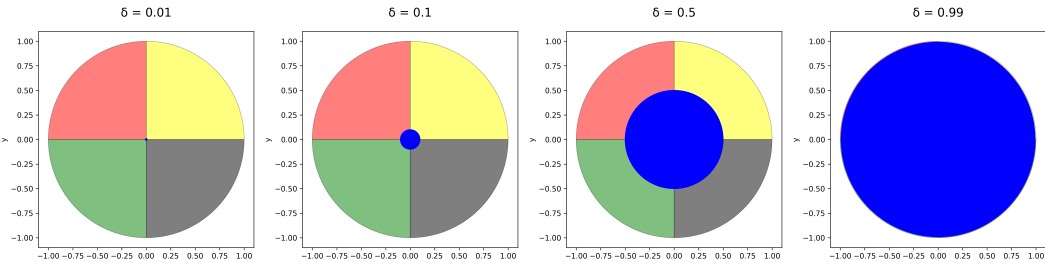

Figure 3: Wheel bandits for increasing values of $\delta \in (0, 1)$, where the optimal action for the blue, red, green, black, and yellow regions are given by actions $1, 2, 3, 4, 5$, respectively.

**Logistic Contextual Bandit.** Fix a dimension $d$. We further consider a logistic contextual bandit to introduce non-linear reward dependencies. With $T = 10\,000$ and $K = 50$ arms, at each round $t \in [T]$, the learner observes a collection of arm-specific context vectors $\mathcal{X}_{t,a} \sim \mathcal{N}(\mathbf{0}_d, I_d)$, where $a = 1, \ldots, 50$ each of which is then normalized to unit norm. The learner selects an arm $a_t \in [K]$ and obtains a Bernoulli reward $r_t \sim \mathrm{Bern}\big(\sigma\big(\phi(\widetilde{\mathcal{X}}_{t,a_t})^\top \theta^\star\big)\big)$, where $\sigma(u) = \frac{1}{1+e^{-u}}$, $\theta^\star \sim \mathcal{N}(\mathbf{0}_d, \mathbf{I}_d)$ (scaled to unit norm), and $\phi : \mathbb{R}^d \to \mathbb{R}^d$ where $\phi(\mathcal{X}_t) = \mathcal{X}_t$ is the identity feature map extracting the $d$-dimensional context for each arm. We place a Gaussian prior $\theta_0 \sim \mathcal{N}(\mathbf{0}_d, \sigma_0^2\mathbf{I})$, with $\sigma_0 = 0.01$. Similarly, we derive the following synthetic datasets:

1. **Random-Synthetic-Logistic-20:** set $d = 20$ in the logistic contextual bandit environment.
2. **Random-Synthetic-Logistic-40:** set $d = 40$ in the logistic contextual bandit environment.

**Wheel Bandit.** Fix $\delta > 0$. The wheel bandit, as defined in Riquelme et al. (2018), is a contextual bandit problem with the following structure (see Figure 3). Let $d = 2$ be the context dimension and $\delta \in (0, 1)$ be the exploration parameter. Contexts are sampled uniformly at random from the unit circle in $\mathbb{R}^2$, denoted as $X \sim \mathcal{U}(D)$. The problem consists of $k = 5$ possible actions $a_1, \ldots, a_5$. Action $a_1$ provides reward $r \sim \mathcal{N}(\mu_1, \sigma^2)$, independent of context. In the inner region, where $\|X\| \leq \delta$, $a_2, \ldots, a_5$ are sub-optimal with $r \sim \mathcal{N}(\mu_2, \sigma^2)$, where $\mu_2 < \mu_1$. In the outer region, where $\|X\| > \delta$, the optimal action depends on the quadrant of the context $X = (X_1, X_2)$ where for $(X_1 > 0, X_2 > 0)$, $a_2$ is optimal, $(X_1 > 0, X_2 < 0)$, $a_3$ is optimal, $(X_1 < 0, X_2 < 0)$, $a_4$ is optimal, and $(X_1 < 0, X_2 > 0)$, $a_5$ is optimal. The optimal action provides $r \sim \mathcal{N}(\mu_3, \sigma^2)$ for $\mu_3 \gg \mu_1$, whereas other actions (including $a_1$) provide $r \sim \mathcal{N}(\mu_2, \sigma^2)$. We set $\mu_1 = 1.2$, $\mu_2 = 1.0$, $\mu_3 = 50.0$, and $\sigma = 0.01$, and let the horizon of the game be $T = 5000$. As the probability of a context falling in the high-reward region is $1 - \delta^2$, we expect algorithms to get stuck repeatedly selecting $a_1$ for large $\delta$. We derive the following synthetic datasets:

1. **Wheel-2-$\delta$a:** set $\delta = 0.01$ in the wheel bandit environment.
2. **Wheel-2-$\delta$b:** set $\delta = 0.1$ in the wheel bandit environment.
3. **Wheel-2-$\delta$c.** set $\delta = 0.5$ in the wheel bandit environment.
4. **Wheel-2-$\delta$d.** set $\delta = 0.99$ in the wheel bandit environment.

