# OpenReview forum: "Feel-Good Thompson Sampling for Contextual Bandits: a Markov Chain Monte Carlo Showdown"
_NeurIPS.cc/2025/Datasets_and_Benchmarks_Track — NeurIPS 2025 Datasets and Benchmarks Track poster_

### Official Review · Reviewer_G6sk · 2025-06-25

**Rating:** 4
**Confidence:** 3

**Summary:**

This paper proposes a new benchmark for variants of Thomson Sampling (TS) with MCMC, including the vanilla TS, Feel-Good TS (FG-TS), and Smoothed Feel-Good TS (SFG-TS) in the contextual bandits. The benchmarking is evaluated across eight real-world and synthetic datasets with linear and neural bandit settings. The results show that FG-TS generally outperforms vanilla TS by having a lower regret in linear bandits, but does not perform well on neural bandits. In addition, the ablation studies show a trade-off of the optimism bonus, where larger bonuses help when posterior samples are accurate, but hurt when sampling noise dominates.

**Additional Feedback:**

- In Neural-UCB, have the authors considered different exploration parameters $\nu$ (see Algorithm 1 in [1])? I think tuning this parameter will significantly impact the confidence interval of the UCB term, yielding a balance between exploration/exploitation.
- Have the authors considered a higher-dimensional dataset than MNIST, such as CIFAR-10? I think this could yield the benefits of neural networks for contextual bandits.

---

References:

[1] Zhou et al., Neural Contextual Bandits with UCB-based Exploration, ICML, 2020.

**Dataset Code Accessibility:**

Yes

**Ethical Considerations:**

No, there are no or only very minor ethics concerns

**Final Justification:**

Most of my concerns are addressed and promised to be added in the revised version by the authors. Hence, I keep my original score and lean towards accepting this paper.

**Limitations Weaknesses:**

- Lack of ablation studies to evaluate the results with different exploration rates $\alpha$ in linear-bandits, different neural network architectures, and hyperparameter searching to fine-tune the neural network in neural-bandits.

**Strengths Contributions:**

- This paper is generally well-written and is clear to understand the important aspects.
- The experimental contributions of this paper are important for the contextual bandits community by extensive evaluation across different datasets, hyperparameter settings, and ablation studies.
- The results show that extending linear to neural-bandits is non-trivial due to a contradiction between the FG-TS results when compared to the vanilla TS.

---

> ### Author Rebuttal · Authors · 2025-07-30
>
> We thank reviewer G6sk for their feedback on our work. We are glad that they found our experimental contributions important for the community and appreciated our contributions on the linear-to-neural.
>
> Please find our response below.
>
> > Lack of ablation studies to evaluate the results with different exploration rates $\alpha$ in linear-bandits, different neural network architectures, and hyperparameter searching to fine-tune the neural network in neural-bandits.
>
> We thank the reviewer for pushing for more detail on our experimental rigor and setup. We did perform these ablations and apologize if this was not clear. We will revise the manuscript to state these points more explicitly.
>
> For MCMC-based Thompson Sampling, the level of exploration is controlled by properties of the posterior distribution and not by a single parameter like $\alpha$. We perform extensive ablations on the key parameters governing this: (1) feel-good bonus $\lambda$, which directly controls optimism (Table 4), (2) inverse-temperature $\beta$, which controls posterior variance (Table 1).
>
> To ensure a fair and reproducible comparison, we followed the established protocol from [1], which involved selecting the best performance for each method across architectures (2-layer and 4-layer MLPs) and activation functions (ReLU, LeakyReLU). Our hyperparameter search for learning rates and other settings was also conducted on a per-environment basis.
>
> > In Neural-UCB, have the authors considered different exploration parameters?
>
> In our work, our primary focus was the novel analysis of FG-TS, and we included baselines like Neural-UCB with standard parameters from the literature for context [1]. We thank the reviewer for bringing this up and agree that tuning the exploration parameter is important. We will add an ablation study on this parameter to Section 6 and the Appendix in the final version.
>
> > Have the authors considered a higher-dimensional dataset than MNIST, such as CIFAR-10? I think this could yield the benefits of neural networks for contextual bandits.
>
> We agree that evaluating on a high-dimensional image dataset, larger than MNIST, would be valuable. We will run our experiments on CIFAR-10 and incorporate these new results in the camera-ready version of the paper. This will further test the scalability of our methods and the generality of our findings in a complex visual domain.
>
> We hope this has eased the reviewer’s concerns, and would be happy to answer any further questions.
>
> Thank you,
>
> Authors
>
> ===========================
>
> References:
>
> 1) Langevin Monte Carlo for Contextual Bandits. Xu et. al. ICML 2022

---

### Official Review · Reviewer_rHY4 · 2025-06-27

**Rating:** 4
**Confidence:** 3

**Summary:**

The paper studies the effect of approximation error in posterior sampling for FG-TS and TS algorithms from a practical perspective. It shows that while FG-TS with various sampling techniques achieves good performance in linear and logistic settings, its performance may degrade and vary in neural bandits and real-world benchmarks. These results suggest that there is significant room for improvement in sampling techniques and the FG-TS algorithm in general, especially when dealing with more practical scenarios involving posterior sampling errors and model misspecification.

**Dataset Code Accessibility:**

Yes

**Ethical Considerations:**

No, there are no or only very minor ethics concerns

**Final Justification:**

After carefully considering the other reviewers’ comments and the authors’ response, I lean toward accepting the paper and will keep my score.

**Limitations Weaknesses:**

Although the intuition and hypothesis posed in the paper are intuitive and reasonable, the experimental setup does not clearly support these arguments.

First, the paper claims that the approximated posterior could affect the regret of the algorithm in general, but no approximation error of the posterior is reported in the experiments. The authors simply explain the TS parameters (e.g., $\lambda$, sampling algorithm choice), but the approximation error is not reported. I strongly believe that measuring the approximation error of the posterior and analyzing its correlation with overall regret could significantly strengthen the experimental results and better support the claims made in the paper.

Second, I found the claim in lines 302–306 interesting, but I could not see any supporting insights in the experiments. Perhaps an isolated experimental design to further investigate these claims could significantly strengthen the findings of the paper.

Third, in the neural contextual bandit setting, perhaps the authors could elaborate on the computational and statistical trade-offs of using simple versus sophisticated sampling techniques.

Fourth, the tables could be better presented, as it is currently hard to identify patterns and draw key insights from them.

Extra question: In line 95, I don’t see why “FG-TS favors aggressive exploration in contextual bandits, [so] it is expected to be more amenable for real-world datasets.” My intuition is that, since there is potential misspecification error, aggressive exploration is more likely to cause harm. Could you elaborate on this claim?

**Strengths Contributions:**

This paper is overall well written, with strong and clear motivation. Approximate posterior sampling is indeed an important practical issue when applying Thompson-sampling-based methods in practice. The paper also shows interesting results through experiments: while FG-TS achieves better performance in linear and logistic models, its performance may degenerate quickly in the Neural Bandit setting, where misspecification and neural posteriors are highly non-Gaussian.

---

> ### Author Rebuttal · Authors · 2025-07-30
>
> We thank reviewer rHY4 for their positive assessment and for their constructive feedback to strengthen the quality of our paper.
>
> > I strongly believe that measuring the approximation error of the posterior and analyzing its correlation with overall regret could significantly strengthen the experimental results and better support the claims made in the paper.
>
> We agree that directly measuring the posterior approximation error and correlating it with regret would be a powerful way to support our claims. The primary challenge, as you have noted, is that the true posterior is intractable in the logistic and neural settings, which is why MCMC is necessary, making direct error computation not possible. Our approach was therefore to use the choice of sampler as a proxy for approximation quality, by comparing algorithms with known theoretical differences in accuracy (e.g., fast-but-biased LMC vs. more accurate MALA).
>
> However, the analysis that the reviewer suggests is tractable to do in the linear settings. In the camera-ready version, we will add a figure in section 6 that plots the true linear posterior against the empirical distributions from our MCMC samplers. We will also report the correlation between a distance metric (eg, Wasserstein distance) and the final cumulative regret.
> > Supporting insights for claims in lines 302-306
>
> We thank the reviewer for finding this claim interesting and pushing us to clarify its empirical support.
>
> The first part of our claim, that neural network posteriors are highly non-Gaussian, is supported by [2] and [4], which independently show that distinct minima found by SGD are not isolated but are connected by continuous paths of low loss. They conclude that the optima form a single, complex, connected manifold rather than distinct valleys. This topology is non-Gaussian, since a Gaussian or mixture-of-Gaussians would imply isolated modes separated by high-loss barriers.
>
> The second part, that this posterior is poorly captured by the quadratic surrogates used in the FG derivation, is shown in [1, 3]. A quadratic surrogate implies a symmetric loss landscape around the minimum. However, [3] shows that the landscape is highly asymmetric as the solution found by SGD lies at the edge of a wide, flat basin near a region of sharp ascent. This asymmetry, where the geometry differs depending on the direction of perturbation, violates the properties of a quadratic function, which makes it a poor local approximation.
>
> Our key finding that FG-TS underperforms in the neural setting is the empirical consequence of applying an algorithm based on quadratic assumptions to this complex, non-quadratic setting. We will revise section 6 to make this connection more explicit.
>
> > Perhaps the authors could elaborate on the computational and statistical trade-offs of using simple versus sophisticated sampling techniques.
>
> Thanks for this suggestion! To help provide a more thorough benchmark, we can run experiments to compare average runtimes across the algorithms for large horizons and add a figure and discussion in Section 6,
>
> > Fourth, the tables could be better presented, as it is currently hard to identify patterns and draw key insights from them.
> Thanks for this suggestion! We can bold the highest values in the cumulative regret tables (as well as the runtime tables). We would definitely be open to alternate presentation suggestions you may have.
>
> > Extra question: In line 95, I don’t see why “FG-TS favors aggressive exploration in contextual bandits, [so] it is expected to be more amenable for real-world datasets.” My intuition is that, since there is potential misspecification error, aggressive exploration is more likely to cause harm. Could you elaborate on this claim?
>
> We thank and appreciate the reviewer for the extra question. The reviewer’s intuition is correct, and our study provides empirical evidence to support it in complex settings. There are 2 competing hypotheses: (1) the view that aggressive exploration (like FG-TS) should help escape local optima in real-world problems, and (2) the reviewer’s intuition that this same optimism could be harmful when faced with model misspecification. We show that in well-specified settings (linear/logistic bandits), the first hypothesis holds and FG-TS is beneficial. And a priori, one might expect this benefit to carry over to more complex settings. However, in the neural setting, our findings indicate that the optimism interacts poorly with approximation errors and implicit SGD noise, leading to worse performance. We will revise line 95 to frame our paper as an investigation of this trade-off.
>
> Thank you again for your valuable guidance and constructive feedback. We hope this eases any concerns in our work, and we would be happy to answer any further questions or hear any feedback you may have.
>
> Thank you,
>
> Authors
>
> ======================================
>
> References:
>
> 1) Provable and Practical: Efficient Exploration in Reinforcement Learning via Langevin Monte Carlo. Ishfaq et. al.. ICLR 2024
> 2) Loss Surfaces, Mode Connectivity, and Fast Ensembling of DNNs. Garipov et. al.. Neurips 2018.
> 3) Averaging Weights Leads to Wider Optima and Better Generalization. Izmailov et. al.. UAI 2018.
> 4) Essentially No Barriers in Neural Network Energy Landscape. Draxler et. al.. ICML 2018.

---

### Official Review · Reviewer_iswW · 2025-06-30

**Rating:** 4
**Confidence:** 2

**Summary:**

Feel-good Thompson Sampling is a contextual bandit strategy that modifies the likelihood function to encourage more exploration.
When posteriors are exact, Feel-good Thompson Sampling achieves optimal regret in the linear setting.
However, its practical performance in more realistic settings with approximate posteriors is not well understood.
This paper aims to close that gap by providing a benchmark consisting of linear and logistic bandits that allows for an empirical evaluation of various Feel-good Thompson Sampling strategies.

**Dataset Code Accessibility:**

Yes

**Dataset Code Comments:**

- To foster adoption, I’d suggest packaging the code as a proper software package that can, for example, be installed via pip.

- The LINEAR_LOGISTIC benchmark has a separate requirements file that appears to be a pip freeze output. I recommend adding clear installation instructions to the README and possibly avoiding strict version pinning.

- Overall, the code looks ok, but could be improved at some points. I'd suggest adding at least some unit tests for the core components.

**Ethical Considerations:**

No, there are no or only very minor ethics concerns

**Final Justification:**

The authors addresses most of my concerns and provided additional empirical results.

**Limitations Weaknesses:**

- It seems to me that the main contribution of the paper is an empirical evaluation of existing strategies from the literature, rather than the benchmarking suite itself. While this is a valid contribution, I’m not sure whether this track is the most appropriate fit, or if the main track would be more suitable. For a paper focused on a benchmarking suite, I would have liked to see more technical detail on the suite itself—for example, whether comparisons can be parallelized, and whether results can be automatically compiled and visualized.

- The benchmarks considered are either purely synthetic or resemble simple use cases, such as MNIST, which I also find rather artificial. I would have liked to see more challenging real-world use cases that highlight the practical usefulness of Feel-good Thompson Sampling.

**Strengths Contributions:**

- As someone who is not deeply familiar with the bandit literature, it appears that the paper considers a diverse set of bandit strategies, explores several variations, and performs a thorough empirical evaluation.

- The contribution of the paper is well-motivated and seems to address a gap in the literature.

---

> ### Author Rebuttal · Authors · 2025-07-30
>
> We thank reviewer iswW for their feedback on our work and for recognizing our work as a thorough empirical evaluation.
>
> > I’m not sure whether this track is the most appropriate fit, or if the main track would be more suitable
>
> We appreciate the reviewer for pointing this out. We believed that this datasets and benchmarks track would be more suitable for our work since the call for papers encouraged submissions that fit the following categories: "Benchmarks on new or existing datasets, as well as benchmarking tools...", "Systematic analyses of existing systems on novel datasets yielding important new insight", “In-depth analyses of machine learning challenges... that yield important new insight." However, we will add more technical detail on the suite, and provide instructions in the paper for how to set up the existing experiments, as well as add new benchmarks and algorithms. Our suite indeed does parallelize the algorithms, and we enable a wandb setup that allows the user to view the outputs of each trajectory as well as a compiled figure with the mean cumulative regrets across algorithms which is also saved locally.
>
> > The benchmarks considered are either purely synthetic or resemble simple use cases, such as MNIST, which I also find rather artificial. I would have liked to see more challenging real-world use cases that highlight the practical usefulness of Feel-good Thompson Sampling.
>
> Thanks for your suggestion. We appreciate the reviewer’s suggestion for complex real-world use cases. Our choice of benchmarks was a deliberate methodological decision to build an interpretable bridge from theory to practice in the Datasets and Benchmark track.
>
> We have since run the suite of MCMC contextual bandit algorithms on the financial and jester datasets [1, 2] which reflect more realistic use-cases, and which also corroborate our claims. Additionally, we will run the suite on the CIFAR-10 dataset [3] and will update the camera-ready version of the manuscript to reflect these changes. Following [4], we will also run the suite on the RestaurantRatings dataset, which is less standard but more realistic, and we will add a table of results and corresponding discussions in the Appendix.
>
> We started with purely synthetic cases (linear/logistic) because these controlled environments have either true posteriors that are known (linear) or well-behaved (logistic), which allows us to isolate the effect of the sampler's approximation error on the algorithm's performance. Conversely, starting immediately with a complex, misspecified real-world problem makes it difficult to disentangle performance issues caused by the sampler from those caused by model mismatch.
>
> > Regarding the comments on fostering adoption, and code/accessibility:
>
> We are deeply grateful for the reviewer’s suggestions on improving the codebase! We are committed to making our code accessible and useful as possible for the community. In the camera-ready version and in our current github which we will update, we will:
>
> 1) Package the code for pip installation.
> 2) Clean up the dependencies. We will provide a clean requirements.txt file (or pyproject.toml) with clear, minimal installation instructions in the main README.md, removing the strict version pinning from the pip freeze output. Note that most of this is already done in the existing github version.
> 3) Add unit tests for core components like the MCMC samplers and loss functions to verify correctness and make it easier for others to build upon our work.
>
> We hope this eases any concerns in our work. We believe these clarifications and additions strengthen and further establish its contribution as a robust, reproducible benchmark for the community. Please do not hesitate to reach out with any additional questions or feedback.
>
> Thank you,
>
> Authors
>
> ======================================
>
> References:
>
> 1) Deep Bayesian Bandits Showdown: An Empirical Comparison of Bayesian Deep Networks for Thompson Sampling. Riquelme et. al. ICLR 2018
> 2) Eigentaste: A Constant Time Collaborative Filtering Algorithm. Goldberg et. al. Information Retrieval, 2001
> 3) Cifar-100 and cifar-10 (Canadian Institute for Advanced Research), 2009.
> 4) Tight Regret and Complexity Bounds for Thompson Sampling via Langevin Monte Carlo. Huix et. al. AISTATS 2023

---

> > ### Author Response · Authors · 2025-08-04
> > **Follow-Up on Author Response**
> >
> > Dear Reviewer iswW,
> >
> >
> > We hope this message finds you well.
> >
> >
> > As the author-reviewer discussion period is coming to an end, we would like to kindly ask whether we have adequately addressed your concerns. If our responses have adequately addressed your concerns, we kindly ask you to consider updating your score.
> >
> >
> > If there are any remaining concerns, we would greatly appreciate it if you could share them with us, so we may have enough time to provide a detailed response. Thank you once again for your time and valuable feedback!
> >
> >
> > Thanks,
> >
> > Authors

---

### Official Review · Reviewer_hB3c · 2025-07-02

**Rating:** 5
**Confidence:** 2

**Summary:**

This work evaluates the performance of Feel-Good Thompson Sampling (FG-TS) and its smoothed variant (SFG-TS) in the contextual bandit setting. While FG-TS has been shown to achieve minimax-optimal regret in linear bandit problems with exact posteriors, its behavior under approximate posterior sampling—common in high-dimensional or neural settings—has not been systematically studied. The authors benchmark FG-TS and SFG-TS against vanilla Thompson Sampling (TS) and other exploration strategies across eight synthetic and real-world datasets, including logistic and neural contextual bandits. They also analyze the impact of various algorithmic factors, such as preconditioning, feel-good bonus scaling, and stochastic gradient-based samplers. The paper provides theoretical insights, empirical results, and open-source code for reproducibility.

**Additional Feedback:**

This work provides a significant contribution to the study of exploration strategies in contextual bandits, particularly by benchmarking FG-TS and SFG-TS under approximate posterior settings. The thorough empirical evaluations and theoretical insights make it a valuable resource for both researchers and practitioners. However, the limitations in neural bandit performance, scalability, and practical applicability leave room for improvement.

**Dataset Code Accessibility:**

Yes

**Ethical Considerations:**

No, there are no or only very minor ethics concerns

**Final Justification:**

This work offers the first comprehensive empirical and theoretical study of Feel-Good Thompson Sampling (FG-TS) and its smoothed variant in both linear and nonlinear contextual bandits, addressing important gaps related to approximate posterior sampling. With extensive experiments, insightful analyses on algorithmic factors, and strong reproducibility commitments, I believe this paper meets the acceptance criteria.

**Limitations Weaknesses:**

**1. Performance Variability**: The authors acknowledge significant run-to-run variability in cumulative regret on certain datasets (e.g., MNIST and Covertype). This suggests sensitivity to hyperparameters such as learning rates, mini-batch sizes, and MCMC step sizes, which were not exhaustively tuned.

**2. Limited Neural Bandit Performance**: FG-TS and its smoothed variants underperform compared to vanilla TS in neural bandits, which are increasingly relevant in real-world applications. The paper provides limited discussion on how these limitations could be addressed, such as by better modeling non-Gaussian posteriors or adjusting the optimism bonus.

**3. Lack of Scalability Analysis:** While the paper evaluates runtime complexities theoretically, it does not provide a detailed empirical analysis of scalability across large-scale datasets or higher-dimensional problems. For example, methods like HMC, which require costly matrix inversions, may be impractical for real-world deployment.

**Strengths Contributions:**

**1. Novel Empirical Benchmarking**: The work provides the first systematic empirical study of FG-TS and SFG-TS in both linear and non-linear contextual bandit regimes. This fills a critical gap in the literature, where prior studies primarily focus on exact posteriors.

**2. Comprehensive Experiments**: The authors evaluate FG-TS/SFG-TS across a wide range of settings, including linear, logistic, and neural bandits, as well as six real-world datasets. Detailed comparisons are made against strong baselines such as LinUCB, Neural-ϵ-Greedy, and NeuralUCB.

**3. Theoretical and Practical Contributions**: The work extends FG-TS to approximate posterior sampling via Markov Chain Monte Carlo (MCMC) methods, including Langevin Monte Carlo (LMC), Metropolis-Adjusted Langevin Algorithms (MALA), and Hamiltonian Monte Carlo (HMC). It also provides runtime analyses, regret guarantees, and convergence proofs in the appendix.

**4. Key Insights**:
>The results reveal a trade-off between optimism and posterior accuracy. FG-TS performs well in linear and logistic settings under high-fidelity posteriors but struggles in neural contextual bandits due to posterior mismatch and excessive exploration.

>Preconditioning improves performance by addressing ill-conditioned posteriors, especially in high dimensions.

>The smoothened FG-TS objective (SFG-TS) helps in under-concentrated posterior regimes but may degrade performance in well-concentrated settings.

**5. Reproducibility**: The authors provide open-source PyTorch implementations and detailed experimental setups, ensuring reproducibility and encouraging further research.

**6. Clear Presentation of Metrics**: The work reports both cumulative regret and simple regret, providing a nuanced understanding of algorithm performance.

---

> ### Author Rebuttal · Authors · 2025-07-30
>
> We thank reviewer hB3c for their constructive feedback on our work. We appreciate that they found our study to be a valuable resource for researchers and practitioners, and are grateful for the opportunity to address their concerns and clarify our contributions.
>
> > The authors acknowledge significant run-to-run variability in cumulative regret on certain datasets (e.g., MNIST and Covertype)
>
> 1) Firstly, we would like to preface our response by saying that we have (since submission) added experiments that also ablate on the inverse temperature parameter $\beta$ as well as on an underdamped sampling friction coefficient $\gamma$, and have updated all our experiments to choose learning rates appropriate to each environment. We will revise our manuscript in the camera-ready version to reflect these changes,
>
> 2) Moreover, the run-to-run sensitivity that the reviewer refers to here does not stem from tuning, but is rather a fundamental challenge that emerges from three distinct sources of noise in an online setting, in order of decreasing impact: (a) the inherent stochasticity of the rewards; (b) the noise from mini-batch gradients within the MCMC samplers; (c) the approximation error from using a finite number of MCMC steps to sample from the posterior.
>
> 3) In fact, the high variability on complex datasets like MNIST and COVERTYPE highlights that the feedback loop in contextual bandits can amplify these approximation errors, which is an important consideration for practitioners in the community. We reveal this sensitivity as a data point for the community to note that while MCMC methods are powerful, their application in online learning requires careful management of approximation quality for stability. Such variability in vanilla Thompson Sampling has already been noted [1, 2, 3]. To our knowledge, our paper is the first to systematically document and analyse this for FGTS.
>
> > The paper provides limited discussion on how these limitations could be addressed, such as by better modeling non-Gaussian posteriors or adjusting the optimism bonus.
>
> As we show in the paper, since FG-TS has an inherent advantage over TS in linear contextual bandits, we would a priori expect this advantage to also transfer to the neural bandit setting; however, our experiments refute this belief. [4] recently proposed studying an underdamped feel-good Thompson sampling which incorporates a friction coefficient which can improve performance of FG-TS. However, despite our experiments that ablate over optimism bonus’, inverse sampling temperature, and underdamped friction coefficients, FG-TS performs consistently worse than TS in the neural bandit settings. Therefore, in the neural bandit setting, and as we discuss in the paper, we recommend Neural Thompson Sampling methods [5] over FG-TS.
>
> > While the paper evaluates runtime complexities theoretically, it does not provide a detailed empirical analysis of scalability across large-scale datasets or higher-dimensional problems. For example, methods like HMC, which require costly matrix inversions, may be impractical for real-world deployment.
>
> Thanks for your comments. We will additionally run experiments to compare average runtimes across the algorithms for large horizons and add a figure and discussion for the runtimes in Section 6. We will also add a remark that for some of the algorithms, the computational bottlenecks may not necessarily be matrix primitives rather than the large number of iterations. For example, in HMC, the real concern is that each HMC update requires $O(d)$ gradient calls per leapfrog step, often with 10-50 steps per sample which is the actual bottleneck.
>
>
> We hope this eases any concerns in our work, and we would be happy to answer any further questions.
>
> Thank you,
>
> Authors
>
> ======================================
>
> References:
>
> 1) Stochastic Regret Minimization via Thompson Sampling. Guha & Munagala. COLT 2014
> 2) On Approximate Thompson Sampling with Langevin Algorithms. Mazumdar et. al. ICML 2020
> 3) More Efficient Randomized Exploration for Reinforcement Learning via Approximate Sampling. Ishfaq et. al., RLC 2024
> 4) Provable and Practical: Efficient Exploration in Reinforcement Learning via Langevin Monte Carlo. Ishfaq et. al.. ICLR 2024
> 5)  Neural Thompson Sampling. Zhang et. al. ICLR 2021

---

> > ### Author Response · Authors · 2025-08-04
> > **Follow-Up on Author Response**
> >
> > Dear Reviewer hB3c,
> >
> > We hope this message finds you well.
> >
> > As the author-reviewer discussion period is coming to an end, we would like to kindly ask whether we have addressed your concerns. If our responses have adequately addressed your concerns, we kindly ask you to consider updating your score.
> >
> > If there are any remaining concerns, we would greatly appreciate it if you could share them with us, so we may have enough time to provide a detailed response. Thank you once again for your time and valuable feedback!
> >
> > Thanks,
> >
> > Authors

---

> > ### Comment · Reviewer_hB3c · 2025-08-05
> >
> > I appreciate the authors’ detailed and thoughtful response. Their rebuttal has greatly enhanced the clarity and presentation of the work, offering a clearer view of the main contributions and technical details. Moreover, their discussion of potential future research directions adds further value to the paper.
> >
> > As my concerns have been fully addressed, I am happy to increase my score accordingly.

---

### Note · Authors · 2025-08-12

We sincerely thank the reviewers for their thoughtful and constructive reviews, and appreciate their detailed suggestions for enhancing the impact of our work for the larger community. We have addressed specific concerns and questions in individual rebuttals.

Our goal was to move beyond the idealized theoretical analysis of Feel-Good Thompson Sampling in existing literature and to provide the first systematic, reproducible benchmark to understand its performance under more realistic conditions of approximate posteriors. We are encouraged that the reviewers appreciated these contributions, and are excited to incorporate the following into the final version:

1) **Expanding the range of benchmarks:** Based on our discussion with reviewers iswW and G6sk, we will _expand our benchmark to include more challenging and realistic benchmarks_. We will now include the results of our simulations on the **financial**, **jester**, and **CIFAR-10** datasets. We hope this addresses the reviewers’ call for more realistic benchmarks when testing the scalability and generalizability of these methods under more complex settings.
2) **Adding empirical analyses and supporting evidence:** As suggested by reviewer rHY4, we will _add an analysis of the posterior approximation error in the linear setting_, by plotting a distance metric between the true and sampled posteriors against the final regret to link sample fidelity to performance. Moreover, while the time complexities are not the central thesis of this work, a comprehensive overview of the empirical runtimes across all algorithms will be provided for completeness. This will further demonstrate the practical computational trade-offs between simple samplers like LMC and complex ones like HMC, which reviewers rHy4 and hB3c have inquired about.
3) **Improving the usability and reproducibility of our repo:** We are grateful for the practical suggestions from reviewer iswW to improve the accessibility of our codebase. We will _package the code for pip installation_, _add unit tests for core components_, and _provide a clean requirements.txt with installation instructions_.

We hope that by incorporating these changes, our work will present novel findings, as well as an extensible and user-friendly benchmark that may be useful for future research in contextual bandits.

Thanks for your guidance and for helping us improve our contribution.

Best,

Authors

---

### Decision · Program_Chairs · 2025-09-18

**Decision:**

Accept (poster)

**Comment:**

Main reason to accept: The paper sets out to benchmark the practical performance of feel-good Thompson sampling (FG-TS) in  realistic settings with approximate posteriors that may be very misaligned, in linear and nonlinear contextual bandits problems. (FG-TS is known to achieve optimal regret in the linear setting when the posteriors are exact.) It provides the first such comprehensive empirical and theoretical study of FG-TS and its smoothed variant.

Main reason to reject: Limitations include that the study seems mostly focused on comparing and evaluating existing strategies in the literature, rather than benchmarking the evaluation methodology itself, and that the study relies on either purely synthetic or resemble simple use cases, such as MNIST.

On balance, I feel the contributions outweigh the negatives, thus recommending accept.